# A new intermittent regime of convective ventilation threatens the Black Sea oxygenation status

Arthur Capet[1], Luc Vandenbulcke[1], and Marilaure Grégoire[1]

[1]MAST, FOCUS, University of Liège, Belgium

**Correspondence:** A. Capet (acapet@uliege.be)

**Abstract.** The Black Sea is entirely anoxic, except for a thin ($\sim$100 m) ventilated surface layer. Since 1955, the oxygen content of this upper layer has decreased by 44 %. The reasons hypothesized for this decrease are, first, a period of eutrophication from the mid-1970s to the early 1990s and, second, a reduction in the ventilation processes, suspected for the recent years (post-2005). Here, we show that the Black Sea convective ventilation regime has been drastically altered by atmospheric warming during the last decade. Since 2008, the prevailing regime is below the range of variability recorded since 1955, and is characterized by consecutive years during which the usual partial renewal of intermediate water does not occur. Oxygen records from the last decade are used to detail the relationship between cold water formation events and oxygenation at different density levels, to highlight the role of convective ventilation in the oxygen budget of the intermediate layers, and to emphasize the impact that a persistence of the reduced ventilation regime would bear on the oxygenation structure of the Black Sea and on its biogeochemical balance.

*Copyright statement.* TEXT

## 1 Introduction

By reducing water density and increasing vertical stratification, global warming is expected to impede ventilation mechanisms in the world ocean and regional seas with potential consequences for the oxygenation of the subsurface layer (Bopp et al., 2002; Keeling et al., 2010; Breitburg et al., 2018). On a global scale, the reduction of ventilation processes constitutes a larger contribution to marine deoxygenation than the warming-induced reduction of oxygen solubility (Bopp et al., 2013). While the reduction in ventilation mechanisms is often evidenced, it remains challenging to determine whether such changes are the signal of natural variability or rather witness a significant regime change attributed to global warming (Long et al., 2016).

The Black Sea provides a miniature global ocean framework where processes of global interest occur at a scale more amendable for investigation. Its deep basin is permanently stratified and the ventilation of the subsurface layer relies in substantial parts on the convective transport of cold, oxygen-rich water formed each winter at the surface. Between 1955 and 2015, the Black Sea oxygen inventory has declined by $40\%$ (Capet et al., 2016), which echoes the significant deoxygenation trend that affected the world ocean over a similar period (Schmidtko et al., 2017).

The permanent stratification of the Black Sea results from two external inflows (Öszoy and Ünlüata, 1997). The saline Mediterranean inflow enters the Black Sea by the lower part of the Bosporus Strait, the sole and narrow opening of the Black Sea towards the global ocean. The terrestrial fresh water inflow, for its greatest part, enters the Black Sea on its northwestern shelf. The contrast in density (salinity) between these two inflows maintains a permanent stratification in the open basin (halocline) that prevents ventilation of the deep layers. This lack of ventilation induces the permanent anoxic conditions that characterize 90% of the Black Sea waters. Between the oxic and anoxic (euxinic) layers, a suboxic zone, where both dissolved oxygen and hydrogen sulphide are below reliable detection limits (Murray et al., 1989), is maintained by biogeochemical processes(Stanev et al., 2018).

Just above the main halocline, the ventilation of the Black Sea subsurface waters ($\sim 50$–$100$m), is ensured by convective circulation. It proceeds from the sinking of surface waters, made colder and denser by loss of heat towards the atmosphere in winter time (Ivanov et al., 2000). A similar ventilation process is observed, for instance, in the Mediterranean Gulf of Lion (e.g., MEDOC group et al., 1970; Coppola et al., 2017; Testor et al., 2017). In the Black Sea, however, the dense oxygenated waters never reach the deepest parts, as their sinking is restrained at intermediate depth by the permanent halocline. The accumulation of cold waters at intermediate depth forms the so-called Cold Intermediate Layer (CIL). The process of CIL formation thus provides an annual ventilating mechanism that structures the vertical distribution of oxygen (Konovalov and Murray, 2001; Gregg and Yakushev, 2005; Capet et al., 2016) and, by extension, that of nutrients (Konovalov and Murray, 2001; Pakhomova et al., 2014) and living components of the ecosystem (Sakınan and Gücü, 2017).

The semi-enclosed character of the Black Sea, superimposed with the fact that ventilation is limited to the upper $\sim100$ m, makes it highly sensitive to variations in external forcing. In particular, the variations of atmospheric conditions (e.g. air temperature, wind curl) result in pronounced and relatively fast inter-annual alterations of the Black Sea physical structure (Oguz et al., 2006; Capet et al., 2012; Kubryakov et al., 2016).

While several studies evidenced a warming trend in the Black Sea surface temperature (Belkin, 2009; von Schuckmann et al., 2018), Miladinova et al. (2017) showed that the Black Sea intermediate waters present an even stronger warming trend. This difference between the surface and subsurface temperature trends can be explained by the fact that the CIL dynamics buffers the atmospheric warming trends and minimize its signature in sea surface temperature (Nardelli et al., 2010).

The inter-annual variability in CIL formation can be explained for its larger part on the basis of winter air temperature anomalies (Oguz and Besiktepe, 1999; Capet et al., 2014), although intensity of the basin-wide cyclonic circulation (Staneva and Stanev, 1997; Capet et al., 2012; Korotaev et al., 2014), fresh water budget (Belokopytov, 2011) and the intensity of short-term meso-scale intrusions also play a role (Gregg and Yakushev, 2005; Ostrovskii and Zatsepin, 2016; Akpinar et al., 2017). An extensive description of the CIL dynamics, detailing the contributions and variability of the mechanisms mentioned above was recently provided by Miladinova et al. (2018). One aspect is particularly relevant to our study: in winter time, the deepening of the mixed layer and the uplifting of isopycnals in the basin center (as the cyclonic circulation intensifies), expose subsurface waters to atmospheric cooling. If a well-formed CIL was present during the previous year, subsurface waters exposed to atmospheric cooling are already cold, which increases the amount of newly formed CIL waters (Stanev et al., 2003). Due to this positive feedback and to the accumulation of CIL waters formed during successive years, the inter-annual CIL dynamics

is better described when winter air temperature anomalies are accumulated over 3 to 4 years, rather than considered on a year-to-year basis (Capet et al., 2014), which is in agreement with the 5 years upper estimate provided by Lee et al. (2002) for the residence time within the CIL layer.

Given this non-linear context, there are reasons to suspect that global warming, by increasing the average air temperature around which annual fluctuations occur, may induce a persistent shift in the regime of the Black Sea subsurface ventilation. Indeed, Stanev et al. (2019) used Argo float data (2005–2018) to highlight a recent constriction of the CIL layer, following a trend leading to conditions where the CIL, as a layer colder than the underlying waters, would no longer exist, The authors further indicate implications on the Black Sea thermohaline properties, as this recent weakening of the CIL layer goes along with increasing trends in surface and subsurface salinity, indicative of diapycnal mixing at the basis of the former CIL layer.

Here, we combine different data sources to analyze the variability in the Black Sea intermediate layer ventilation over the last 65 years and, in particular, investigate the existence of a statistically significant shift in the CIL formation regime, in regards to the variability observed over this period. The hypothesis of a significant regime shift is tested against the more traditional linear and periodic interpretation of the observed trends (e.g. Belokopytov, 2011), as the consequence for Black Sea ventilation and the future of the Black Sea oxygenation status in particular, are drastically different.

Indeed, Konovalov and Murray (2001) evidenced a clear relationship between oxygen conditions in the lower part of the CIL layer, and the temperature in that layer which is directly related to interannual variations in the CIL formation intensity. This relationship explain a large part of the inter-annual fluctuations in oxygen concentration in that layer, which occur at a time scale of a few years, and are superimposed on the larger scale change in oxygenation condition that is attributed to an increase in the primary production induced by the eutrophication phase of the late 1970s.

Our analysis thus aims to pursue these investigations, and in particular to focus on the annual convective ventilation as an individual component of the complex Black Sea deoxygenation dynamics (Konovalov and Murray, 2001), in the context of the recent warming trend affecting the Black Sea (Miladinova et al., 2018).

Section 2 details the datasets considered to characterize the Black Sea CIL and oxygenation dynamics and the method of regime shift analysis. In section 3, we analyse the long-term CIL dynamics through the lens of regime shift analysis. In section 4, we use outputs from a three-dimensional hydrodynamic model and recent Argo records to relate CIL formation rates to changes in the Black Sea oxygenation conditions. In section 5, we discuss those results in the frame of larger time scales, while we conclude in Sect. 6.

## 2   Material and methods

### 2.1   The cold intermediate layer cold content

While annual CIL formation rates are difficult to assess directly from observations, the status of the CIL can be quantified locally on the basis of vertical profiles of temperature and salinity. This simple indicator, based on routinely monitored variables, provides a suitable metric to combine various sources of data while summarizing an essential aspect of the thermo-haline

conditions. The CIL cold content $C$ is defined as the heat deficit within the CIL, integrated along the vertical:

$$C = -c_p \int_{CIL} \rho(z)[T(z) - T_{CIL}] \, dz, \tag{1}$$

where $z$ is depth, $\rho$, the *in situ* density, $c_p$, the heat capacity of sea water and $T_{CIL} = 8.35\,°C$, the temperature threshold which together with a density criterion $\rho > 1014.5\,\mathrm{kg\,m^{-3}}$, defines the CIL layer over which the integration is performed (Stanev et al., 2003, 2014; Capet et al., 2014). Although the use of a given temperature threshold to define the occurrence of convective mixing is subject to discussion, the existence of a fixed temperature threshold to characterize the CIL as a distinct water mass, and in particular to identify its lower boundary, is evident given the fixed value of $\sim 9°C$ that characterize the underlying deep waters (Stanev et al., 2019). The above definition has been chosen for consistency with previous literature.

$C$ is expressed in units of $\mathrm{J\,m^{-2}}$ and provides a vertically integrated diagnostic which is more informative than, for instance, the temperature at a fixed depth or the depth of a given iso-thermal surface. Although $C$ is a deficit, we inverted the sign of $C$ in comparison with previous literature (Stanev et al., 2003; Piotukh et al., 2011; Capet et al., 2014) for the convenience of working with a positive quantity. Large $C$ values thus correspond to large heat deficit in the CIL, i.e. to low temperature in a well-formed CIL layer, which is characteristic of cold years. A decrease in $C$ corresponds to a weakening of cold water formation (typically for warm years), an increase in the intermediate water temperature and/or a decrease in the vertical extent of the CIL.

$C$ has been estimated for each year over the 1955–2019 period using four data sources summarized in Table 1. These sources include *in situ* historical (ship-casts) and modern (Argo) observations, as well as empirical and mechanistic modelling (Fig. 1). Annual and spatial average values for the deep sea (depth $> 50$ m) were derived from each dataset, while considering the errors induced by uneven sampling in the context of pronounced seasonal and spatial variability. Each data source has particular assets and drawbacks, and involves specific data processing to reach estimates of annual and spatial $C$ averages as described below. All processed annual time-series are made available in netCDF format on a public repository (see 'Data availability').

***In situ ship-cast profiles***: The asset of ship-based profiles is their extended temporal coverage. The drawbacks are the difficulty to untangle spatial and temporal variability (as for any non-synoptic data source), the uneven sampling effort and the low data availability posterior to 2000. The $C^{Ships}$ time series was provided by the application of the DIVA detrending methodology on ship-cast profiles extracted from the World Ocean Database (Boyer et al., 2009) in the box 40°–47°30' N, 27°–42°E for the period 1955–2011. DIVA is a sophisticated data interpolation software (Troupin et al., 2012) based on a variational approach. The detrending methodology (Capet et al., 2014) provided inter-annual trends, here representative for the central basin, cleared from the errors induced by the combination of uneven sampling and pronounced variability along the seasonal and spatial dimensions. We redirect the reader to Capet et al. (2014) for further details on data sources, data distribution and methodology.

***Atmospheric predictors***: The statistical model considered here consists of a lagged regression model based on winter air temperature anomalies, i.e. using the form $C_i^{Atmos} = a_0 + a_1 \cdot ATW_i + a_2 \cdot ATW_{i-1} + a_3 \cdot ATW_{i-2} + a_4 \cdot ATW_{i-3}$, where $i$ is a year index, and $ATW_i$ stands for the anomaly of the preceding winter air temperature (December-March).

**Table 1.** Overview of the four datasets used to characterize the CIL inter annual variability. Details are provided for each dataset in Sect. 2.1.

| Dataset (Period) | Rationale | Assets (+) & Drawbacks (-) | References |
|---|---|---|---|
| Ship casts (1956–2011) | *In situ* profiles analyzed with the DIVA detrending methodology to disentangle spatial and temporal variability. | + Large time cover<br>+ Direct observation<br>- Uneven spatial and seasonal sampling<br>- Annual gaps | Boyer et al. (2009)<br>Capet et al. (2014) |
| Atmospheric Predictors (1956–2012) | Empirical combination of atmospheric descriptors (winter air temperature anomalies) calibrated to reproduce the above time-series. | + Full time cover<br>- Not observation<br>- Validity of statistical model not guaranteed outside its range of calibration. | Dee et al. (2011)<br>Capet et al. (2014) |
| GHER3D (1981–2017) | Three-dimensional hydrodynamic model (GHER). Unconstrained simulation (no data assimilation). 5km resolution. ERA-interim atmospheric forcing. | + Synopticity<br>+ Underlying mechanistic understanding<br>- Not observation | Stanev and Beckers (1999)<br>Vandenbulcke et al. (2010)<br>Capet et al. (2012) |
| Argo (2005–2019) | Drifting autonomous profilers. Average of synchronous profiles. | + Direct observation<br>+ Intra-annual resolution<br>- Uneven spatial sampling<br>- Recent years only | Stanev et al. (2013)<br>Akpinar et al. (2017)<br>Stanev et al. (2019) |

This model was obtained by a stepwise selection amongst potential descriptor variables (incl. summer and winter air temperature, winds and fresh water discharge), in order to reproduce the inter-annual variability of $C^{Ships}$ (Capet et al., 2014) and proposed as an alternative to the winter severity index defined by Simonov and Altman (1991). $C^{Atmos}$ is thus naturally representative of the same quantity, i.e. annually and spatially averaged $C$. The asset of this approach is the opportunity to fill the gaps between observations in past years, using atmospheric reanalysis of 2m air temperature provided by the European Centre for Medium-Range Weather Forecasts (ECMWF) for the period 1980–2013. Its drawbacks lie in its empirical nature and indirect relationship to observable sea conditions. $C^{Atmos}$ was only extracted for the years covered in Capet et al. (2014), considering that the potential non-linearity in the air temperature-$C$ relationship may be exacerbated for the low $C$ values typical of recent years.

*Three-dimensional (3D) hydrodynamic model*: The Black Sea implementation of the 3D hydrodynamic model GHER has been used in several studies (Grégoire et al., 1998; Stanev and Beckers, 1999; Vandenbulcke et al., 2010). In particular, Capet et al. (2012) presents the model setup used in this study and analyze the simulated CIL dynamics. This simulation, extending over the period 1981–2017, has been produced without any form of data assimilation, on the basis of the ERA-Interim set of atmospheric forcing provided by the ECMWF data center (Dee et al., 2011). Aggregated weekly outputs of the GHER3D model are made available on a public repository (see 'Data availability'). $C^{Model3D}$ was derived from synoptic weekly model outputs and averaged for each year and spatially over the deep basin (depth $> 50$m). The assets of this approach are the synoptic coverage in time and space and the mechanistic nature of the model, that implies a reproducible understanding of the process of CIL formation. The drawback lies in the numerical and conceptual error that might affect unconstrained model outputs.

*Argo profilers*: The assets of autonomous Argo profilers are a high temporal resolution and the continuous coverage of recent years, which offers unprecedented means to explore the CIL dynamics at fine spatial and temporal scales (Akpinar et al., 2017; Stanev et al., 2019). The drawbacks are the mingled spatial and temporal variability inherent to Argo data, the incomplete spatial coverage, and the lack of data prior to 2005. This dataset was collected and made freely available by the Coriolis project and programmes that contribute to it (http://www.coriolis.eu.org). The request criteria used were [Bounding box : 40-47N;27-42E; Period (DD/MM/YYYY) : between '01/01/2005' and '31/12/2019'; Data type(s) : ('Argo profiles','Argo trajectory'); Required Physical parameters : ('Sea temperature','Practical salinity'),Quality : Good]. In average, this set includes about 9 floats per year, with a minimum of two floats for 2005 and more than 12 floats from 2013 to 2019. $C$ values were derived from individual Argo profiles (Fig. 1). All available profiles in a given year were averaged to produce the annual Argo time series $C^{Argo}$. Although homogeneous seasonal sampling can be assumed, we note that the uneven spatial coverage of Argo profiles might induce a bias in the inferred trends. This potential bias stems from the horizontal gradient in $C$, that is structured radially from the central (lower $C$) to the peripheral (higher $C$) regions of the Black Sea (Stanev et al., 2003; Capet et al., 2014). As Argo samplings are generally more abundant in the peripheral regions, i.e. outside of the divergent cyclonic gyres, this suggests that $C^{Argo}$ might be slightly biased towards high values.

Table 2 give specific comments on the dependence relationship between the different time series presented above. Only $C_i^{Atmos}$ and $C_i^{Ships}$ can be considered as strictly dependant. $C_i^{Model3D}$ is influenced by the same datasets that were used to build $C_i^{Atmos}$ and $C_i^{Ships}$, but through drastically different processing pathways and can thus be practically considered as independent.

A composite time series was constructed as the weighted average of the 4 time-series, restricted to available sources for years during which all sources were not available:

$$C_i = \frac{\sum_j w_i^j \cdot C_i^j}{\sum_j w_i^j}, \tag{2}$$

where $i$ is an annual index, $j$ stands for a source index ($j \in \{Model3D, Atmos, Argo, Ships\}$). In order to emphasize the value of direct observations, the weights $w_i^{Argo}$ (resp. $w_i^{Ships}$) equals to 1 if $C_i^{Argo}$ (resp. $w_i^{Ships}$) is defined (i.e. the time

**Table 2.** Dependence relationships between the different time series.

| | Ship Casts | Model3D | Argo |
|---|---|---|---|
| Atmos | The statistical model providing $C_i^{Atmos}$ is built on the basis of $C_i^{Ships}$. So, even if $C_i^{Atmos}$ is more homogeneous and complete than $C_i^{Ships}$, it can not be considered as independent. | Atmospheric conditions used to build $C_i^{Atmos}$ are issued from the same data sets (ECMWF) that were used to force the 3D model. So, formally, both approaches are influenced by a common data-set, but through drastically different processing pathways. We consider no direct dependency in this case. | Strictly independent. |
| Ship Casts | - | The 3D model simulations involve no data assimilation. The model has been calibrated by testing different parameterizations of the atmospheric fluxes bulk formulations, using T and S $insitu$ data from the same set that has been used to build $C_i^{Ships}$. However, this calibration was not based on $C_i^{Ships}$ itself. Also, the selected parameterization remains fixed for the whole simulation time. So, although both times series are influenced by a common data sets, we consider there is no direct dependency. | Strictly independent. |
| Model3D | - | - | Strictly independent. |

series covers the year $i$) , and 0 otherwise, while $w_i^{Model3D}$ (resp. $w_i^{Atmos}$) equals to 0.5 if $C_i^{Model3D}$ (resp. $w_i^{Atmos}$) is defined, and 0 otherwise.

The composite time series was then used as a synoptic metric for the inter-annual variability of the convective ventilation of the Black Sea intermediate layers.

The consistency of the different CIL cold content data sources is demonstrated by the high correlations obtained between the annual time series (from 0.91 to 0.98, see detailed comparative statistics in appendix A). Despite the small number of overlapping years between certain series (eg. seven years between $C_i^{Ship}$ $C_i^{Argo}$, see Fig. A1), all correlations are significant ($p < 0.05$). The close correspondence between independent time series (see discussion above), issued respectively from strictly observational and purely mechanistic modeling approaches, provides a high confidence in their accuracy and ensures the robustness of the forthcoming analysis.

More precisely, the standard deviations estimated from the different series are similar ($\sim 100\,\mathrm{MJ\,m^{-2}}$), despite their distinct temporal coverage. The root mean square errors that characterize the disagreement between the different data sources remain

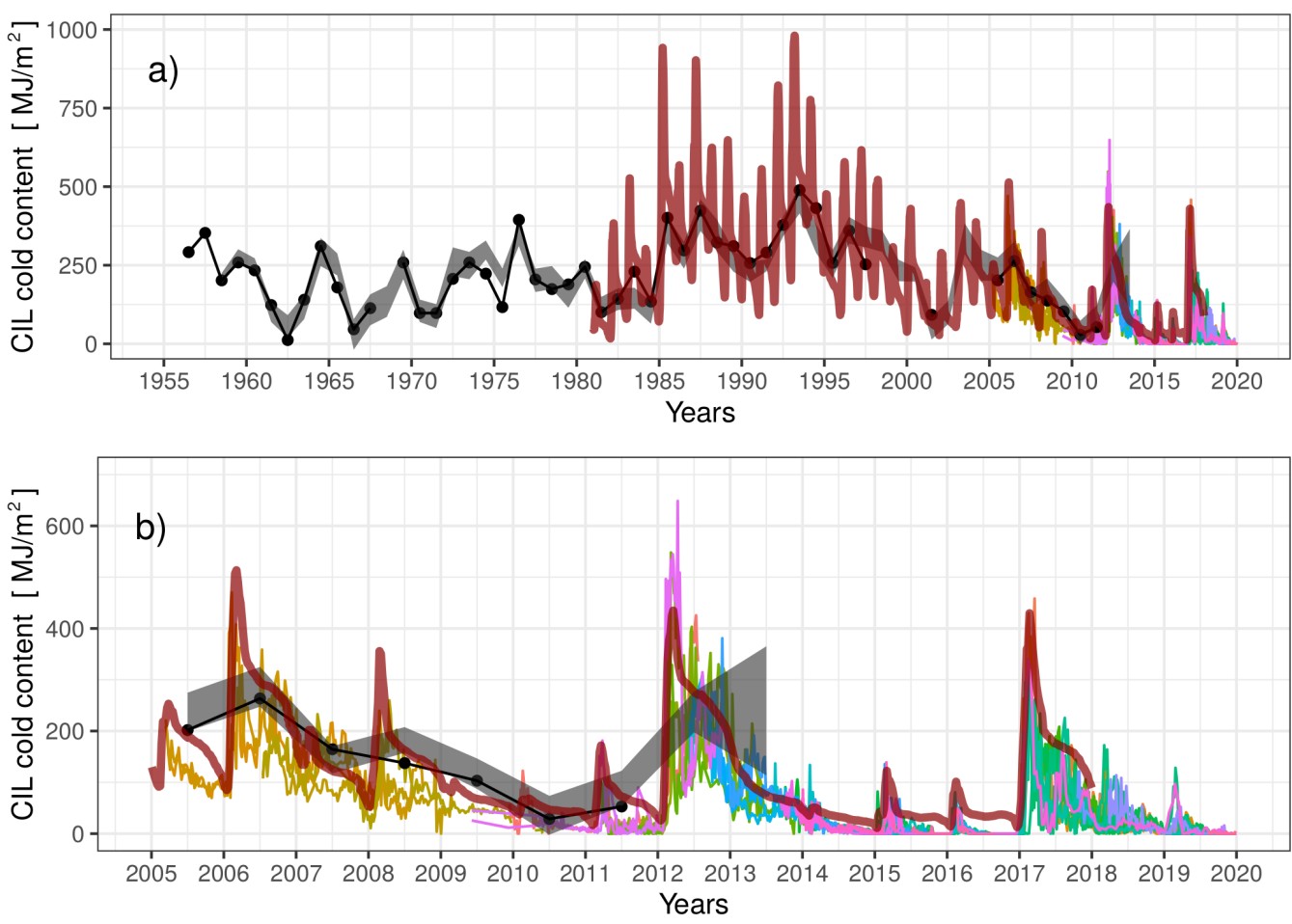

**Figure 1.** Time series of the Black Sea CIL cold content ($C$) originated from various data sources (Table 1), displayed at original temporal resolution: (black dots) inter-annual trend derived from ship casts; (gray shaded area) confidence bounds ($p<0.01$) of the statistical model based on winter air temperature anomalies; (dark red thick line) GHER3D model; (thin colored lines) Individual Argo floats. a) Complete period of analysis, b) focus on the recent years.

below this temporal standard deviation (in all but one case, see appendix A for details). This justifies to merge the different sources in a unique composite time series, enabling a robust long term analysis of the variability in the Black Sea CIL formation.

## 2.2 Regime shift analysis and descriptive model selection

The inter-annual variability of the Black Sea CIL formation is analyzed in the framework of regime shift analysis. The natural first step towards identification of a regime shift in a time series is the identification of change points (Andersen et al., 2009).

The rationale behind change point models is to identify periods over which a time series depicts statistically distinct regimes. In its simplest form, a change point model will aim to identify distinct regimes that differs in terms of their mean, i.e. during

which fluctuations take place around distinct averages. Note that other type of change point analyses can be done, which would consider other metrics (variance, autocorrelation, skewness) instead of the mean to break up the series. For the sake of simplicity, only the first moment (mean) is considered in this study.

The change point model used for this regime shift analysis has been derived and verified following the methodology described in the documentation of the R package `strucchange` (Zeileis et al., 2003). The procedure includes the following steps.

First, the presence of at least one significant change point in the time series was tested, against the null hypothesis that considers annual fluctuations around a fixed average value for the entire time series. To this aim, the `strucchange` package provides different methods based on the generalized fluctuation test framework as well as from the F test (Chow test) framework.

Second, the locations of the most likely change points in the time series were identified. Assuming that $N$ change points separates $N+1$ periods, this step thus consists in identifying the locations of the change points and the specific mean value specific to each period. This identification proceeds from an optimization procedure aiming to minimize the residual sum of squares (RSS) between the time series and the change point model (ie. constant mean value for each specific period).

Five change points models were derived for the composite time series, considering from one ($N=1$) to five ($N=5$) change points. The final step consists in selecting, among those five models, the one that 'best' describes the time series. Obviously, considering additional change points can only reduce the RSS. This is generally true for any descriptive model, and has led to the definition of the Akaike Information Criterion (AIC) for model selection. Basically, the AIC consider the RSS of each model but includes a penalty for the number of parameters (Akaike, 1974), such that if two models bear the same RSS, the one involving less parameters will be favored. Note that in our case, the parameters identified for change point models includes both the locations of change points and the specific mean for each period.

The model with the smallest AIC should be favored for interpretation. In section 3.1, the AIC is also used to compare the regime shift models to linear and periodic models of $C_i$.

More technical details and verification of underlying assumptions are given in appendix B.

## 2.3 Oxygen

BGC-Argo oxygen observations were obtained from the Coriolis data center for a period extending from 01/01/2010 to 01/01/2020. Only descending Argo profiles were considered, to minimize discrepancies associated with oxygen sensors response time (Bittig et al., 2014). To minimize the impact of spatial variability, oxygen saturation was considered using a potential density anomaly ($\sigma$) vertical scale and the year 2010 was discarded for lack of observations. While both oxygen concentration [$\mu$M] and oxygen saturation [%] were considered in our first analyses, the narrow range of thermo-haline variability in the layers of interest results in very small variations of the oxygen solubility. As a consequence, considering one or the other of these two variables led to very similar results, and we opted for oxygen saturation in the following.

Figure 2 indicates that the use of $\sigma$ vertical coordinates minimizes the range of spatial variability (see years 2014–2018, when more Argo were operating) and gives sense to the use of monthly medians as an integrated indicator of the basin-wide

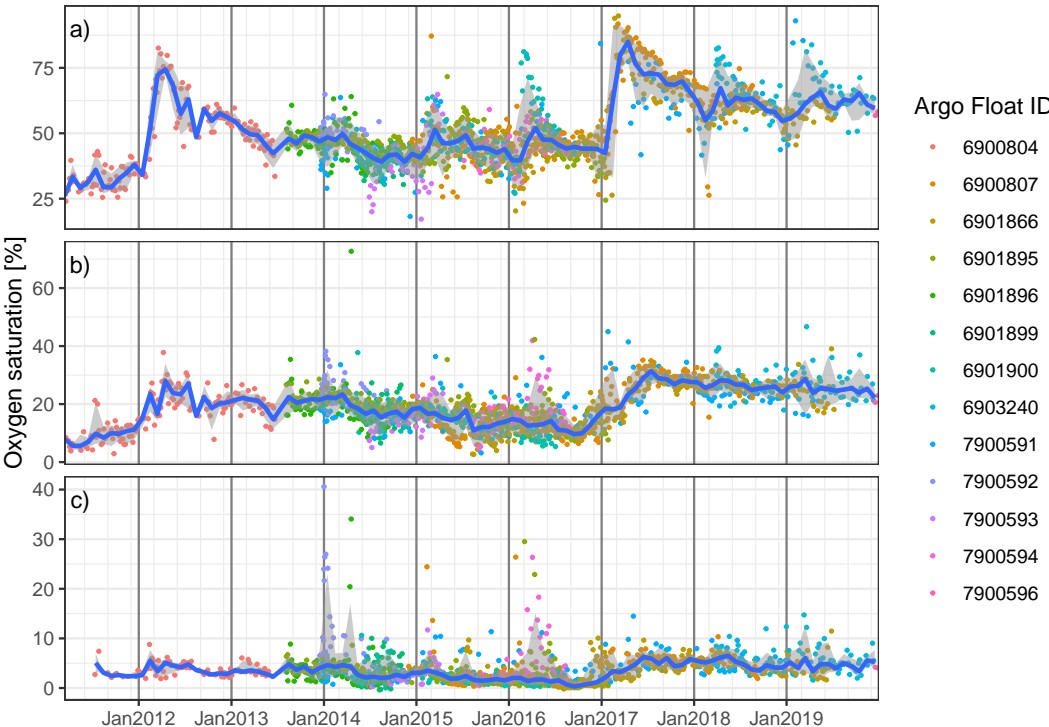

**Figure 2.** Oxygen saturation levels derived from individual BGC-Argo profiles at $\sigma$ values of a) 14.5, b) 15.0 and c) 15.5 $\mathrm{kg\,m^{-3}}$. Colored points correspond to different Argo floats. The blue line represents monthly medians while the shaded area covers monthly interquartile ranges.

oxygenation status at different layers. For deeper density layers (Fig. 2c), a larger interquartile range is induced by Argo floats profiling in the vicinity of Bosporus influenced area, as plumes of Bosporus ventilation introduce a larger horizontal variability in oxygen saturation.

## 3   The Black Sea cold intermediate layer dynamics over 1955–2019

### 3.1   Descriptive models

The composite time series $C_i$ is depicted in Fig. 3, along with individual components.

The poor statistics associated with a linear model description of $C_i$, in the form $C_i \sim l_1 \cdot i + l_2$ ($i$ stands for an annual index, adjusted $R^2 = 0.05$, AIC=794, with $l_1 = -1.59 \pm 0.78 \mathrm{MJ\,m^{-2}\,yr^{-1}}$), dismiss the perception of a linear trend extending over the entire period. Using the periodic model, $C_i \sim p_1 + p_2 \sin(\frac{2\pi}{p_3} \cdot i)$, gives a better representation of the cold content inter-annual variability (AIC=763), and provides broad characteristics of $C_i$: the mean value, $p_1 = 222 \pm 12 \mathrm{\,MJ\,m^{-2}}$, the amplitude of inter-annual variability, $p_2 = 114 \pm 16 \mathrm{\,MJ\,m^{-2}}$, and the periodicity of pseudo-oscillations, $p_3 = 43.04 \pm 0.02$ years.

A combination of linear and periodic model, with the form $C_i \sim lp_1 + lp_2 \sin(\frac{2\pi}{lp_3} \cdot i + lp_4 \cdot i)$ slightly improves the descriptive statistics (AIC=758.5). However, all of the above descriptive models overestimate $C$ in the recent years, as the composite time series $C_i$ shows a departure from its usual range of variability during the last decade. This is evidenced by ranking the 65 years of $C_i$ on the basis of their cold content. It is remarkable that, among the ten years with the least cold content, eight occurred after 2010.

Each of the change point models appears as statistically more informative, *sensu* AIC, than a linear or periodic interpretation of the time series. In particular the 4-segments model (i.e. 3 change points, AIC=752) should be favored for interpretation.

## 3.2 Regime shifts in the cold intermediate layer cold content

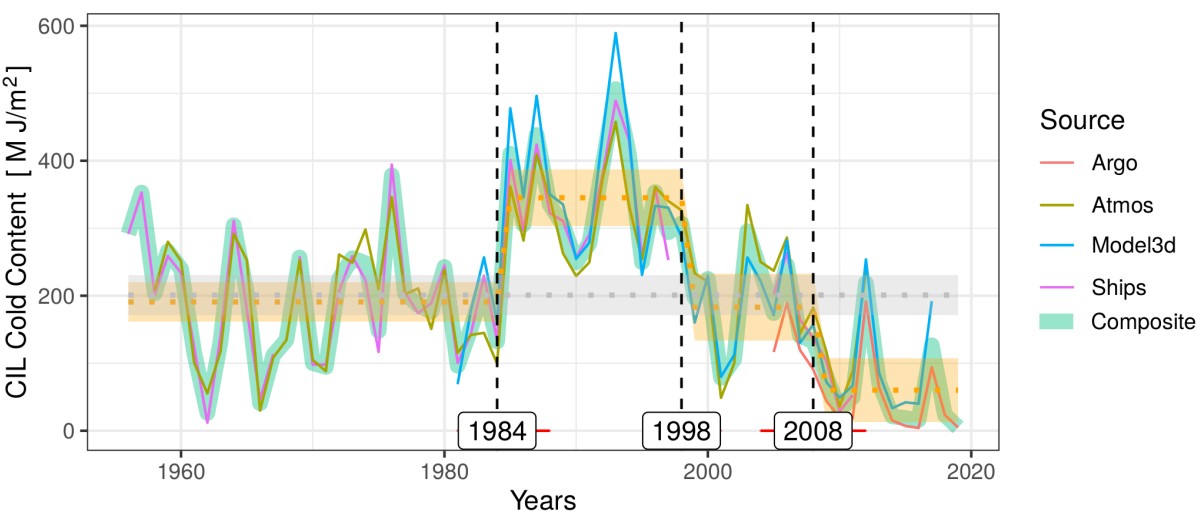

**Figure 3.** Multi-decadal variability of the Black Sea CIL cold content and distinct periods identified by the regime shift analysis. Confidence intervals on mean $C$ values are indicated by the orange shaded area for each period, and by the gray shaded area for the null hypothesis (i.e. considering no regime shifts). Confidence intervals on the time limits of each period are indicated with red ranges.

The evolution of $C_i$ over 1955-2019 is thus best described by discriminating four periods ($P_1 - P_4$, Fig. 3), objectively identified through regime shift analysis.

A "standard" regime is identified that is consistent for periods $P_1$ (1955–1984) and $P_3$ (1998–2006), which depict averages $\langle C \rangle_{P_1} = 191 \pm 15 \, \mathrm{MJ\,m^{-2}}$ and $\langle C \rangle_{P_3} = 183 \pm 29 \, \mathrm{MJ\,m^{-2}}$, respectively. This routine regime is also consistent with the average $C$ obtained without considering any change points, $\langle C \rangle = 201 \pm 15 \, \mathrm{MJ\,m^{-2}}$ (Fig. 3).

Departing from this routine, a cold period ($P_2$) is visible from 1984 to 1998, during which $C$ fluctuates around a larger average value, $\langle C \rangle_{P_2} = 345 \pm 26 \, \mathrm{MJ\,m^{-2}}$. This cold period has been described in numerous studies (e.g. Ivanov et al., 2000; Oguz et al., 2006), and is attributed to strong and persistent anomalies in atmospheric teleconnection patterns (East Asia/West Russia and North Atlantic oscillations; Kazmin and Zatsepin (2007); Capet et al. (2012)). Noteworthy, similar cold periods were identified earlier in the 20th century (late 1920s–early 1930s and early 1950s; Ivanov et al. (2000)).

From 2008 to 2019, a warmer period ($P_4$) is identified during which $C$ varies around a lower average $\langle C \rangle_{P_4} = 60 \pm 28 \ \mathrm{MJ\,m^{-2}}$. The regime shift analysis thus evidences that a general weakening of the cold water formation and associated ventilation prevails in the Black Sea since about ten years. Warm years and low cold content were also observed during the years 1961 and 1963, but those were not identified as part of a statistically distinct "warm" regime and should be considered as strong fluctuations within $P_1$. The regime shift analysis thus indicates that the current restricted ventilation conditions have no precedent in modern history.

## 4    Cold intermediate layer formation as an oxygenation process

The intra-annual resolution provided by the 3D model and Argo time series (Fig. 1) suggests that the partial CIL renewal, that was taking place systematically each year before 2008, has now become occasional. Here we focus on period P4, better detailed in our datasets, to characterize the CIL formation as a basin wide ventilation process and its relationship with changes in oxygen saturation at different $\sigma$ levels.

Basin wide CIL formation and destruction rates were computed from the synoptic 3D model outputs, as differences between weekly $C$ values (Fig. 4). The seasonal sequence depicts CIL formation peaks from late December to March, typically reaching $C$ formation rates of 5, 10 and 1 $\mathrm{MJ\,m^{-2}\,d^{-1}}$ for the period P1/3, P2 and P4, respectively (Fig. 4a,b,c). The CIL cold content is then eroded at different rates before, during and at the end the thermocline season, with a damped erosion rate through the thermocline season between 0 and about -1 $\mathrm{MJ\,m^{-2}\,d^{-1}}$. CIL formation processes have been described extensively in the past (e.g. Akpinar et al., 2017; Miladinova et al., 2018), in more detail than allowed by the integrated perspective adopted here. This integrated point of view, however, serves to point out the striking quasi-absence of CIL formation peaks for the years 2001, 2007, 2009, 2010, 2013 and 2014 (Fig. 4d). In fact, during the period of Argo oxygen sampling, only 2012 and 2017 depict important CIL formation events, while minor CIL formation events are shown for 2015 and 2016.

Oxygen saturation in this period varies in concordance with CIL formation until $\sigma$ layers of about 16.0 $\mathrm{kg\,m^{-3}}$ (Fig. 5). Large increases can be observed from December to March in the years 2012, 2015, 2016 and 2017 when CIL formation is significant, which denotes the impact of convective ventilation. The narrow interquartile ranges depicted on Fig. 5, denote the efficiency of the isopycnal diffusion of oxygen : the amount of oxygen imported with the newly formed CIL waters is distributed horizontally and contributes to increase the average oxygen saturation of a given $\sigma$ layer.

While major CIL formation events in 2012 and 2017 induced a significant increase in oxygen saturation through the whole oxygenated water column, the minor events in 2015 and 2016 seems to have a limited penetration depth. For instance, oxygen saturation at $14.6 \ \mathrm{kg\,m^{-3}}$ only stagnates during 2015 and 2016 as compared to 2014, while oxygen saturation at $15.1 \ \mathrm{kg\,m^{-3}}$ keeps decreasing during these two years, indicating that the amount of oxygen brought to this layer during minor ventilation events is not sufficient to counter-balance the biogeochemical oxygen consumption terms (i.e. respiration and oxidation of reduced substances diffusing upwards).

Following our attempt to summarize large datasets and to characterize a basin-scale annual oxygenation rate, we computed for each layer an annual oxygenation index as the difference between the median oxygen saturation in November between

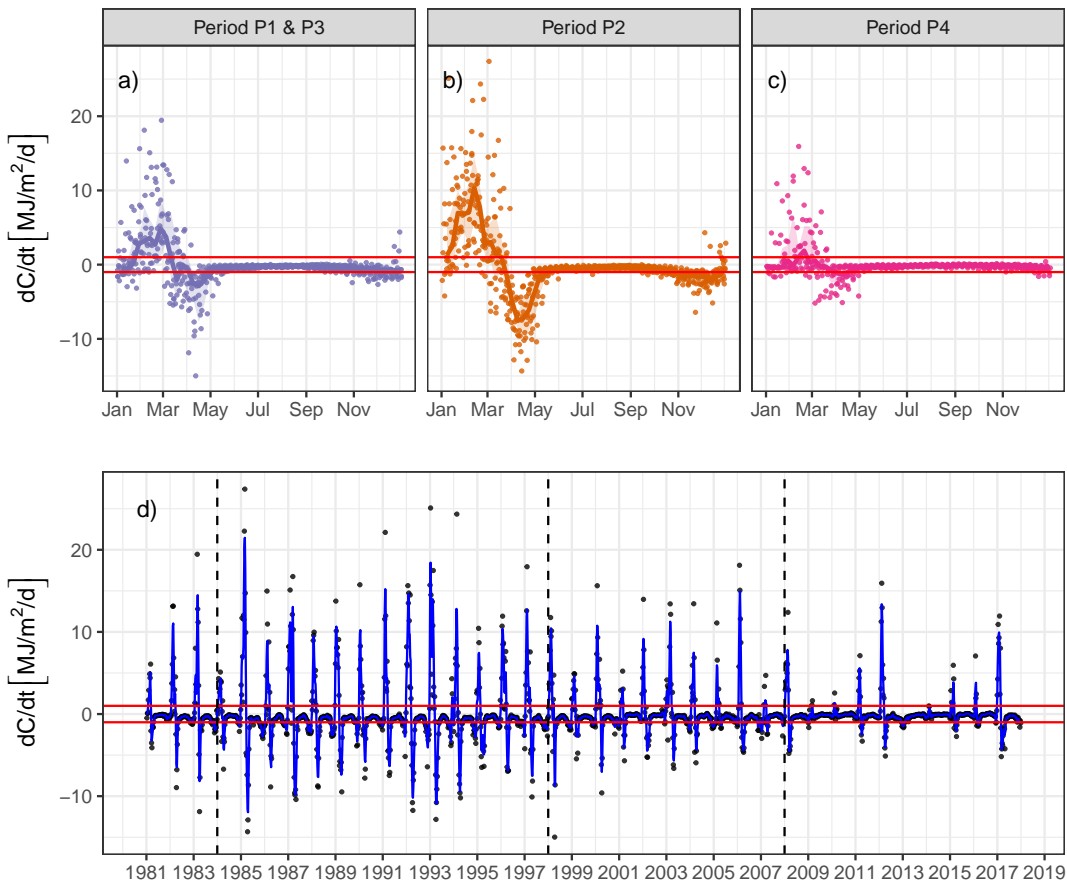

**Figure 4.** Weekly averaged basin wide CIL cold content formation and destruction rates ($dC/dt$), obtained as differences between the weekly integrated CIL cold content provided by the GHER3D model: a), b), c) on a seasonal frame with weekly medians (line) and interquartile range (shaded area), merging years from the periods P1 and P3 (considered together), P2 and P4; d) on an inter-annual scale, with 3-weeks running average (blue line). Vertical dotted lines separates the four periods evidenced by the regime shift model. The red lines delineate the thresholds of $\pm\,1\ \mathrm{MJ\,m^{-2}\,d^{-1}}$, corresponding to the lower bound of CIL erosion rate during summers.

successive years. The rationale behind this approach is that CIL formation typically extends from December to March (Fig. 4a,b,c).

In order to obtain a general indication on the (pycnal) penetration depth of the convective ventilation associated with CIL formation, we assessed the Pearson correlation coefficient between this annual oxygenation index, and a first order assessment of annual CIL formation, obtained as the annual difference in the $C$ composite time series. The correlation between oxygena-
tion and CIL formation is high near the surface and decreases continuously as deeper pycnal levels are considered. Those correlations remain significant ($p < 0.1$) until $\sigma = 15.4\ \mathrm{kg\,m^{-3}}$ (Fig. 6).

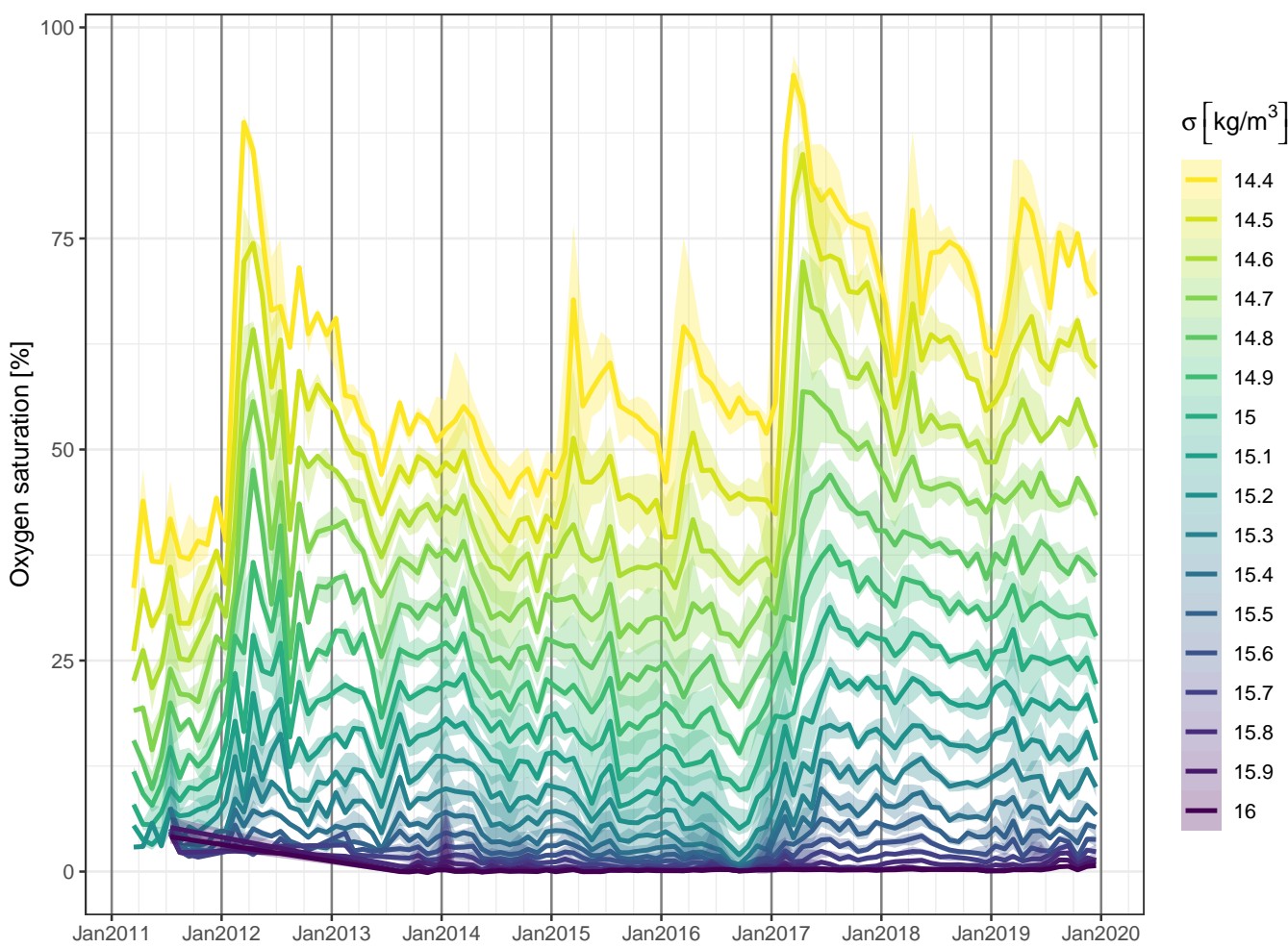

**Figure 5.** Monthly medians of oxygen saturation at different $\sigma$ layers. Shaded areas indicate the monthly interquartile range (Fig. 2).

## 5 Discussion

The regime shift paradigm describes an abrupt and significant change in the observable outcome of a non linear-system, as could result from a threshold in this system response to an external forcing. A periodic model, on the other hand, supposes
either a linear response to periodically varying external forcing, or an oscillation resulting from internal dynamics. It is our hypothesis, supported by the quantitative consideration presented above, that the regime shift model should be favored for interpreting the recent evolution of the Black Sea CIL dynamics. The prerequisite for the regime shift analysis was first to issue an unified, synoptic metric to characterize inter-annual variations in the CIL content. The consistency between the different data sources demonstrates the robustness of this metric. To our knowledge, no multiple source comparison have been achieved
previously over such an extended period. Note that some dependencies exist among certain sources, as has been discussed explicitly in Sect. 2.

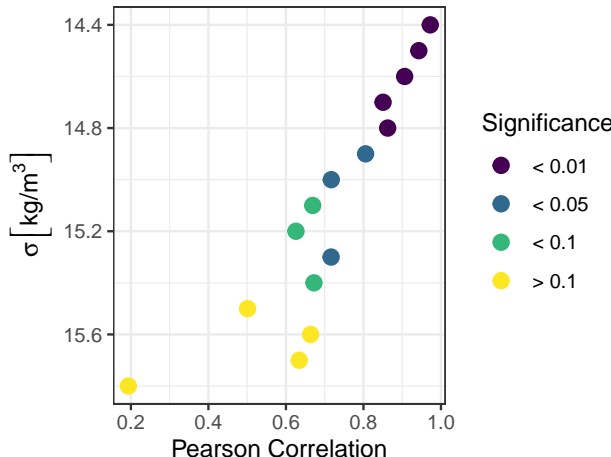

**Figure 6.** Pearson Correlation Coefficient between basin wide annual oxygenation and CIL formation estimates, for different $\sigma$ layers. Size of the points relates to the order of magnitude of associated p-values, while colors classify those correlations among classes of significance.

Although, we acknowledge that the statistical advantage (AIC) of the regime shift description is subtle, we consider that it deserves further consideration as this difference in interpretation is fundamental in what regards the expected consequences on the Black Sea oxygenation status and in particular the threat on Black Sea marine populations, whose ecological adaptation (and rate of exploitation) have been built upon a ventilation regime and consequent biogeochemical balance, that may no longer prevail.

Indeed, it appears that the intermittency of significant CIL formation events characterizes the new ventilation regime: the ventilation of the Black Sea intermediate layers does not occur each year anymore but is occasionally absent for one or two consecutive years. Moreveover, major CIL formation events, which bear the potential of a deeper oxygenation, appears as significantly less frequent.

The extent to which the current regime differs from the previous ventilation regimes is clearly illustrated on a T-S diagram (Fig. 7) : not only are in-situ measurements from the P4 period found in a range of the T-S diagram that was almost never recorded before (temperature above 8.35°C, and $\sigma$ within [14.5–15]$\mathrm{kg\,m^{-3}}$; Fig. 7a,b,c,d), but two-dimensional density estimates (obtained with R function MASS:kde2) also indicate that such occurrences are now the rule rather than the exception.

As indicated by Stanev et al. (2019), this trend may lead to the disappearance of a characteristic layer of the Black Sea, that constituted a major component of its thermo-haline structure and constrained the exchanges between surface, subsurface and intermediate layers. In particular, the authors highlight surface and subsurface salinity trends that indicate recent occurrences of diapycnal mixing at the lower base of the intermediate layer, where waters are characterized by a strong reduction potential due to the presence of reduced iron and manganese species, ammonium and finally hydrogen sulfide (Pakhomova et al., 2014).

On a decadal time scale, the average oxygen signature of a given isopycnal layer within the CIL depends on the frequency of CIL formation events of sufficient intensity (Sect. 4), which is in line with the ventilation dynamics depicted by Ivanov et al. (2000) for the upper pycnocline. Although inter-annual fluctuations in CIL formation rate still occur, the regime shift analysis

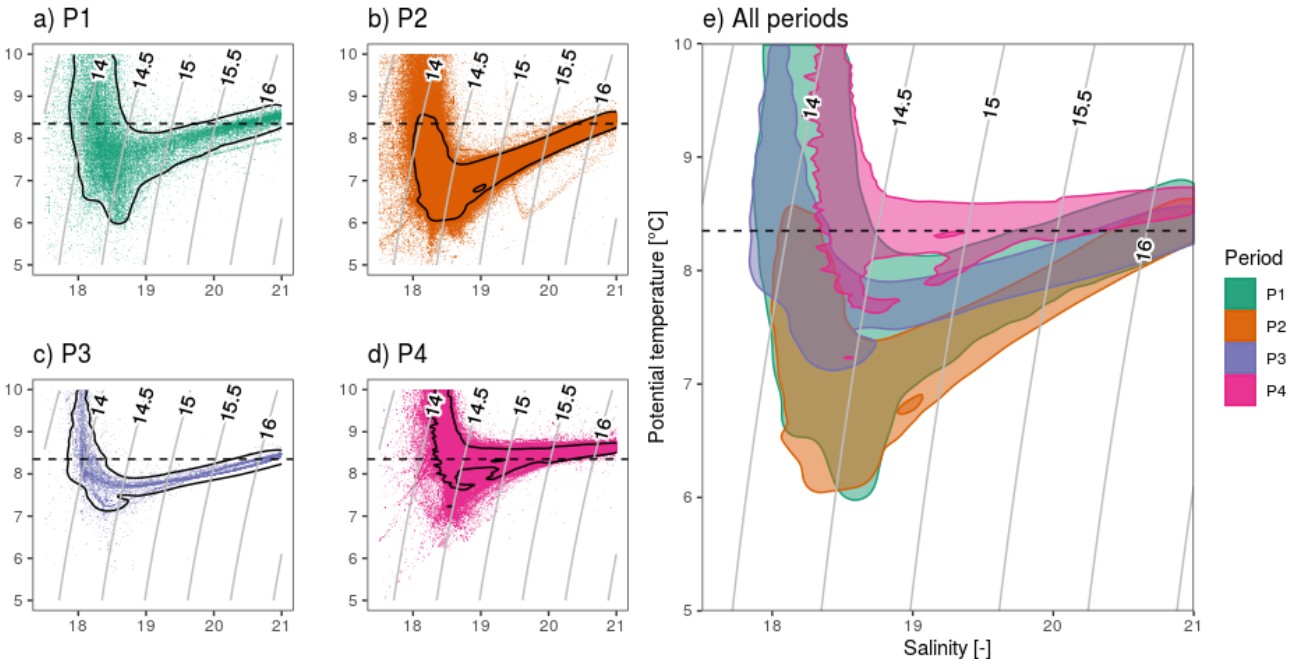

**Figure 7.** Potential temperature versus salinity (T-S diagram) for bottle, CTD, and Argo in-situ data available from the World Ocean Database for the period 1955–2020 (Boyer et al., 2013). Data from periods P1, P2, P3 and P4 are shown on panels a), b), c) and d), respectively. Black contours delineates 75% of the observations for each period, and are repeated with colors in e) for comparison between periods. The black dotted line locates the $T_{CIL} =8.35°C$criterion used to identify CIL waters.

specifically describes a reduction in the frequency of significant ventilation events, and therefore a potential decrease in the oxygen saturation signature in the lower part of the CIL.

A deeper consequence of this reduction relates to the fact that the lower CIL layer also acts as the upper member of the two end-member mixing line that characterizes the Black mid-pycnocline (Murray et al., 1991; Ivanov et al., 2000), i.e. between density of about 14.6 to 16 $\mathrm{kg\,m^{-3}}$, following entrainment of CIL waters by the Bosphorus inflow and subsequent lateral ventilation (Buesseler et al., 1991), the lower end-member being composed by Bosporus waters. Considering the characteristic residence time for the upper (about 5 yr; Lee et al. (2002)) and intermediate pycnocline (9-15 yr; Ivanov et al. (2000)), it is

appropriate to consider such temporal average to characterize the oxygen signature of the CIL member composing the mixture of pycnocline waters.

The display of historical oxygen saturation data (1955–2020) on a T-S diagram (Fig 8a), indeed evidences in general a deeper oxygenation during high CIL formation regimes (see lower part of the diagram, characteristic to P2), than in regimes during which significant CIL formation events are rare, or the extreme cases were no CIL waters are visible (see upper part

of the diagram, characteristic to P4). This indicates that the analysis linking oxygenation and CIL formation for the recent period (Sect. 4), can be extended to larger time scales by considering changes in the frequency of significant CIL formation

events. Thus, the depth until which the reduction in CIL formation may impact on biogeochemical balance of the Black Sea (by affecting oxygenation level) will depend on the period over which the actual ventilation regime will remain.

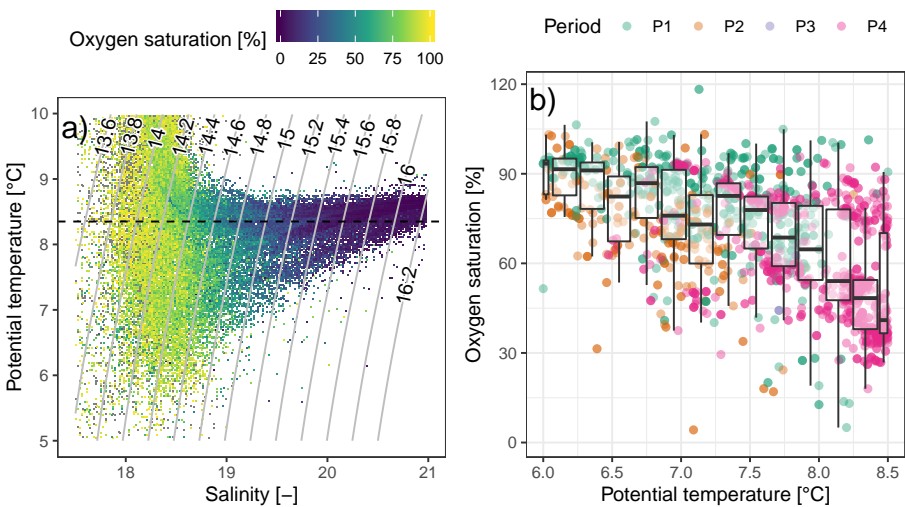

**Figure 8.** a) Historical oxygen records displayed on the TS diagram. Data collected from the World Ocean Data base for the period 1955–2020 (Boyer et al., 2013). Isopycnal $\sigma$ layers are indicated by the curved grey lines. b) Highlight of the oxygen saturation conditions along the $14.55 - 14.65 \, \text{kg m}^{-3}$ $\sigma$ range of the T-S diagram, i.e. at the core of the CIL layer.

Beyond the changes in convective ventilation highlighted above, it thus appears as a lead priority to assess the biogeochem-
335 ical consequences of this new thermo-haline dynamics of the Black Sea. In particular, the influence of CIL formation on the biogeochemical components of the oxygen budget should be addressed in more details, asking for instance how the presence or absence of CIL formation influences on planktonic growth, trophic interactions, and organic matter respiration rates. We voluntarily adopted here a wide integrative point of view, so as to highlight the large scale relevance of the depicted regime shift on Black Sea oxygenation. However, we still consider that a detailed assessment of all components of the oxygen budget,
i.e. considering ventilating processes and biogeochemical production/consumption terms, is required in order to infer the future evolution of the Black Sea oxygenation status (e.g. Grégoire and Lacroix, 2001).

Although, there are clear indications of a long-term warming trend in the Black Sea (Belkin, 2009), it remains a delicate task to strictly dissociate the contributions of global warming from that of regional atmospheric oscillations (Kazmin and Zatsepin, 2007; Capet et al., 2012). One such assessment in the neighboring Mediterranean Sea (Macias et al., 2013) concluded that
global warming trend and regional oscillation contributed to the recent regional sea surface temperature trend (1985–2009), for 42% and 58%, respectively. While corresponding assessment will have to be routinely reevaluated for the Black Sea as time series expand, it may conservatively be considered that global warming had a significant contribution to warming winters in the Black Sea. This contribution is expected to increase in the next decades (Kirtman et al., 2014).

## 6 Conclusions

We have analyzed the variability of the Black Sea CIL formation over the last 65 years and investigated the existence of regime shifts in this dynamics. To this aim, we have produced a composite time series of the CIL cold content ($C$), that is considered as a proxy for the intensity of the convective ventilation resulting from the formation of dense oxygenated waters. This composite time series is built from four different data products issued from observations and modelling, so as to optimize its temporal extent in regards to preceding studies (e.g. Oguz et al., 2006). The consistency between those products, and in particular the 355 close correspondence between observational and mechanistic predictive time series supports the reliability of the composite series and its adequacy to describe the evolution of the Black Sea subsurface convective ventilation during the last 65 years.

The composite time series was analyzed to detect different regimes, corresponding to periods characterized by significantly distinct averages. Over the last 65 years, we identified 3 main regimes: 1) a routine regime prevailing during 1955–1984 and 1996–2008 that is consistent with the full-period average $C$, 2) a cold regime (high $C$, 1984–1996) which has been previously 360 documented (see references in Sect. 3) and 3) a warm regime (low $C$, 2008–2019) which is characterized by the intermittency of the annual partial CIL renewal. Statistical considerations indicate that the abrupt shift can not adequately be described by a combination of long term linear and periodic trends.

The synoptic CIL formation rates provided by the 3D hydrodynamical model, and the detailed description of oxygenation conditions provided by BGC-Argo floats, allowed us to detail the role of CIL formation in oxygenating, through convective 365 ventilation, the upper part of the Black Sea intermediate layers (i.e. between $\sigma$ of 14.4 and 15.4 $\mathrm{kg\,m^{-3}}$). Given that cold winter air temperature is the leading driver of CIL formation (Oguz and Besiktepe, 1999; Ivanov et al., 2000; Capet et al., 2014), given that CIL formation constitutes a predominant ventilation mechanism for the Black Sea intermediate layer, and assuming that oxygen conditions constitutes an environmental structuring factor affecting the ecosystem organization, its vigor and its resilience, this shift in the Black Sea ventilation regime may be associated with global warming and is expected to affect 370 its biogeochemical balance and to threaten marine populations adapted to previously prevailing ventilation regimes.

To understand how global warming impacts on the marine deoxygenation dynamics is a worldwide concern. The relatively fast and clear response that stems from the specific Black Sea geomorphology makes it a natural laboratory to study this dependency and related phenomena. Here, we showed that non-linear dynamics and feedbacks in ventilation mechanisms resulted in a significant shift of the average ventilation regime, in response to rising air temperature. Since the temporal extent 375 of low oxygen conditions is critical for ecosystems, we stress the importance to assess the potential for similar ventilation regime shifts in other oxygen deficient basins.

*Data availability.* The data used are listed in Table 1 Argo data were were collected and made freely available by the Coriolis project (http://www.coriolis.eu.org) and programs that contribute to it. Era-Interim atmospheric conditions were obtained from ECMWF interface (http://apps.ecmwf.int/datasets/). Aggregated weekly outputs of the GHER 3D model, as well as processed annual time series from the 380 different sources are publicly available on a ZENODO repository : https://doi.org/10.5281/zenodo.3691960.

## Appendix A: Comparison of the $C$ time series issued from different data sources

The $C$ time series are denoted $C_i^m$ for source $m$ ($i$ is the year index). Each pair of time series $(C_i^m, C_i^n)$ are compared over the years $i \in I^{m,n}$ for which $C_i^m$ and $C_i^n$ are both defined. The following statistics are given for each pair of data sources in Fig. A1 :

- $N^{m,n}$ the number of elements in $I^{m,n}$

- Pearson correlation coefficient :

$$\frac{\sum\limits_{i \in I^{m,n}} (C_i^m - \overline{C^m})(C_i^n - \overline{C^n})}{\sqrt{\sum\limits_{i \in I^{m,n}} (C_i^m - \overline{C^m})^2} \sqrt{\sum\limits_{i \in I^{m,n}} (C_i^n - \overline{C^n})^2}} \tag{A1}$$

- The RMS difference between time series :

$$\sqrt{\frac{\sum\limits_{i \in I^{m,n}} (C_i^n - C_i^m)^2}{N^{m,n}}} \tag{A2}$$

- The average bias :

$$\frac{\sum\limits_{i \in I^{m,n}} (C_i^n - C_i^m)}{N^{m,n}} \tag{A3}$$

- The percentage bias :

$$\frac{\sum\limits_{i \in I^{m,n}} (C_i^n - C_i^m)}{\sum\limits_{i \in I^{m,n}} \frac{\left(C_i^n + C_i^m\right)}{2}} \cdot 100 \tag{A4}$$

For a better appreciation of variation scales, the temporal standard deviation is also shown for each data source.

The last value of the atmospheric predictor time series ($C_{2013}^{Atmos}$; Fig. 1) was not considered in the composite time series, as it was based on the two, rather than three, predictor values available at the time of publication (hence the larger associated uncertainty). It is remarkable, however, that the published prognostic values for 2012 and 2013, match with independent Argo estimates (Capet et al., 2014).

## Appendix B: Regime shift analysis

The basic change point problem that is considered in this study can be expressed as follows: to identify the change point $i = k$ in a sequence $x_i$ of independent random variable with constant variance, such that the expectation of $x_i$ is $\mu$ for $i < k$ and $\mu + \Delta\mu_k$ otherwise. Obviously, this problem can be generalized to several change points. The procedure for change point detection is

stepwise and has been achieved following the methodology described in the documentation of the R package `strucchange` (Zeileis et al., 2003).

First, the existence of at least one significant change point had to be tested. `strucchange` provides seven statistical tests to compute the $p$-value at which the null hypothesis of no change points can be rejected. The presence of change points can be tested on the basis of F-Statistic tests or generalized fluctuation tests (Zeileis et al., 2003, and references therein). Table A1 provides the significance level at which the null hypothesis can be rejected for each test implemented in the `strucchange` R package. Among the seven tests considered to assess the presence of at least one significant change point in $C_i$, six reject the

null hypothesis with a significance level $p < 0.05$.

     Second, the locations of the $N$ most likely change points were identified. In this study, we considered from one to five change points. The change point locations can be estimated by finding the index values $k_n = [k_1, .., k_N]$ that maximize a likelihood ratio, defined as the ratio of the residual sum of squares for the alternative hypothesis (i.e. change points at $k_n$ with $\Delta\mu_{k_n} \neq 0$) to that of the null hypothesis (i.e. no change point, $\Delta\mu_{k_n} = 0$).

Formally, the methodology to identify and date structural change is designed for normal random variables, two conditions which can not be guaranteed for environmental time series such as considered here. We detail here why 1) the departure of $C$ distribution from a Gaussian distribution, 2) the *autocorrelation* in the composite time series $C_i$, and 3) the biases between source-specific components of $C_i$, do not affect the conclusions drawn above.

## B1   Normality

Skewness in the distribution of $C$ values and its departure from normality is visible at low $C$ values (not shown), as expected for physical reasons: $C$ is a vertically integrated property, naturally bounded by zero. However, the Shapiro-Wilk test that measures the correlation between the quantiles of $C_i$ and those of a normal distribution, indicates no significant departure from normality: $W = 0.975$, $p = 0.23$. For completeness, a Box-Cox transformation ($\lambda = 0.7$) of the original data has been tested which slightly enhances the Shapiro-Wilk test ($W = 0.98$, $p = 0.39$), but brings no sensible alteration in the conclusions of the

structural change analysis.

## B2   Autocorrelation

Similarly, $C_i$ can not be considered as a random variable. In particular, we introduced in Sect. 1 the CIL preconditioning and partial renewal mechanisms, both physical reasons for which autocorrelation may be expected in $C_i$. Indeed, correlations between the original and $k$-lagged time series are, at first glance, significant up to $k = 5$ (Table A2; the confidence interval above

which autocorrelation can be considered to be significant is given by $1.96/\sqrt{N} = 0.25$). However, it should be considered that the regime shift evidenced in this study may itself induce apparent autocorrelation statistics. To evidence that this is indeed the case encountered here, the correlation between the original and lagged time series of *the residuals* stemming from the 4–segments change point model are indicated in Table A2. The fact that no significant autocorrelation persists when change points are considered indicates that the non-randomness of $C_i$ does not jeopardize conclusions drawn from the application of

the structural change methodology.

**Table A1.** Tests for the presence of significant structural changes in $C_i$. $p$-value indicates the probability that the null hypothesis (i.e. there are no significant change points) should be maintained.

| Approach | Test | $p$-value |
|---|---|---|
| F-Statistics | supF test | 1.7e-04 |
| | aveF test | 5.3e-03 |
| | expF test | 1.7e-08 |
| Fluctuations | OLS-based CUSUM test | 1.0e-02 |
| | Recursive CUSUM test | 3.6e-01 |
| | OLS-based MOSUM test | 1.0e-02 |
| | Recursive MOSUM test | 1.0e-02 |

**Table A2.** Correlations between (second column) time-lagged replicates of the original $C_i$ and (third column) time-lagged replicates of the residuals of the 4–segment model.

| Lag | Original | Residuals |
|---|---|---|
| 0 | 1.0 | 1.0 |
| 1 | 0.57 | 0.13 |
| 2 | 0.38 | -0.22 |
| 3 | 0.32 | -0.11 |
| 4 | 0.33 | 0.03 |
| 5 | 0.26 | -0.05 |

## B3 Biases between components of the composite time series

Given that biases exist between different data sources, it might be argued that the composite time series is skewed by the uneven temporal coverage of the different sources. For instance, if a strongly biased source would solely cover a given period, the composite series would be biased over that period. To ensure that this issue does not affect the presented conclusions, $C_i^{unbiased}$ was constructed as $C_i$, but removing from each component $C_i^m$ the bias identified with the longest $C_i^{Ships}$ time series (which series is used for reference does not impact on structural change conclusions). When $C_i^{unbiased}$ is considered instead of $C_i$, similar results are obtained in terms of change point models significance and positions of the change points.

*Author contributions.* AC processed the data, set the regime shift methodology, achieved the analyses, issued the visualizations and wrote the initial version of the manuscript. All authors contributed to define the general methodology, to discuss the results and to revise the final mansucript. MG supervised the research.

*Competing interests.* No competing interests are identified.

*Disclaimer.* TEXT

*Acknowledgements.* This study was funded by the FRS-FNRS FNRS convention BENTHOX (PDR T.1009.15) and directly benefited from the resources made available within the PERSEUS project, funded by the European Commission, grant agreement 287600. LV is funded by the EC Copernicus Marine Environment Service program (contract: BS-MFC). Computational resources have been provided by the supercomputing facilities of the Consortium des Equipements de Calcul Intensif en Federation Wallonie Bruxelles (CECI) funded by the Fond de la Recherche Scientifique de Belgique (FRS-FNRS).

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

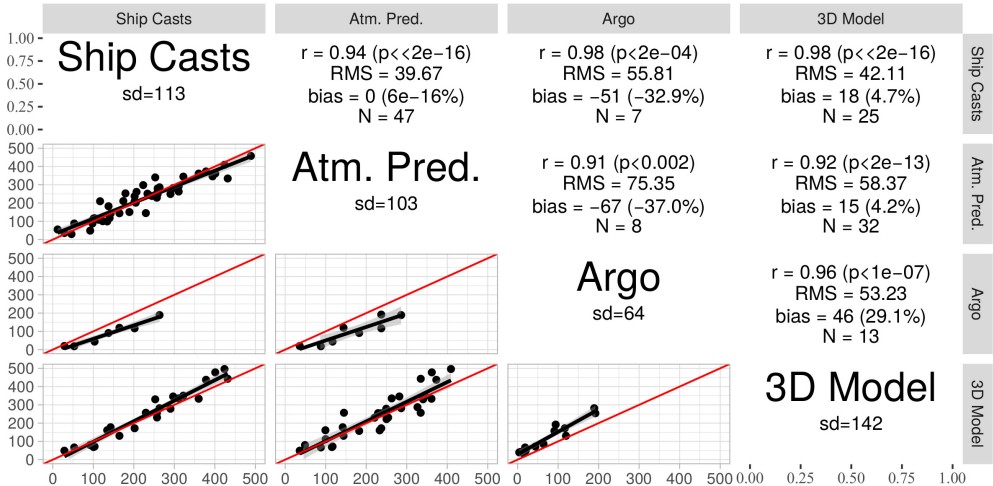

**Figure A1.** Statistics of comparison between the different data sources.