# Peer review of "Climate change induced a new intermittent regime of convective ventilation that threatens the Black Sea oxygenation status"

_Biogeosciences, 2020_

## Referee Comment (RC1) · Fabian Große (Referee) · 12 May 2020

**Review of Capet et al.: *"Climate change induced a new intermittent regime of convective ventilation that threatens the Black Sea oxygenation status"***

The manuscript by Capet and co-authors presents a statistical analysis of cold intermediate layer (CIL) content and formation and its impact on oxygen levels in the Black Sea from the 1960s to 2019 combining different data sources (incl. observations and models). The two key findings are: (1) temporal changes in CIL water can only be described well by a model when taking into account regime shifts; (2) CIL water formation has entered a new warm regime (i.e. low formation) around 2008, which affects oxygen levels (through reduced ventilation) and is likely to also affect the biogeochemistry in the Black Sea.

The manuscript is concise, well written and generally easy to follow. The presented statistical analysis seem sensible and appropriate to address the posed research questions. However, I need to state clearly that I am not very experienced in complex statistical analysis and it would be good if someone more knowledgeable in that field could evaluate this aspect of the study.

Most of my comments are only minor, primarily ask for some clarification, and should be easy to address by the authors. I have only a few somewhat bigger points:

(1) The study does not isolate the climate change impact (as suggested by the title) on the change in Black Sea ventilation. In my understanding, this cannot be achieved by the applied method, hence, I suggest adapting the title as a simple fix.

(2) It is not clear from the descriptions of the datasets whether the models are completely independent from the observational data, i.e. whether the same observations have been used for model calibration. If that's the case, this needs to be discussed.

(3) In the discussion, the authors should highlight more explicitly what the new insights of this study are compared to previous work on the topic (I understand it is the regime shift).

(4) The appendix seems to be incomplete as there are references to non-existent figures.

Overall, I recommend publication after moderate revisions. My detailed comments follow below; line numbers are given as, e.g. 'L123'.

**Specific comments**

Title: I suggest changing it to "A new intermittent regime of convective ventilation threatens the Black Sea oxygenation status" as the study cannot really isolate the climate change impact from that of regional atmospheric oscillations.

L58: This should also be mentioned when describing the statistical model using atmospheric forcing in the Methods

L71/72: I am not fully convinced that your analysis actually separates the convective ventilation from the biogeochemical processes (BGC). The observed oxygen from Argo is affected by both physics and BGC. In order to isolate the physical (ventilation) component, would it make more sense to analyze oxygen saturation concentration (and AOU for changes in BGC)?

L73-76: Maybe this "section list" is not needed? If you keep it, the last sentence seems to be incomplete.

L85: Assuming that density increases with depth, this density criterion only defines the upper limit of the CIL. How is the lower limit defined?

L111: Include the overall number of Argo profiles used and also include the minimum and maximum number for individual years (and state corresponding years) in order to give an idea about sampling error. Given the time series of CIL content rate of change (Fig. 6), I am also wondering whether it would make more sense to only use summer/fall profiles (as rate of change is very small during that period; analogous for the other datasets)?

Table 1: I suggest listing the different datasets in the same order as in the text. I understand from the text description that the statistical model, based on atmospheric predictors, uses more than one (winter air temperature anomalies) predictors, right? If so, please state all of them either in the text, in the table or both. For convenience, the time periods covered by the different datasets could be added to the table.

L150-151: What is the smallest number of overlapping years between datasets? Is it appropriate to calculate correlations in the cases of least overlap? Regarding the atmospheric predictor model and the 3D hydrodynamical model, have any of the observational datasets (Argo and ship casts) been used to calibrate/develop either model? If so, those datasets would not be fully independent (during overlapping periods), which would affect the composite $C$ time series (especially, its uncertainty as it may in-/decrease if they were in-/dependent) and should be addressed in the discussion.

L156: I think Appendix A is incomplete as I couldn't find the information referred to here.

Section 2.3: It would be nice to show a time series similar to Fig. 1a also for oxygen in order to put the more recent development in CIL oxygen into historical context. I understand that this is not possible for the two models but perhaps it could be done for the ship casts? This would be particularly useful with respect to climate change.

L174/175: I am wondering whether the recently submitted work by Gordon et al. (https://doi.org/10.5194/bg-2020-119), who suggest a correction of BGC-Argo oxygen observations based on the sensor response time, could help to make use of both descending and ascending profiles?

L203: It's probably my personal taste but I don't really like the term "routine" regime. Maybe "standard", "normal" or simply "average"?

L213-215: Would it make sense to show annual (or winter) average surface air temperature as small panel in Fig. 5 to better demonstrate this link?

L218: More out of interest: would it be possible to also get intra-annual resolution from the statistical model? E.g. by not using winter time averages of the descriptor variables but monthly averages or so?

L225: I don't understand what you mean with "before, during and at the end the thermocline setting". Mainly because I am not sure what you mean with "thermocline setting"; is it the thermocline formation or the period during which a near-surface thermocline exists?

L231: Oxygen is from Argo. Would it make sense to do the intra-annual comparison of CIL formation rates also based on Argo then (or a combined $C$ using 3D model and Argo)?

Discussion: As stated in the very beginning, it would be good if differences to earlier studies/novelty would be highlighted more explicitly. In my understanding, the main differences in terms of methodology are the longer period and the basin-scale integrated approach, which is needed to detect the regime shifts. This needs to be made more clear and it should also be discussed what the advantages (for the purpose of this study) and possible limitations are.

L286/287: If one of the main conclusions focuses on the combined linear-periodic model than this combined model should be included in Figs. 3 and 4.

Appendix: There seem to be two figures missing: Fig. A1 (L307/308) and Fig. D1 (L379). The appendix contains a few aspects that would fit into the discussion in most journals (incl. BG), e.g. the discussion of the suitability of the statistical methods. I assume this is owed to the previous submission to GRL. The authors may consider moving parts of it into the discussion (or even into the Methods, e.g. the statement on oxygen data on L378-380). However, it also works the way it is. Maybe the editor can have a say on that.

**Technical corrections**
L2-3: "from the mid-1970s to the early 1990s"; specify "recent years", e.g. "post-2005"? comma after "Here"
L6: maybe "years without renewal of intermediate water"?
L7: "density levels"
L15: "While the reduction"
L20: "at the surface"
L24" "Strait"
L27: no comma after "(halocline)"; remove "the" after "prevents"
L33: "winter time"
L40: "upper ~100"
L41: "forcing"; "e.g." (check throughout the manuscript, also for "i.e.", the first "." is often missing)
L42: give time scale for alterations, e.g. "(order of days)"
L53: "Miladinova et al. (2018)"
L57: "feedback"
L62: (2005-2018)
L63: "trend leading to conditions"
L65: "with increasing trends"
L67: comma before "in particular"
L68: remove "extended"
L71: "an individual component"

L74: "regime shift analysis"

L78: use full term for CIL in section heading

L81: "provides"

L87: "water mass"

L89: "for consistency with existing literature"

L93: "i.e."

L99: "dataset"

L100: I'd suggest "errors" instead of "artifacts" since one usually uses the term "sampling error"; space before "Each data"

L102: "netCDF"

L107: "This data was"

L110: "Good]."

L115: Could you provide the total numbers of profiles in the central and peripheral basin regions?

L123: again I'd suggest "errors"

Table 1: "guaranteed" instead of "granted" (under drawbacks for atmospheric predictors); "Three-dimensional" instead of "3D" in the rationale for GHER3D

L124: please cite the "reference study" here

L126: "consists of"

L129: remove "defined earlier"

L134: please cite the "reference study" here

L136: "Three-dimensional (3D) hydrodynamic model"

L136-144: Please include the time period for which the model was run.

L147: "metric for the"

L151: specify those series with little overlap

L154: here and throughout the manuscript: no space between "M J" in the unit; just "MJ"

L159: "inter-annual"

L161: "i.e."

L164: "models' ability"

L168: "i.e."

L169: "which" instead of "that"

L171: "shift"

Fig. 1: add legend; larger panel and axes labels; "MJ" instead of "M J" in y axes units; no "-" between quantity and units on y axes (this applies to all figures); caption: "Table" instead of "Tab."; "time series" instead of "trend"; why does the statistical model show a range and what value (e.g. mean) of that model was used for the analysis?

L181: "Argo floats were operating"

L182/183: "Argo floats profiling"

Fig. 2: larger axes and panel labels; caption: "Argo floats"

L185: use full term for CIL

L187: "in Fig."

L188: include use latex command "\cdot" instead of "." In "$I1.i$"; state what "$i$" is

L193: "overestimate"

Fig. 3: in the legend, you use "Model3d", while you use "C^3dModel" on L140 (I suggest to use one term consistently); axes labels and legend could be larger; suggest adding the linear and periodic functions to the legend (or the terms and put the functions into the caption)

L196: "shift"; "i.e."

L198: ";" before "not shown"

L199: "i.e."

Fig. 4: caption: "shift"; "point"

L200: suggest using full term for "CIL"

Fig. 5: here you use "3D model" in the legend (different to Fig. 3), be consistent; caption: "blue shaded area" looks more purple; "gray shaded area" very hard to see, perhaps make slightly less transparent; no comma after "i.e."

L207: no comma after "e.g."

L209: ";" after "oscillation"

L210: "20$^{th}$"; "1950s; Ivanov et al., 2000))."

L213: "prevails in the Black Sea since about ten years."; "low cold content"

L217: suggest using full term for "CIL"

L219: "before 2008" instead of "in precedent regimes"; specify "the latest period"

L224: "period P2, P1/3 and P4, respectively."

L226: for better readability: "about -1 MJ"

L227: "in more detail"

L228: "quasi-absence" and add reference to figure panel

L229: "during" instead of "among"; "depict"

L230: "while lower"; "simulated" or "shown" (this figure is not based on observations)

L231: unit missing for "16.0"

L232: "increases"; "in the years 2012 and 2015—2017 when CIL formation is significant"

L234: no space between "~ 14"

L241: "decreases continuously"

L242: "remain"; space before unit

L247: "regime shift model" instead of "first"

Fig. 6: the yellow dots in P4 panel are difficult to see, suggest using different color; add panel labels (a, b, c, d) for mores specific in-text referencing; add "regime name" to panel titles; caption: "time series"; "in Fig. 5"

L250: "built"

L251: "prevail" instead of "be considered as routine"? I think it's more important to emphasize that earlier assumptions might not be valid anymore

L254: "trend"

Fig. 7: the shaded areas are hard to see, maybe make them a little less transparent? Caption: "areas"; "." at end of caption

Fig. 8: significance is expressed by p-value, so why additional log10 of it? "Pearson Correlation Coefficient"; Labels and legend should be a bit larger

L265: "i.e."

L269: "Sea"

L271: put ", respectively" at end of sentence

L284: "which has been"

L286: space before "Statistical"; "indicate"

L289: "i.e."; parenthesis not closed after sigma values

L297: "feedbacks"

L301/302: all links should be included in that section, not as footnotes

L306: "denoted"

L310-317: equations should be numbered; in the RMS equation, "N^m,n" should be under the square root; in the relative bias equation, use "\cdot" instead of "."

L319: ";" before "Fig. 1"; no "." after "Fig. 1"

L321: add reference for "published prognostic values"

L331: "p-value" with cursive "p"

L334: "six reject"

L340: "i.e."; use "Akaike Information Criterion (AIC)" as you refer to this appendix before introducing AIC in the main text

L341: this should be "Appendix C" (afterwards the numbering of appendix sections B1/B2/C1 seems off)

L343: "guaranteed"

L356: "Table C1; the"

L360: "Table C1"

Table C1, caption: "second column" and "third column"

L365: "coverage"; comma after "For instance"

L367: remove one "identified"

L370: "in-situ data"

L371: use "in situ" or "in-situ" consistently throughout the manuscript (preferably the former in cursive letters), here you use both

L380: no space before the "."; don't use "in this study" instead of "in the following"

L381: The heading for the "Author contributions" section is missing

---

## Referee Comment (RC2) · Anonymous Referee #2 · 29 May 2020

The title sounds interesting and attractive. However, for me, the result sounds as follows: There were two cold winters in the period of 2012-2019, and in each of them was not only the amount of cold waters larger, but the concentrations of oxygen were also higher. There is nothing new, except that this has been recorded by BioArgo. Neither, the scientific content of the manuscript supports this title.

We read in the Abstract: "Oxygen records from the last decade indicate a clear relationship between cold water formation events and oxygenation status at different isopycnal levels, suggesting a leading role of convective ventilation in the oxygen budget of the upper intermediate layers." This just repeats what has been known for many years. This

finding is just a confirmation of previous knowledge (see several papers of Konovalov et al). Possible shifts of temperature-oxygen relationships in different periods have also been broadly addressed in these studies. Authors say nothing about that. This brings me to the major criticism. Authors have to make clear what the new knowledge is, which can be gained from their study compared to older ones.

One big problem is that there is a lack of balance in the manuscript. Much attention has been given to the long-term variability. On this subject, I cannot find anything new. The intimate link between analyses of 65-year and 7-year time series is not clear.

The good part of the research is the analysis of data after 2012. However, its relation to the previous periods is completely unclear, and neither is it well articulated in the manuscript. In this part, authors should clearly describe which floats they use, and how these floats capture the temporal and spatial variability. Important to know is whether what is observed is a clear signal or just noise. The statistics of data, and how representative for the Black Sea state they are, need deeper consideration. Fig. 1 shows perhaps that data used in the analyses are not homogeneous. If the data is not homogenous, the subsequent interpretations of long-term changes are not credible.

The statement "... suggests that the CIL renewal, that was taking place systematically each year in precedent regimes, has now become occasional." contradicts what is known from earlier studies. They have to at least refer to Lee et al. (2002. Anthropogenic chlorofluorocarbons ...) who claim that the residence time is ∼5yr at 80-120m. I would recommend that they explain what the problem in this earlier estimate is, if any.

The issue of regime shift is not convincing. The question is: can oxygen data over 7 years only identify regime shift? What was the previous oxygen regime? What I see in the oxygen data are just two ventilation events, not a shift. Authors ignore referring to important works about regime shift (e.g., the review article of Oguz in Front. Mar. Sci., 25 April 2017). They have to study the references in this review.

Based on my comments above, I am very sorry that I cannot recommend publishing

the manuscript in this form.

---

## Referee Comment (RC3) · James W. Murray (Referee) · 5 Jun 2020

Review: Biogeosciences BG-2020-76 Authors: Capet, Vandenbulcke and Gregoire Title: Climate change induced a new intermittent regime of convective ventilation that threatens the Black Sea oxygenation status This is a very interesting paper. Basically the authors: 1. Assemble a time series data set of temperature data from the Black Sea from about 1955 to the present combining real observations and model results. 2. They integrate cold temperature anomalies in the Cold Intermediate Layer (CIL), relative to a reference temperature of 8.35°C which defines the upper and lower boundary of the CIL, which they call "the cold content". 3. They analyze the (extensive)

variability in this "cold content", using several different approaches (linear and sine functions), but settle on a technique they call the "regime shift" paradigm or hypothesis. 4. Using regime shift they identify four periods that characterize different amounts of "cold content" 5. They argue that these periods reflect variable degrees of ventilation of the CIL 6. The last 11 years have been a period with unusually low ventilation and they argue that this is due to ocean warming. I have a few specific comments, 1. The paper is a little hard to read because of the advanced data analysis techniques used and (though generally well written in English) some awkward word choices. I'm not sure who can fix that. 2. L1 Abstract – a ∼100 m ventilated surface layer is referred to but does that mean 0-100m No, it means the Cold Intermediate Layer which is more like 50 to 100m I think you should be more specific. 3. L20 The early literature (e.g., Tolmazin 1985, Progress in Oceanography 15, 217) argued that as it appeared that the sea surface in the central gyres never got cold enough for replenishment of the CIL by winter convective, that the main source of water to the CIL on an annual basis was from the NW shelf where the key density surface was cooled. I agree that we have much more data now and starting with Gregg and Yakushev (2005), who observed a ventilation event (with real data), we now know that ventilation can occur from the central gyre regions. But the NW shelf hypothesis as been totally left out of all papers since the 1990s, such as those by Akpinar, Ivanov, Oguz and others. I looked back at those papers and they don't even mention the Tolmazin argument, much less argue why it would not play a role. So as far as I can tell, the NW shelf is a possible source of ventilated water for the CIL. If the Tolmazin hypothesis has been disproved, I missed that. I think Capet et al should take that into account. It may not show up in their model, depending on how it is parameterized. 4. L29 Murray et al (1989) discovered the suboxic zone. Stanev et al (2018) is a nice paper but used model results to argue for what causes its origin. 5. L104 Why not describe the data sources in the same order as presented in Table 1? 6. L126 How were the Atmospheric Predictors converted into C and CIL temperature variability in the water column?? I don't think anything is said. 7. L136 Does that 3D hydrodynamic model include source

water from the NW shelf? 8. Figure 5 with the intervals obtained from regime shift analysis is compelling. But I think back to the geological axiom "If I hadn't believed it, I wouldn't have seen it!". Visually (without the vertical lines) it looks like there is more variability than shows up as the std. deviations. But this is not my area of expertise so I hope other reviewers can evaluate that. 9. L173 Section 2.3 The absolute values of O2 are uncertain unless the sensors are carefully calibrated. I suspect they were not. The relative changes are probably OK. 10. L181 Why 2018 and not 2020? 11. L289 I think Konovalov and Murray (2001) showed this in a figure. 12. I really prefer the real data over the model results, which just reflect what equations and parameters were put into the model. For this reason I would really like to see a T-S plot (with real data) maybe averaged (with std. dev.) for the 3 regimes, blown up to highlight the CIL region. The intervals in Figure 5 should show up clearly. 13. I think it would be useful to explain another reason why the CIL is important. In my view (see Murray et al., 1991) it is because it plays an important role in the formation of all deep-water in the Black Sea. To a first approximation, all deep water in the Black Sea forms by linear 2-end member mixing between the Bosporus outflow and the CIL. This must be because salinity increases all the way to the bottom and the only source of salinity is the Bosporus. See Figure 12 of Murray et al (1991). This mixing occurs on the SW shelf (Latif et al., 1991). Any curvature in the T-S plots is due to temporal variability in the signature of the CIL endmember. This mixing can be seen in the T and S sections from the Bosporus to the shelf break from Gregg et al (1999). See Murray et al (1991) for more discussion. If the CIL is warmer, and less dense, how will that impact deep water ventilation? I think the deepest layers will be ventilated less frequently.

Please also note the supplement to this comment:
https://www.biogeosciences-discuss.net/bg-2020-76/bg-2020-76-RC3-supplement.pdf

---

## Referee Comment (RC4) · Michael Dowd (Referee) · 6 Jun 2020

Review:

"Climate change induced a new intermittent regime of convective ventilation that threatens the Black Sea oxygenation status"

I was asked by the editors to review this manuscript with an eye towards an assessment of the statistical methods that were used.

Oceanographically, the notion of different convective regimes and their link to Black Sea oxygen dynamics is established using (mainly) observational data. A causal relation,

of course, cannot be inferred from this, but the physical linkages are plausible ones and great effort has gone into making sensible metrics for the cold intermediate layer, with a solid understandng of the limitations of each of the data types used. I don't know enough about the literature pertaining to convective processes, nor regional Black Sea oceanography, to comment on significance of this contribution and its novelty within oceanography. For the remainder of this this review, I will focus on the statistics and data analysis aspects.

The paper quantifies the ventilation of the Black Sea intermediate layers through a heat (cold) content metric, i.e. the CIL defined in eqn (1). Estimates for the values for the CIL over time are computed from 4 sources: the first two are observational data (ARGO floats and CTD casts); the next is output from a statistical model that relates air temperature to CIL; and the last is numerical ocean model output. The different CIL estimates have very good correspondence to one another, and a composite CIL time series is produced for further analysis. The central feature of the paper is a change-point analysis of the CIL series was undertaken to isolate different convective regimes. This is then followed by correlation analysis to relate CIL changes to oxygen dynamics.

From an visual examination of the data, I doubt that the 4 regimes found exist as distinctive equilibrium states (if that is how one defines a regime), but are rather part of a continuously varying process with cycles and trends like we see in all climate data. These CIL data do clearly show underlying long period cycles and a decreasing trend since 1980. As for the existence of regimes, looking at the graphs without the regimes (Fig 3), and with the regimes superimposed (Fig 5) – one's eye is drawn to the regimes in Fig 5, but these are subtle at best. A statistical changepoint analysis will always pick up regimes, and in my experience the AIC criteria used here tends to choose overly complex model (here, implying more changepoints and regimes). But the above is a subjective assessment. Quantitatively, the analysis undertaken is clearly spelled out, its assumptions addressed, its application done properly, with the result that 4 regimes are statistically identified. These regimes here are changes in the level (mean) of the

CIL over time; there appears to be no changes in the higher order statistical moments. Note that there are many different ways to do changepoint analyses, and each need not be only based on the mean level, but could use other metrics (variance, auto-correlation, skewenss) to break up the series. I feel there is value in putting forth this regime analysis and the results for discussion to the wider community, and hence support publication of the paper. I would, however, downplay the ambitious claims of the title. This is an analysis that brings out the recent decline in the oxygenation state, its relation the ventilation and the CIL, and suggests the possibility of CIL regimes. But these regimes are subtle at best and not easily separable from the general downward trend in CIL. The last regime might be different from those in the past (less cold), but further work would be needed to verify/validate this as a new regime.

My main criticism is that I found it difficult to follow the methodology and how it was applied. The only reason that I was able to do so was since I have used most all these techniques before, and so could 'read between the lines' as to what was being done. The latter comment is important since it means that adequate and understandable descriptions of the statistical approaches used need to be included in the main body of the manuscript (not just buried the appendix details). Important point are glossed over (the composite time series, the atmospheric predictors model for CIL, and in particular the regime shift model). To compound this, basic statistics are discussed at length (like overfitting, AIC, and model selection, and fitting the curves of Fig 3), but in a way that is not clearly linked the problem. Below I provide specific suggestions. Overall, however, it should be straightforward to insert the necessary methodological detail into the text, and streamline the appendices, so that the results would be understandable and reproducible by an educated reader. I view these as relatively minor changes (since everything is there, if you read carefully with an adequate background).

SPECIFIC:

Title: a bit over-reaching. There is no explicit link made to climate. But the paper clearly demonstrates the CIL decline and de-oxygenation since 1980 whether or not these are

really new regimes.

Describe the "Atmospheric Predictors" model in the text (line 126). It looks like a lagged regression model. Important to be specific. Why not a basic equation?

Describe the composite CIL time series (line 145) in more detail. It is a central quantity for the paper.

Describe the regime shift model in text. It is a change-point model. These are easy to explain, if tricky to code. Be clear in the text the R-package you used, what it is based on, and that there are lots of different kinds of changepoint analysis and algorithms. Because of this lack of description, it is unclear why the authors then talk about model selection in the next paragraph (because different numbers of changepoints imply distinct statistical models). I think this paragraph on model selection and fit/complexity metrics to be too general and elementary – best put in the appendix.

Regime shift analysis and number of changepoints. I am unclear on whether the R-package you used here computes the number of breakpoints as well as their time locations. The reason I bring this up is since you have used an AIC criterion to choose the number of breakpoints – is this your addition to the analysis, or is it part of the R-package? State clearly as this is the central step that determines the 4 regimes (and hence underpins your conclusions).

Figure 3 and supporting text is not needed. Fitting a linear trend and a sinusoid to the CIL doesn't add anything to the paper (and the data is repeated in Fig 5 for the regime analysis).

Figure 4. Similarly, all the model selection stuff like AIC versus changepoints is over-explained and too generic. It is enough to say in the text that AIC identified 3 breakpoints as optimal, then refer to appendix.

Figure 6. Define dC/dt (difference between median oxygen concentrations between sucessive years). Is this the annual oxygenation index you refer to in line 239?

Figure 8. No need for both significance (colours) and p-values (size) since they give the same information. Significance values are thresholds for p-values.

Rationalize the material in appendices. It is rather a strange grab bag of material, and perhaps is a consequence of a previous review process?

Appendix A: Put a basic understandable description of the regime analysis the main body of text (as noted). Details here. Note also that there are many changepoint analysis of various levels of statistical sophistication. You've picked one of many.

Appendix B. I think this is a discussion of how the statistical assumptions of this regime analysis – which is based on linear regression - could be violated, and their possible consequences. Not sure you need most of this, and the important parts should be folded into Appendix A. Overall, I'll agree that within each regime, using average annual values, the normal iid assumption of OLS regression is OK (and hence the inference underlying the changepoint detection).

Appendix D. This appendix could be omitted and a brief statement in the text made as to how and why oxygen concentration vs saturation were used.

---

## Author Comment (AC1) · 17 Aug 2020

We thank the editors and reviewers for their insightful comments and precious advice.

Since several comments coincide between reviewers we decided to address their comments in a common document, which is provided to all reviewers and can be found as supplementary material to this comment.

We'd like to express here our general gratitude to the reviewers and appreciation for the pertinence of their remarks.

[Figure]

Please also note the supplement to this comment:
https://bg.copernicus.org/preprints/bg-2020-76/bg-2020-76-AC1-supplement.pdf

[Figure]

**Supplement:**

**BG-2020-76**

**Link to manuscript**

- https://www.biogeosciences-discuss.net/bg-2020-76/

**Answers to Reviews**

We thank the editors and reviewer team for their insightful comments and precious advice.
Since several comments coincide between reviewers we decided to address their comments in this common document, which is addressed to all reviewers. The reviewer's comments are thus reproduced here in their integrality (in blue), with specific answers including when relevant references to comments from other reviewers. Some of the most critical remarks result from imprecise wording, leading to misunderstandings. Below, we propose some rephrasing of the respective paragraphs, so that these misunderstandings can be avoided. Finally, we'd like to express here our general gratitude to the reviewers and appreciation for the pertinence of their remarks.

In summary, to address the main reviewer's comments we propose to:
- Extend the description of past Black Sea oxygenation research in the introduction, for a better positioning of the novelties of our research ->see  *2.3, 2.4, 2.9, 2.11, 3.12, 3.13.*
- Enforce the links between the recent (Argo) period and the longer-term period (1955-present)  *3.12, 3.13.*
- Revise the description of the statistical approach, and streamline appendixes -> see *4.6, 4.7*
- Adopt oxygen saturation concentration -> see *1.7*.
- Clarify technical points, regarding
    - AIC derivation -> *4.8*
    - Statistical model from atmospheric conditions -> *1.13*
    - Independence between distinct time-series -> *1.2*
- Alleviate miscomprehension due to lack of clarity -> *2.8, 2.9, 2.10, 4.11*
- Restate clearly the new insights of this study, accounting for the additions mentioned above -> *1.3*
- Change the title *-> 1.1*

**Reviewer #1**

*The manuscript by Capet and co-authors presents a statistical analysis of cold intermediate layer (CIL) content and formation and its impact on oxygen levels in the Black Sea from the 1960s to 2019 combining different data sources (incl. observations and models). The two key findings are: (1) temporal changes in CIL water can only be described well by a model when taking into account regime shifts; (2) CIL water formation has entered a new warm regime (i.e. low formation) around 2008, which affects oxygen levels (through reduced ventilation) and is likely to also affect the biogeochemistry in the Black Sea. The manuscript is concise, well written and generally easy to follow. The presented statistical analysis seems sensible and appropriate to address the posed research questions.*

 ***However, I need to state clearly that I am not very experienced in complex statistical analysis and it would be good if someone more knowledgeable in that field could evaluate this aspect of the study.***
**->** We believe Reviewer 4 fulfills this role and we thus refer to our answer to Reviewer 4 for statistical considerations.

*Most of my comments are only minor, primarily ask for some clarification, and should be easy to address by the authors. I have only a few somewhat bigger points:*

**1.1** *The study does not isolate the climate change impact (as suggested by the title) on the change in Black Sea ventilation. In my understanding, this cannot be achieved by the applied method, hence, I suggest adapting the title as a simple fix.*
-> This point has been underlined by several reviewers. The title will thus be modified, following the suggestion given by Reviewer1, ie. "A new intermittent regime of convective ventilation threatens the Black Sea oxygenation status".

**1.2** *It is not clear from the descriptions of the datasets whether the models are completely independent from the observational data, i.e. whether the same observations have been used for model calibration. If that's the case, this needs to be discussed.*
-> The table below describes the direct and indirect dependencies that may result from the construction of the 4 times series.

| | *Atm* | *Model3D* | *Argo* |
|---|---|---|---|

| | | | |
|---|---|---|---|
| *Diva* | Statistical model using Atmospheric predictors is built on the basis of the Diva time-series. So, even if Atm is more homogeneous and complete than the Diva series, those can not be considered as strictly independent. | The 3D model simulations involve no data assimilation. The model has been calibrated ("hand" screening of Atm. fluxes bulk formulation params.) using T and S in situ data from the same set that has been used to proceed with the long term Diva Analysis, but not using directly the C time series resulting from the DIVA analysis.
So it's not strictly independent, in the sense that both times series are both influenced by the same initial data sets. But we consider there is no direct dependence. | Strictly independent. |
| *Atm* | | Atmospheric conditions from which are extracted the predictors for the statistical model (average winter air temperature) are issued from the same data sets (ECMWF) that are used to force the model. Again, both approaches are influenced by a common data-set, but through drastically different processing pathways. We consider no direct dependence in this case. | Strictly independent. |
| *Model3D* | | | Strictly independent. |

We propose :
- to extend Section 2.1 by adding at L145, a succinct description of direct and indirect dependence between times series, based on the information reported in the table below.
- To refer explicitly to this discussion in the sentence L151-153.

*1.3 In the discussion, the authors should highlight more explicitly what the new insights of this study are compared to previous work on the topic (I understand it is the regime shift).*
First, following comments from Rev2, we will further develop the introduction with a paragraph dedicated to past research on Black Sea oxygenation and ventilation dynamics, in order to better position our findings with respect to those of previous studies.

Considering the points addressed in this review, we can reformulate our main findings as follows :
- We issue an unified, synoptic metric to characterize inter-annual variations in CIL formation. The consistency between independent sources demonstrates the accuracy of this metric. Note that some dependencies exist among certain sources, this is discussed explicitly.
- Analysis of the time-series reveals a current restricted ventilation regime that is unprecedented over the last 64 years. Interannual variations appear to be better described by considering abrupt change models than smooth dynamics.

- In particular the actual regime includes years during which **no** new CIL waters are formed, which appears to be a new phenomenon. We highlighted this in conference proceedings (EGU GA, May 2018, Kiel conference on marine deoxygenation, October 2018) . Stanev et al. 2019 later discussed this issue on the basis of Argo observations. Here, we propose a long term multi-source description of the CIL dynamics, to insist on the anomaly of this feature in regards to an extended period.
- The general relationship between CIL formation and oxygenation of intermediate waters has been previously described in the literature. Here, we exploit the dense Argo data sets (7 years) to document in more detail and at a finer scale the response of oxygen conditions to CIL formation at different pycnal levels. In particular, we illustrate the fast decrease in oxygen conditions for years where no CIL water was formed.

The whole study stresses the current prevalence of a restricted ventilation regime, its impact on the Black Sea oxygenation status, and the urgency of research efforts dedicated to alleviate or at least foresee potential environmental and economic consequences.

*1.4 The appendix seems to be incomplete as there are references to non-existent figures.*
-> Figures A1 and D1 have been erroneously displaced at the very end of the pdf, which is probably why the reviewer missed those (also, as the figures are quite heavy. I experienced some lags while scrolling down to that part of the pdf on the Copernicus website). It will be ensured that all figures are distributed coherently in the revised manuscript. However, following the advice of reviewer 4, Appendix D will be removed, and replaced by corresponding addition in the text. The specific issue of Appendix D is further addressed in *1.7*

*Specific comments*
*1.5 Title: I suggest changing it to "A new intermittent regime of convective ventilation threatens the Black Sea oxygenation status" as the study cannot really isolate the climate change impact from that of regional atmospheric oscillations.*
-> Agreed. see *1.1.*

*1.6 L58: This should also be mentioned when describing the statistical model using atmospheric forcing in the Methods*
-> Several comments pointed towards a more explicit description of this statistical model. We centralized our answer in *1.13.*

*1.7 L71/72: I am not fully convinced that your analysis actually separates the convective ventilation from the biogeochemical processes (BGC). The observed oxygen from Argo is affected by both physics and BGC. In order to isolate the physical (ventilation) component, would it make more sense to analyze oxygen saturation concentration (and AOU for changes in BGC)?*
-> For the sake of simplicity, and according to the suggestion, all oxygen considerations will be re-issued on the basis of saturation concentration (ie. in situ concentration as a percentage of oxygen solubility at in situ conditions). We attempted to indicate with Appendix D and Fig D1 that, for the layer below sigma 14.3 kg/m³ there is a quasi-strict linear relationship between molar concentration and saturation concentration, which is due to the fact that waters below these layers have a very narrow range of thermal and haline variability, so that the oxygen solubility only vary in a very narrow range, and the consideration of molar concentration or

saturation concentration bears quasi-identical results in our analyses. However Rev4 suggested removing this part and we agree it can be simplified.

L71/72 will be rephrased. Indeed, we're unsure how ventilation effects can strictly be separated from biological effects when working with in-situ observations. What we meant in L71/72 *"Our analysis thus aims to isolate annual convective ventilation as a particular component of the complex Black Sea deoxygenation dynamics (Konovalov and Murray, 2001; Capet et al., 2016) that typically involves intricate biogeochemical and physical processes."* is only related to the protocol of the analysis. By evidencing a direct correlation between CIL ventilation and oxygenation at depth, we aimed to isolate the effect of this specific driver (ie. ventilation), although of course 1) correlation is not automatically causal, and 2) indirect impacts (eg. CIL influencing BGC processes in some ways) cannot be ruled out and are in fact most probably occurring.. To our knowledge, a strict distinction of the ventilation and BGC terms should involve a dedicated model study which is left for further work. Please consider also related issue *2.4.*

**1.8** *L73-76: Maybe this "section list" is not needed? If you keep it, the last sentence seems to be incomplete.*
-> We'd like to maintain the section list to support partial reading. It will be completed.

**1.9** *L85: Assuming that density increases with depth, this density criterion only defines the upper limit of the CIL. How is the lower limit defined?*
-> On the basis of the temperature threshold. 'CIL points' from a profile are required to meet both temperature and density criterion. While the density criterion is needed to define the upper boundary, the temperature criterion is sufficient to identify the lower boundary, due to the stable higher temperature of deep waters.

    In our processing, if a profile does contain 'CIL points', it is only considered for analysis if the CIL's upper and lower boundaries are identifiable (ie. 'non-CIL points' are present above and below CIL points). If a profile does not contain any CIL point, it is considered (with cold content=0), if it contains more than 6 points, and the deepest observations are deeper than 80m. This excludes the possibility of missing an existing CIL layer at the sampling location. The above criterium will be detailed upon revision of the manuscript.

**1.10** *L111: Include the overall number of Argo profiles used and also include the minimum and maximum number for individual years (and state corresponding years) in order to give an idea about sampling error.*
-> The numbers of used Argo profiles will be characterized accordingly.

**1.11** *Given the time series of CIL content rate of change (Fig. 6), I am also wondering whether it would make more sense to only use summer/fall profiles (as rate of change is very small during that period; analogous for the other datasets)?*
-> The answer is technical, and we would like to avoid overweighting the manuscript. We'll answer the reviewer's question here, but our position would be not to include it for the sake of readability, considering that it is sufficient to state clearly that an annual averaging perspective is considered for all sources.

    The most difficult time series to compose is probably the one from ship-borne profiles (ie. "Diva"), since it has to compose with (strongly) uneven sampling (both seasonally and spatially). This time series is extensively described in a dedicated publication (doi.org/10.1007/s10236-013-0683-4). In the present manuscript, we consider strictly the time series described in this publication (no temporal extension or revision of the methodology), in order to be able to simply refer to this previous work.

The time series is obtained as annual "trends" which is a by-product of the DIVA detrending methodology. To make it short, it corresponds to annual "additive anomalies" associated with observation points in order to consider these measurements in the construction of a long-term climatology. In particular, those annual trends are identified conjointly with seasonal trends, in the frame of an iterative process. So the notion of "annual average trend" is intimately embedded in the procedure. From thereon, the same "averaging-window" was adopted for each data source, as a requisite for comparability.

**1.12** *Table 1: I suggest listing the different datasets in the same order as in the text.*
-> We will reorder items in the table in order to have consistent ordering in the table and text.

**1.13** *I understand from the text description that the statistical model, based on atmospheric predictors, uses more than one (winter air temperature anomalies) predictors, right? If so, please state all of them either in the text, in the table or both.*
The statistical model in question does only consider winter air temperature (AT) anomalies, but the AT time series is considered several times with different time lags. So practically, it's $C \sim a\, T_{w} + b\, T_{w-1} + c\, T_{w-2} + d\, T_{w-3}$, where C is annual anomaly of CIL cold content, $T_w$ is AT anomaly of the preceding winter, $T_{w-1}$ is AT anomaly of preceding winter + 1year, and so on.. (eg. for C anomaly of 2001, $T_{w}$ is from Dec2000 to Mar2001, and $T_{w-1}$ is from Dec1999 to Mar2000, etc.. ). All this is explicitly stated in the reference given, as well as the fact that other predictors (e.g. winds, summer temperature) were considered but discarded by the stepwise procedure. This issue will be clarified in the text.

**1.14** *For convenience, the time periods covered by the different datasets could be added to the table.*
-> Time periods will be added to Table 1

**1.15** *L150-151: What is the smallest number of overlapping years between datasets? Is it appropriate to calculate correlations in the cases of least overlap?*
-> Those numbers are given explicitly in figure A1 that was erroneously transferred at the very end of the pdf. Smallest number is 8 years between "Argo" and "Atmos" time series, providing a p-value of <0.002 for the correlation.

**1.16** *Regarding the atmospheric predictor model and the 3D hydrodynamical model, have any of the observational datasets (Argo and ship casts) been used to calibrate/develop either model? If so, those datasets would not be fully independent (during overlapping periods), which would affect the composite C time series (especially, its uncertainty as it may in-/decrease if they were in/dependent) and should be addressed in the discussion.*
-> This point of inter-dependencies is answered in **1.2.** The question of the composite time-series construction is addressed in **4.5** (note that model time-series that may be affected by dependencies are given a smaller weight in this composition). We did not attempt to estimate the uncertainty associated with the composite time series, as deriving uncertainty estimates for individual sources in a way that is consistent and allows a combined estimation of uncertainty for the composite time-series would be too complex.

**1.17** *L156: I think Appendix A is incomplete as I couldn't find the information referred to here*
-> Indeed, the information was to be found on FigA1 (at the end of the manuscript).

**1.18** *Section 2.3: It would be nice to show a time series similar to Fig. 1a also for oxygen in order to put the more recent development in CIL oxygen into historical context. I understand that this is not possible for the two models but perhaps it could be done for the ship casts? This would be particularly useful with respect to climate change.*

-> The long term relationship between oxygen and CIL cold content extracted from ship-casts has been explicitly re-illustrated in Capet et al 2016, Biogeosciences, including some of the present Argo data. We believe it is inappropriate to reproduce this here. Yet, following suggestions from Rev3, we now propose to address the CIL-oxygen in large scale historical context (based on World Ocean Database data), using a TS diagram -> see. **3.13**.

**1.19** *L174/175: I am wondering whether the recently submitted work by Gordon et al. (https://doi.org/10.5194/bg-2020-119), who suggest a correction of BGC-Argo oxygen observations based on the sensor response time, could help to make use of both descending and ascending profiles?*

-> Thanks for this very interesting reference. In addition to the fact that complication arises to use both ascending and descending profiles, and although we understand that these complications can be technically alleviated on the basis of the proposed reference, we considered the following additional arguments : 1) descending and ascending profiles are very close to each other since they pertain to the same sampling cycle. While those might be relevant when considering very short time scales, they are quite redundant at the time scales of interests, and 2) in the dataset obtained from the coriolis center with the given selection criteria, we obtain 1800 descending O2 profiles and about 250 ascending profiles. We thus consider that including ascending profiles is not worth the burden of technical description in this precise case.

**1.20** *L203: It's probably my personal taste but I don't really like the term "routine" regime. Maybe "standard", "normal" or simply "average"?*

-> "Routine Regime" would be replaced by "average regime". Thanks for the suggestion.

**1.21** *L213-215: Would it make sense to show annual (or winter) average surface air temperature as a small panel in Fig. 5 to better demonstrate this link?*

-> In L213-215, we referred to 'warm regime' in terms of CIL cold content. Besides, the "Atmospheric" time series is explicitly based on winter temperature anomaly, in a cumulative way which is more directly relevant to the CIL cold content (cf. **1.13**). Also, we decline the reviewer's suggestion. We will, however, refer here to Oguz et al 2006 (https://doi.org/10.1016/j.jmarsys.2005.11.011) as regards long term variations in the winter air temperature.

**1.22** *L218: More out of interest: would it be possible to also get intra-annual resolution from the statistical model? E.g. by not using winter time averages of the descriptor variables but monthly averages or so?*

-> We believe this level of detail is beyond the scope given to the paper. Detailed works on CIL formation and seasonal development have been done and are referenced within the manuscript. See in particular Capet et al, 2014, Akpinar et al 2017, Miladinova et al 2018. Also in L218 and paragraph, the idea is to quantify CIL formation and destruction rate from the difference between consecutive stocks. This can only be done on the basis of complete inventory such as can only be obtained for the synoptic 3D model product.

**1.23** *L225: I don't understand what you mean with "before, during and at the end the thermocline setting". Mainly because I am not sure what you mean with "thermocline setting"; is it the thermocline formation or the period during which a near-surface thermocline exists?*

-> Poor wording indeed. "thermocline setting" will be replaced by "thermocline season".

**1.24** *L231: Oxygen is from Argo. Would it make sense to do the intra-annual comparison of CIL formation rates also based on Argo then (or a combined C using 3D model and Argo)?*

-> This relates also to the answer given in **1.22**. CIL formation is not achieved at once, nor in a single location. Rather it proceeds from multiple small scale events in the open part, and in the northwestern shelf (see Akpinar 2017 and Miladinova 2018 for respective importance of both formation pathways). Argo sampling of the CIL is incomplete (important spatial variability and subsampling). Although we assumed to use it to build an annual metric of CIL content (with the caution notice of Sect. 2.1), we wouldn't use Argo to derive CIL formation rates at smaller temporal scales which demands differentiation between complete inventory estimates.

**1.25** *Discussion: As stated in the very beginning, it would be good if differences to earlier studies/novelty would be highlighted more explicitly. In my understanding, the main differences in terms of methodology are the longer period and the basin-scale integrated approach, which is needed to detect the regime shifts. This needs to be made more clear and it should also be discussed what the advantages (for the purpose of this study) and possible limitations are.*

-> see **1.3.**

**1.26** *L286/287: If one of the main conclusions focuses on the combined linear-periodic model then this combined model should be included in Figs. 3 and 4.*

-> It will be added to Figure 3 and fig 4 will be removed. Please consider **4.9** and **4.10**

**1.27** *Appendix: There seem to be two figures missing: Fig. A1 (L307/308) and Fig. D1 (L379).*

-> Both figures A1 and D1 appear on the very last page of the pdf, available at the BG discussion website. As D1 is relatively heavy, scrolling down that page is somehow slow, which may explain why it has been overlooked. A lighter version will be embedded upon revision and figures will be placed consistently.

**1.28** *The appendix contains a few aspects that would fit into the discussion in most journals (incl. BG), e.g. the discussion of the suitability of the statistical methods. I assume this is owed to the previous submission to GRL. The authors may consider moving parts of it into the discussion (or even into the Methods, e.g. the statement on oxygen data on L378-380). However, it also works the way it is. Maybe the editor can have a say on that.*

-> Appendix A will be maintained as is. Appendix D will be suppressed, with essential parts included in the Methods. As concerns appendix B and C, please refer to **4.6** and **4.7** (essential part of the regime shift analysis method will be embedded in the text, details will be maintained in a streamlined appendix).

*Technical corrections: [ … ]*

-> These technical corrections will be considered in the revised manuscript, but are not addressed in detail in this response.

**Reviewer #2**

*2.1 The title sounds interesting and attractive. However, for me, the result sounds as follows: There were two cold winters in the period of 2012-2019, and in each of them was not only the amount of cold waters larger, but the concentrations of oxygen were also higher. There is nothing new, except that this has been recorded by BioArgo.*

This appears as a reductive description of the manuscript results.

Result section can be divided in :

- Construction of composite long term (1955-2019) time series for CIL cold content using four sources of different nature (models, observations), and assessment of the coherences and discrepancies between these sources.
- Analysis of the above time-series by comparing different descriptive models, incl. identification of regime shifts and detailing of the various regimes.
- Detailing the contribution of Cold water formation to oxygen ventilation (over Argo period), including vertical (pycnal) distribution of the ventilating action of CIL formation.

All those steps are presented with quantitative considerations that we believe to be relevant for further works addressing 1) Black Sea specific dynamics and 2) interlinks between ocean warming and deoxygenation.

*2.2 Neither, the scientific content of the manuscript supports this title.*

The title will be modified, following the suggestion given by Reviewer1, ie. "A new intermittent regime of convective ventilation threatens the Black Sea oxygenation status".

*2.3 We read in the Abstract: "Oxygen records from the last decade indicate a clear relationship between cold water formation events and oxygenation status at different isopycnal levels, suggesting a leading role of convective ventilation in the oxygen budget of the upper intermediate layers." This just repeats what has been known for many years. This finding is just a confirmation of previous knowledge (see several papers of Konovalov et al).*

We acknowledge that the term "suggesting" is inappropriate, and apologize for this. The fact that CIL formation contributes to ventilation is indeed not a novel discovery and wasn't considered as such. As underlined by reviewer2 this relationship has been considered several times in the past Black Sea litterature, including in the works of S. Konovalov.

However,

- considering this dynamics from the standing point of Argo profilers provided an insight at an intra-annual scale (ie. weekly), and enabled an detailed appreciation of the depth (pycnal) penetration of the CIL ventilation effect at an intra-annual time-scale, which is new.
- Here, we used Argo data to illustrate not only the ventilation effect, but mostly the absence of ventilation for the years where CIL ventilation does not take place, ie. in the context of the regime shift highlighted in the first part of the manuscript. This was done not to reveal the general ventilation role of CIL formation (as this was done before) but to highlight the potential consequences of a regime shift in CIL formation by revealing the absence of ventilation during the years where no new CIL waters are formed. This will be highlighted more explicitly upon revision of the Discussion Sect. 5 and abstract.

We will also further develop the introduction with a paragraph dedicated to past research on Black Sea oxygenation dynamics, in order to better position our findings with respect to those of previous studies.

*__2.4__ Possible shifts of temperature-oxygen relationships in different periods have also been broadly addressed in these studies. Authors say nothing about that.*

For the sake of discussion we consider that the reviewer refer to in-situ conditions, and to the long-term dynamics described and commented for instance by Konovalov et al. 2001 (in particular the well known Fig. 10, reproduced below), a dynamics that has been related to shifts in the biogeochemical terms of the oxygen budget operating over large time scales.

[Figure]

Fig. 10. Variations in oxygen concentration versus temperature in the layer of the main pycnocline ($\sigma_t = 15.4$). (Average values for individual cruises listed in Table 1 and liner fit for the periods before 1975 and after 1985).

Our manuscript does not consider the material needed to address biogeochemical terms (eg. nutrients and production). Instead it focuses on highlighting and documenting the CIL ventilation term of the oxygen budget equation and more precisely the absence of this term for some years of the recent decade. The data considered to this aim cover a relatively short period [2011-2019], for which major shifts in biogeochemical regimes, such as the historical Black Sea eutrophication and recovery, can be conservatively ruled out.

Finally, shifts in the long-term oxygen and CIL cold content relationships have been explicitly illustrated in Capet et al 2016, Biogeosciences, and we did not wish to repeat these in the present manuscript. See however request from Rev3 (*__3.12__* & *__3.13__*), to put the finding of the Argo Period, in relation with historical perspectives. .

*__2.5__ This brings me to the major criticism. Authors have to make clear what the new knowledge is, which can be gained from their study compared to older ones.*

-> Please refer to *__1.3__*

*2.6 One big problem is that there is a lack of balance in the manuscript. Much attention has been given to the long-term variability. On this subject, I cannot find anything new.*

Regarding the long term variability. To our knowledge this is the first time that CIL dynamics is documented using an unified, integrative metric over such an extended period, that multiple sources of information are compared, and that the regime shift analysis paradigm is compared to other descriptive models, which is important 1) to highlight the existence of thresholds in the response of CIL formation to atmospheric temperature and 2) to highlight the exceptional nature of the absence of CIL formation for some years within the currently prevailing regime.

*2.7 The intimate link between analyses of 65-year and 7-year time series is not clear. The good part of the research is the analysis of data after 2012. However, its relation to the previous periods is completely unclear, and neither is it well articulated in the manuscript.*

The last seven year period falls within the identified last regime, and is used to illustrate the impact of this regime shift on oxygenation conditions. The model time-series allows to interlink the long-term framework to the recent period, since it extend over both period while proceeding from a unique model setup (ie. forcings and physical assumptions). A dedicated effort will be deployed upon revision in order to better express this in the discussion section, which will be completed in agreement with *3.12* and *3.13*

*2.8  In this part* [ie. analysis of data after 2012.]*, authors should clearly describe which floats they use, and how these floats capture the temporal and spatial variability. Important to know is whether what is observed is a clear signal or just noise. The statistics of data, and how representative for the Black Sea state they are, need deeper consideration. Fig. 1 shows perhaps that data used in the analyses are not homogeneous. If the data is not homogenous, the subsequent interpretations of long-term changes are not credible.*

The spatial and seasonal distribution of oxygen Argo floats is easily accessible on numerous platforms including the ones referred to in the manuscript (see for instance the display tool of the Coriolis website). Also, adding maps of floats positions was not considered as essential.

This, of course, does not dismiss the need to properly consider the representativity of the data.

Here the reviewer comments on data analysis after 2012, hence the part on oxygen although the Fig. 1, which is referred to, concerns the CIL cold content.

- Regarding the CIL cold content series extracted from Argo, there is an expected spread between individual floats that is indeed illustrated on Fig 1 and reflects the well known spatial variability in CIL cold content. In the sake of setting this data set on the same level as other data sources annual means were extracted from the Argo dataset (see. L105-115). The resulting annual averages were afterwards compared to other data sources (see. Appendix A. and Fig A1) to ensure consistency of the analysis.
- Regarding oxygen, all available oxygen Argo data were used considering data selection criterion that are described in Sect 3. L173-179. The question of representativeness is considered and explicitly discussed on L. 180-185, and Fig. 2. In particular, it illustrates the spread of oxygen conditions derived from individual floats and shows that the use of pycnal scale provides a sufficient spatial homogeneity to characterize a basin wide condition. Finally, Fig. 7 also provides the interquartile ranges of monthly-binned Argo-derived oxygen concentrations at different pycnal levels. Considering that Argos profilers are spread across the basin, we would expect much more inconsistency and overlap between the inter-quartile ranges areas, if as the reviewer suggests, spatial variability overpasses the temporal

signal embedded in this dataset.  This will be expressed explicitly in the revision of paragraph L231-235.

**2.9** *The statement ". . . suggests that the CIL renewal, that was taking place systematically each year in precedent regimes, has now become occasional." contradicts what is known from earlier studies. They have to at least refer to Lee et al. (2002. Anthropogenic chlorofluorocarbons . . .) who claim that the residence time is ~5yr at 80-120m. I would recommend that they explain what the problem in this earlier estimate is, if any.*

We apologize here for a clear misunderstanding, due to inappropriate wording. Instead of "*complete renewal*" of the CIL layer as seems to have been understood here, we wanted to refer to a "partial renewal", ie. the formation of a certain volume of new cold water, adding to the remaining CIL content of the previous year and contributing to the gradual renewal of the CIL layer. We totally agree with the fact that the residence time of the CIL layer is larger than a year, this is indeed well known and was referred to (maybe imperfectly) on L.48-59.: *"If a well-formed CIL was present during the previous year, subsurface waters exposed to atmospheric cooling are already cold, which increases the amount of newly formed CIL waters (Stanev et al., 2003). Due to this positive feed-back and to the accumulation of CIL waters formed during successive years, the inter-annual CIL dynamics is better described when winter air temperature anomalies are accumulated over 3 to 4 years, rather than considered on a year-to-year basis."*

We will clarify the fact that CIL renewal time is larger than one year in the revised introduction, including references to the works advised by the reviewer.

**2.10** *The issue of regime shift is not convincing. The question is: can oxygen data over 7 years only identify regime shift? What was the previous oxygen regime?*

There seems to be another misunderstanding here.

- The regime shift paradigm is used to characterize the dynamics of CIL formation, and is based on the analysis of 64 years long time series.
- Oxygen data are used to illustrate the impact that this regime shift in CIL formation may bear on the oxygen dynamics.

**2.11** *What I see in the oxygen data are just two ventilation events, not a shift. Authors ignore referring to important works about regime shift (e.g., the review article of Oguz in Front. Mar. Sci., 25 April 2017). They have to study the references in this review. Based on my comments above, I am very sorry that I cannot recommend publishing.*

See *2.10* as concerns the remark on regime shift and oxygen. We will carefully reconsider *Oguz in Front. Mar. Sci., 25 April 2017),* and extend the introduction by considering environmental regime shifts in the Black Sea, however, note that we still want to consider ventilation aspects apart from other drivers. Again, the question of a shift in the biogeochemical terms of the Black Sea oxygenation balance is not directly addressed in the manuscript. We really want to focus on Cold water formation as the main ventilation mechanism, and to highlight a strong, sudden, non-linear reduction in this process, and to illustrate the impact it's likely to bear on Black Sea oxygenation status, regardless of other environmental shifts that may be happening in parallel (ie. at first order).

**Reviewer #3**

*This is a very interesting paper. Basically the authors:*
*    1. Assemble a time series data set of temperature data from the Black Sea from about*
*    1955 to the present combining real observations and model results.*
*    2. They integrate cold temperature anomalies in the Cold Intermediate Layer (CIL), relative to a*
*    reference temperature of 8.35°C which defines the upper and lower boundary of the CIL, which they*
*    call "the cold content".*
*    3. They analyze the (extensive) variability in this "cold content", using several different approaches*
*    (linear and sine functions), but settle on a technique they call the "regime shift" paradigm or hypothesis.*
*    4. Using regime shift they identify four periods that characterize different amounts of "cold content"*
*    5. They argue that these periods reflect variable degrees of ventilation of the CIL*
*    6. The last 11 years have been a period with unusually low ventilation and they argue that this is due to*
*    ocean warming.*

*I have a few specific comments,*

*3.1 The paper is a little hard to read because of the advanced data analysis techniques used and (though generally well written in English) some awkward word choices. I'm not sure who can fix that.*

-> We refer to exchanges with Rev4 as concerns the technical justification part that needs to be retained as essential to the main text, or that can be directed to the appendixes. A certain level of technical detail can not be avoided. As concerns the english, we will ensure to have the revised manuscript checked by native english speaker.

*3.2 L1 Abstract – a ~100 m ventilated surface layer is referred to but does that mean 0-100m No, it means the Cold Intermediate Layer which is more like 50 to 100m I think you should be more specific.*

-> No, in this sentence we indeed refer to the part that is not anoxic so 0-100m, and not 50 to 100m. But we understand the concern as the ventilation of the upper part (ie. 0-50m) is not related to CIL formation. The abstract will be revised to avoid this confusion.

*3.3 L20 The early literature (e.g., Tolmazin 1985, Progress in Oceanography 15, 217) argued that as it appeared that the sea surface in the central gyres never got cold enough for replenishment of the CIL by winter convective, that the main source of water to the CIL on an annual basis was from the NW shelf where the key density surface was cooled. I agree that we have much more data now and starting with Gregg and Yakushev (2005), who observed a ventilation event (with real data), we now know that ventilation can occur from the central gyre regions. But the NW shelf hypothesis has been totally left out of all papers since the 1990s, such as those by Akpinar, Ivanov, Oguz and others. I looked back at those papers and they don't even mention the Tolmazin argument, much less argue why it would not play a role. So as far as I can tell, the NW shelf is a possible source of ventilated water for the CIL. If the Tolmazin hypothesis has been disproved, I missed that. I think Capet et al should take that into account. It may not show up in their model, depending on how it is parameterized.*

-> This important point is discussed extensively in 3.7 below.

**3.4** *L29 Murray et al (1989) discovered the suboxic zone. Stanev et al (2018) is a nice paper but used model results to argue for what causes its origin.*
-> We will update the reference accordingly. We apologize for this mistake.

**3.5** *L104 Why not describe the data sources in the same order as presented in Table 1?*
-> We will reorder items in the table in order to have consistent ordering in the table and text (cf. **1.12**)

**3.6** *L126 How were the Atmospheric Predictors converted into C and CIL temperature variability in the water column?? I don't think anything is said.*
-> The entire procedure, and the resulting statistical model is explicitly detailed in the reference given (Capet et al 2014). We did not restate the procedure and results of this study, because we want to focus the present debate on new investigations and results. However, seeing this comment is coherent with request **1.13,** the statistical model will be given more explicitly.

**3.7** *L136 Does that 3D hydrodynamic model include source water from the NW shelf?*
-> The 5 km 3Dmodel, whose setup is described in the given references, does indeed resolve a shelf source of CIL waters (as well as a source from central gyres), but this is a result of internal dynamics, not a specifically imposed behavior. Because the reviewer seems interested in this question, I may refer to Fig 4 and 5, and Section 6.3 of A. Capet et al. / Deep-Sea Research II 77-80 (2012) 128–142, that uses the same model. It depicts a second EOF of CIL Cold content which could correspond to the relative importance between the two sources of CIL water formation, which relates well to a longitudinal gradient in sea surface temperature.

The more recent works of Miladinova et al 2018, further extends on detailing the relative contribution of the two sources of CIW. I understand and share the reviewer's interest. It would be relevant to investigate whether a trend exists in the respective importance of the two CIL formation mechanisms, and if these two mechanisms imply distinct biogeochemical consequences (they certainly do, as the biogeochemical signature of those water's origin are drastically different). However, we prefer to maintain the focus and the general 'integrated' point of view. Besides, making a distinction between CIW origins would only be possible for the 3D model. In short, this is a related and relevant question, but we consider it extends beyond the scope of the present manuscript. We will however state in Sect 2.1 that the CIL dynamics as resolved by the 3D model is explicitly described in the above reference.

**3.8** *Figure 5 with the intervals obtained from regime shift analysis is compelling. But I think back to the geological axiom "If I hadn't believed it, I wouldn't have seen it!". Visually (without the vertical lines) it looks like there is more variability than shows up as the std. deviations. But this is not my area of expertise so I hope other reviewers can evaluate that.*
-> See exchanges with reviewer4.

**3.9** *L173 Section 2.3 The absolute values of O2 are uncertain unless the sensors are carefully calibrated. I suspect they were not. The relative changes are probably OK.*
-> Agreed. We only consider relative changes here. Also, we used monthly median values of multiple buoys with different ages we may assume an acceptable overall bias.

**3.10** *L181 Why 2018 and not 2020?*

-> This is a mistake. 2010-2018 should be replaced by 2014-2018, ie. the period where multiple buoys are present at the same time, which better illustrates spatial variability.

**3.11** *L289 I think Konovalov and Murray (2001) showed this in a figure.*

-> That is totally true. Please consider our answer in *2.3*

**3.12** *I really prefer the real data over the model results, which just reflect what equations and parameters were put into the model. For this reason I would really like to see a T-S plot (with real data) maybe averaged (with std. dev.) for the 3 regimes, blown up to highlight the CIL region. The intervals in Figure 5 should show up clearly.*

-> Here follows a T-S diagram, focused on the CIL part of the plot, based on in-situ data from the World Ocean Database 2018 including CTD, Bottle, and profilers data over the 4 periods (ie. 1955-2020). The four panels on the left indicate individual measurements, as well as contour surrounding 75 % of observations (obtained by two-dimensional kernel density estimation, see cf. function kde2 from the R package MASS). Panel on the right, repeats the same contours for each period. The four periods indeed show up quite clearly. Even ignoring the contours computations, it clearly appears from the scatterplots that records of temperature higher or equal to 8.35°C in the potential density anomaly range [14.5-15.0], are rare exceptions during P1-P2-P3, and appears as the main rule in P4. The figure will be included and discussed in the discussion section.

[Figure]

*3.13* *I think it would be useful to explain another reason why the CIL is important. In my view (see Murray et al., 1991) it is because it plays an important role in the formation of all deep-water in the Black Sea. To a first approximation, all deep water in the Black Sea forms by linear 2-end member mixing between the Bosporus outflow and the CIL. This must be because salinity increases all the way to the bottom and the only source of salinity is the Bosporus. See Figure 12 of Murray et al (1991). This mixing occurs on the SW shelf (Latif et al., 1991). Any curvature in the T-S plots is due to temporal variability in the signature of the CIL endmember. This mixing can be seen in the T and S sections from the Bosporus to the shelf break from Gregg et al (1999). See Murray et al (1991) for more discussion. If the CIL is warmer, and less dense, how will that impact deep water ventilation? I think the deepest layers will be ventilated less frequently.*

[Figure]

**Temperature and Salinity along the Bosporus**

- **Some CIW advected into strait in upper half of interface**
- **T minimum in deep water erased over South Sill**

Gregg et al (1999)

-> We thank the reviewer for this very important comment. Keeping the focus on oxygen, we issued the following figure (left panel below) which represents oxygen saturation concentration on the T-S diagram (WOD data, all periods, bottles,CTD and profilers data selected with "good" quality flags)). Taking, for instance, the 14.5 kg/m³ isopycnal as the CIL end-member for mixing of deep waters, it clearly appears that this layer is less oxygenated in the case of low CIL content, which relates to the fact that the CIL layer is composed of a mixture between CIL from the previous years and annual added now oxygenated CIL waters. The right panel evidences this for specific sigma layer, In particular, it provides a link between the recent (Argo) period (fig 7 & 8), and the longer time scale as is required by Reviewer 2 (see. *2.7*). These figures will be included to enrich the discussion regarding the implication of a new CIL regime on the ventilation of deep waters.

[Figure]

**Reviewer #4**

*I was asked by the editors to review this manuscript with an eye towards an assessment of the statistical methods that were used. Oceanographically, the notion of different convective regimes and their link to Black Sea oxygen dynamics is established using (mainly) observational data. A causal relation, of course, cannot be inferred from this, but the physical linkages are plausible ones and great effort has gone into making sensible metrics for the cold intermediate layer, with a solid understanding of the limitations of each of the data types used. I don't know enough about the literature pertaining to convective processes, nor regional Black Sea oceanography, to comment on significance of this contribution and its novelty within oceanography. For the remainder of this this review, I will focus on the statistics and data analysis aspects.*

*The paper quantifies the ventilation of the Black Sea intermediate layers through a heat (cold) content metric, i.e. the CIL defined in eqn (1). Estimates for the values for the CIL over time are computed from 4 sources: the first two are observational data (ARGO floats and CTD casts); the next is output from a statistical model that relates air temperature to CIL; and the last is numerical ocean model output. The different CIL estimates have very good correspondence to one another, and a composite CIL time series is produced for further analysis. The central feature of the paper is a changepoint analysis of the CIL series was undertaken to isolate different convective regimes. This is then followed by correlation analysis to relate CIL changes to oxygen dynamics.*

*4.1 From an visual examination of the data, I doubt that the 4 regimes found exist as distinctive equilibrium states (if that is how one defines a regime), but are rather part of a continuously varying process with cycles and trends like we see in all climate data. These CIL data do clearly show underlying long period cycles and a decreasing trend since 1980. As for the existence of regimes, looking at the graphs without the regimes (Fig 3), and with the regimes superimposed (Fig 5) – one's eye is drawn to the regimes in Fig 5, but these are subtle at best. A statistical change point analysis will always pick up regimes, and in my experience the AIC criteria used here tends to choose overly complex model (here, implying more changepoints and regimes). But the above is a subjective assessment. Quantitatively, the analysis undertaken is clearly spelled out, its assumptions addressed, its application done properly, with the result that 4 regimes are statistically identified.*

-> We will stick with the last sentence. We acknowledge the fact that the regime shift perception is one among others, and the manuscript will be screened to ensure that this acknowledgement is clearly expressed. In particular, what appears most important to us is that the recent change in CIL dynamics, as depicted by the regime change approach, is not a slow trending one, but includes a step change towards a new phenology of Black Sea ventilation. This is why we insist on this regime shift description, and on highlighting its potential consequences. In order to underline the difference between the actual and former regimes, besides statistical considerations, we followed the advice of Rev3, and produced the more direct illustration proposed in *3.12*.

*4.2 These regimes here are changes in the level (mean) of theCIL over time; there appears to be no changes in the higher order statistical moments. Note that there are many different ways to do changepoint analyses, and each need not be only based on the mean level, but could use other metrics (variance, autocorrelation, skewness) to break up the series.*

-> It appeared to us that considering only the first order moment would already appear as a complex approach (see other reviewer comments in that sense), yet it is sufficient to make our point. This is why we discarded the option to consider other higher order metrics.

*4.3 I feel there is value in putting forth this regime analysis and the results for discussion to the wider community, and hence support publication of the paper. I would, however, downplay the ambitious claims of the title. This is an analysis that brings out the recent decline in the oxygenation state, its relation the ventilation and the CIL, and suggests the possibility of CIL regimes. But these regimes are subtle at best and not easily separable from the general downward trend in CIL. The last regime might be different from those in the past (less cold), but further work would be needed to verify/validate this as a new regime.*

-> The main idea is to oppose non-linear regime dynamics to smooth linear and sine trends models. Yes, this is gladly open to criticism, and future observations will tell more about the relevance of this paradigm. Yet, it appears worthy of presentation to the interested audience.

*4.5 My main criticism is that I found it difficult to follow the methodology and how it was applied. The only reason that I was able to do so was since I have used most all these techniques before, and so could 'read between the lines' as to what was being done. The latter comment is important since it means that a**dequate and understandable descriptions of the statistical approaches used need to be included in the main body of the manuscript (not just buried the appendix details)***. Important points are glossed over (the composite time series, the atmospheric predictors model for CIL, and in particular the regime shift model). To compound this, basic statistics are discussed at length (like overfitting, AIC, and model selection, and fitting the curves of Fig 3), but in a way that is not clearly linked to the problem. Below I provide specific suggestions.*

*Overall, however, it should be straightforward to insert the necessary methodological detail into the text, and streamline the appendices, so that the results would be understandable and reproducible by an educated reader. I view these as relatively minor changes (since everything is there, if you read carefully with an adequate background).*

-> See specific answers below.

*SPECIFIC:*

**4.6** *Title: a bit over-reaching. There is no explicit link made to climate. But the paper clearly demonstrates the CIL decline and de-oxygenation since 1980 whether or not these are really new regimes.*

-> Title will be changed: see **1.1.**

**4.7** *Describe the "Atmospheric Predictors" model in the text (line 126). It looks like a lagged regression model. Important to be specific. Why not a basic equation?*

-> Specific description will be given in addition to reference, incl. equation. See similar comments in **1.13** and **3.6.**

**4.5** *Describe the composite CIL time series (line 145) in more detail. It is a central quantity for the paper.*

- The construction of the time series was kept very simple. It simply consists of a weighted average of the 4 time-series, restricted to available sources for years during which all sources weren't available. In order to emphasize the value of direct observations, we used relative weights of 1 for observation and 0.5 for model time-series. We will ensure any ambiguity is alleviated in the text.
- Describing the time series itself is the object of Sect. 3.1. The text will be complemented with more general description (beyond the current statement of descriptive model parameters).
- The composite time series is depicted in Fig3 (stated in caption but missing in the color legend) -> **4.9**. It will be more clearly depicted in Fig5 by removing other individual sources time series (**-> 4.10**)

**4.6** *Describe the regime shift model in text. It is a change-point model. These are easy to explain, if tricky to code. Be clear in the text the R-package you used, what it is based on, and that there are lots of different kinds of changepoint analysis and algorithms. Because of this lack of description, it is unclear why the authors then talk about model selection in the next paragraph (because different numbers of changepoints imply distinct statistical models).*

-> We answer here several comments related to this issue.

- We will enforce the description of the regime shift analysis and change-point model in the main text (Sect. 2.2 first paragraph). This paragraph will aim for accessibility and to allow non-specialized readers to get the basic notions of change-point models.
- The appendix will be streamlined, and organized as follows :
  - Appendix A will be maintained as is : Comparison of C time series from multiple sources and definitions of metrics used, including Fig A1, that provides important information (see for instance request **1.15**).
  - Appendix B will give technical details for the change point model and regime shift analysis including : R-package, underlying principles and options used in the R functions, precisions regarding the issue of **4.8** ("are time locations counted as parameters for AIC computation?"), and validity of underlying assumptions. This will be compiled from the actual appendixes ("

Regime shift analysis", "Statistics", "Normality","Autocorrelation"  and "Biases between components of the composite time series".
- ○ Current appendix D will be removed and the discussion on oxygen absolute or saturation concentration will be simplified and embedded in the text.

**4.7** *I think this paragraph on model selection and fit/complexity metrics to be too general and elementary – best put in the appendix.*
-> We would like to maintain the discussion about the regime shift approach, and it's relevance to describe the C time series as compared to other paradigms (linear,etc..). Mostly because this discussion highlights different conceptions of the response of CIL ventilation to atmospheric trends, and potential future evolution of this ventilation (see L244-248). We acknowledge that the statistical arguments in favor of the regime shift paradigm, although objective, are subtle at best. Our aim is mostly to propose this paradigm as a plausible point of view, this will be stressed further in the conclusions. Because we want to maintain this discussion, it appears essential to keep a few lines on model selection in the Methods section. In our opinion the fact that it is too elementary (and we are unsure that it will appear so to all readers) does not justify relegate it to appendix. So, we propose to maintain a streamlined version of this paragraph in the Method section.

**4.8** *Regime shift analysis and number of changepoints. I am unclear on whether the Rpackage you used here computes the number of breakpoints as well as their time locations. The reason I bring this up is since you have used an AIC criterion to choose the number of breakpoints – is this your addition to the analysis, or is it part of the R-package? State clearly as this is the central step that determines the 4 regimes (and hence underpins your conclusions).*
-> The R package strucchange fits change-point models by computing their time location and section-specific averages (homoscedastic approach, see **4.8**) for a selected number of breakpoints (1 to 5 in our case). It provides RSS and BIC for each of those change points models. Here, we decided to use AIC for selection, following advice from colleagues, although both BIC and AIC provide the same ranking between the different change point models (ie. different by the number of change points) and between the optimal change point model and the linear, periodic and linear+periodic models. Of course, AIC computations for change point models accounts for the fact that mean for each period AND time locations of the change points are evaluated. **There is a correction here,** although it does not impact much on the results. Upon data manipulation (successive re-computation upon resubmission of this work) the time series used for AIC computation of the change point model was limited to 2017, while it is now extended to include 2019. The complete time-series has well been used for the design of the model and to get the means associated with segments, however, the AIC were derived in another version of the same scripts where the time series was erroneously limited to 2017 (and therefore shorter than the series used for the other models obviously leading to smaller AIC). So the only consequence is that AIC for the 4-segments model is corrected to 752, instead 730. 752 remains smaller than other models, so that no major changes in the conclusions arise, besides that the subtlety of this statistical support of the regime shift paradigm further needs to be stressed (cf. **4.1**).

**4.9** *Figure 3 and supporting text is not needed. Fitting a linear trend and a sinusoid to the CIL doesn't add anything to the paper (and the data is repeated in Fig 5 for the regime analysis).*

Following arguments given in 4.7, we propose to maintain Fig3, in its present form, while including also the linear+periodic model (as requested in **1.26**). To avoid repetition, Fig5 will include only the composite time-series

**4.10** F*igure 4. Similarly, all the model selection stuff like AIC versus changepoints is over-explained and too generic. It is enough to say in the text that AIC identified 3 breakpoints as optimal, then refer to appendix.*
-> Along successive submissions of this work, these parts have been displaced between supplementary materials, appendix and main body, several times, way and back. Here again, reviewer 1 asks instead to add a model to Figure 3 (cf. **1.26**), while Rev4 proposes to minimize this section and send it back to appendixes. We conclude that this is a matter of personal appreciation. What matters to us here is to stress the fundamental difference between slow-trending/periodic variability, as opposed to the "abrupt" change in ventilation regimes which is a consequence of the non-linear dynamics of CIL formation. To us, the regime shift analysis better corresponds to this type of dynamics. Of course, as stressed above, this will always remain a matter of personal appreciation, even if statistical considerations objectivate the approach (see **4.1**). For this reason, and because linear trends and periodic descriptive models have often been applied to describe Black Sea time series, we'd like to maintain this section in the main body. Yet, in agreement with the reviewer, the section will be streamlined and the importance of model selection minimized. It only serves the purpose to oppose two discussion points of view : the slow-trending or periodic routine to the new unprecedented regime, as will be stated more clearly. **We thus agree that the Fig4 gives unnecessary details, and propose to remove it, simply.** AIC of the models (optimal 3 change point models, linear, periodic, linear + periodic) will be given in the text as is presently done.

**4.11** *Figure 6. Define dC/dt (difference between median oxygen concentrations between successive years). Is this the annual oxygenation index you refer to in line 239?*
   ● Caption will be completed : "dC/dt are basin wide CIL cold content formation and destruction rates, obtained as differences between weekly integrated CIL cold content provided by the 3D model". So dc/dt on Fig 6 is a **Weekly CIL** formation/destruction index.
   ● The **Annual oxygenation** index referred to in L239 is defined on Lines 235-236.
   ● On Fig 8, we assess the correlation between this Annual Oxygenation index, and a corresponding **Annual CIL formation** index, obtained as the differences between annual values of the CIL composite time series.
-> We apologize for the lack of clarity. Upon revision we will pay a close attention to define all those metrics in the relevant parts of the section methods (ie. 2.1 for CIL related indexes and 2.3 for oxygen related indexes) and ensure consistent references to these metrics (ie. always using the exact same words).

**4.12** *Figure 8. No need for both significance (colours) and p-values (size) since they give the same information. Significance values are thresholds for p-values.*
-> We totally agree this is redundant information. We saw no harm in this, but agree now that this may be confusing. Figure 8 will be redrawn with equal size for each point, removing the point size legend, and thus considering only the significance color scale.

**4.13** *Rationalize the material in appendices. It is rather a strange grab bag of material, and perhaps is a consequence of a previous review process?*

-> See answer in *4.6*

**4.14** *Appendix A: Put a basic understandable description of the regime analysis the main body of text (as noted). Details here. Note also that there are many changepoint analysis of various levels of statistical sophistication. You've picked one of many.*
-> See answer in **4.6** (proposition of appendix structure) and **4.2** (selected of changepoint analysis)

**4.15** *Appendix B. I think this is a discussion of how the statistical assumptions of this regime analysis – which is based on linear regression - could be violated, and their possible consequences. Not sure you need most of this, and the important parts should be folded into Appendix A. Overall, I'll agree that within each regime, using average annual values, the normal iid assumption of OLS regression is OK (and hence the inference underlying the changepoint detection).*
-> We agree with the proposition and will integrate only the most important aspects in the dedicated Appendix (see **4.6**).

**4.16** *Appendix D. This appendix could be omitted and a brief statement in the text made as to how and why oxygen concentration vs saturation were used.*
-> We will suppress Appendix D and insert this important issue in the method. Please refer to **1.7** for a full answer on the question of the selected oxygen variable.

---

## Author Response (AR1)

**BG-2020-76**

**Link to previous manuscript**

- https://www.biogeosciences-discuss.net/bg-2020-76/

**Answers to Reviews**

We thank the editors and reviewer team for their insightful comments and precious advice.
Since several comments coincide between reviewers we decided to address their comments in this common document, which is addressed to all reviewers. The reviewer's comments are thus reproduced here in their integrality (in blue), with specific answers including when relevant references to comments from other reviewers. Some of the most critical remarks result from imprecise wording, leading to misunderstandings. Below, we propose some rephrasing of the respective paragraphs, so that these misunderstandings can be avoided. Finally, we'd like to express here our general gratitude to the reviewers and appreciation for the pertinence of their remarks. References are given relatively to the revised manuscript in the form RLxxx.

In summary, to address the main reviewer's comments we have:
- Extended the introduction, for a better positioning of the novelties of our research ->see  *2.3, 2.4, 2.9, 2.11, 3.12, 3.13.*
- Enforced the links between the recent (Argo) period and the longer-term period (1955-present)  *3.12, 3.13.*
- Revised the description of the statistical approach, and streamlined appendixes -> see *4.6, 4.7*
- Adopted oxygen saturation concentration -> see *1.7*.
- Clarified technical points, regarding
  - AIC derivation -> *4.8*
  - Statistical model from atmospheric conditions -> *1.13*
  - Independence between distinct time-series -> *1.2*
- Alleviate miscomprehension due to lack of clarity -> *2.8, 2.9, 2.10, 4.11*
- Restate clearly the new insights of this study, and develop the discussion in that regard -> *1.3*
- Changed the title *-> 1.1*

**Reviewer #1**

*The manuscript by Capet and co-authors presents a statistical analysis of cold intermediate layer (CIL) content and formation and its impact on oxygen levels in the Black Sea from the 1960s to 2019 combining different data sources (incl. observations and models). The two key findings are: (1) temporal changes in CIL water can only be described well by a model when taking into account regime shifts; (2) CIL water formation has entered a new warm regime (i.e. low formation) around 2008, which affects oxygen levels (through reduced ventilation) and is likely to also affect the biogeochemistry in the Black Sea. The manuscript is concise, well written and generally easy to follow. The presented statistical analysis seems sensible and appropriate to address the posed research questions.*

*__However, I need to state clearly that I am not very experienced in complex statistical analysis and it would be good if someone more knowledgeable in that field could evaluate this aspect of the study.__*
-> We believe Reviewer 4 fulfills this role and we thus refer to our answer to Reviewer 4 for statistical considerations.

*Most of my comments are only minor, primarily ask for some clarification, and should be easy to address by the authors. I have only a few somewhat bigger points:*

*__1.1__ The study does not isolate the climate change impact (as suggested by the title) on the change in Black Sea ventilation. In my understanding, this cannot be achieved by the applied method, hence, I suggest adapting the title as a simple fix.*
-> This point has been underlined by several reviewers. The title has thus been modified, following the suggestion given by Reviewer1, ie. "A new intermittent regime of convective ventilation threatens the Black Sea oxygenation status".

*__1.2__ It is not clear from the descriptions of the datasets whether the models are completely independent from the observational data, i.e. whether the same observations have been used for model calibration. If that's the case, this needs to be discussed.*
-> Table R2, now added to Method section describes the direct and indirect dependencies that may result from the construction of the 4 times series.
  ● We now refer explicitly to this discussion in the sentence L151-153.

*__1.3__ In the discussion, the authors should highlight more explicitly what the new insights of this study are compared to previous work on the topic (I understand it is the regime shift).*
First, following comments from Rev2, we extended the introduction, in order to better position our findings with respect to those of previous studies.

Considering the points addressed in this review, we can reformulate our main findings as follows :
  ● We issue an unified, synoptic metric to characterize inter-annual variations in CIL formation. The consistency between independent sources demonstrates the accuracy of this metric. Note that some dependencies exist among certain sources, this is discussed explicitly. To our knowledge, no multiple

source comparison have been achieved previously over such an extended period. This is now expressed explicitly in RL293-295

- Analysis of the time-series reveals a current restricted ventilation regime that is unprecedented over the last 64 years. Interannual variations appear to be better described by considering abrupt change models than smooth dynamics. This is discussed at length in the discussion section, and has been reinforced with Fig. 7, discussed on RL 304-309
- In particular the actual regime includes years during which **no** new CIL waters are formed, which appears to be a new phenomenon. We highlighted this in conference proceedings (EGU GA, May 2018, Kiel conference on marine deoxygenation, October 2018) . Stanev et al. 2019 later discussed this issue on the basis of Argo observations. Here, we propose a long term multi-source description of the CIL dynamics, to insist on the anomaly of this feature in regards to an extended period.
- The general relationship between CIL formation and oxygenation of intermediate waters has been previously described in the literature. Here, we exploit the dense Argo data sets (7 years) to document in more detail and at a finer scale the response of oxygen conditions to CIL formation at different pycnal levels. In particular, we illustrate the fast decrease in oxygen conditions for years where no CIL water was formed.  The fact this detailed illustration of the CIL ventilation effect at different pycnal layers was permitted by the combination of Model and Argo data is explicitly stated in RL363-365

The whole study stresses the current prevalence of a restricted ventilation regime, its impact on the Black Sea oxygenation status, and the urgency of research efforts dedicated to alleviate or at least foresee potential environmental and economic consequences.

*1.4 The appendix seems to be incomplete as there are references to non-existent figures.*
-> Figures A1 and D1 have been erroneously placed at the very end of the pdf, which is probably why the reviewer missed those. Those figures being quite heavy,  I've also experienced some lags while scrolling down to that part of the pdf on the Copernicus website. It has been ensured that all figures are distributed coherently in the revised manuscript. However, following the advice of reviewer 4, Appendix D has been removed, and replaced by corresponding addition in the text (RL212-215). The specific issue of Appendix D is further addressed in *1.7*

*Specific comments*
*1.5 Title: I suggest changing it to "A new intermittent regime of convective ventilation threatens the Black Sea oxygenation status" as the study cannot really isolate the climate change impact from that of regional atmospheric oscillations.*
-> Agreed. see *1.1.*

*1.6 L58: This should also be mentioned when describing the statistical model using atmospheric forcing in the Methods*
-> Several comments pointed towards a more explicit description of this statistical model. We centralized our answer in *1.13.*

*1.7 L71/72: I am not fully convinced that your analysis actually separates the convective ventilation from the biogeochemical processes (BGC). The observed oxygen from Argo is affected by both physics and BGC. In*

*order to isolate the physical (ventilation) component, would it make more sense to analyze oxygen saturation concentration (and AOU for changes in BGC)?*

-> For the sake of simplicity, and according to the suggestion, all oxygen considerations are now re-issued on the basis of saturation concentration (ie. in situ concentration as a percentage of oxygen solubility at in situ conditions). We attempted to indicate with Appendix D and Fig D1 that, for the layer below sigma 14.3 kg/m³ there is a quasi-strict linear relationship between molar concentration and saturation concentration, which is due to the fact that waters below these layers have a very narrow range of thermal and haline variability, so that the oxygen solubility only vary in a very narrow range. The consideration of molar concentration or saturation concentration thus bears quasi-identical results in our analyses. However Rev4 suggested removing appendix D and we agree with this simplification.

Also, the referred paragraph has been rephrased : "*Our analysis thus aims to pursue these investigations, and in particular to focus on the annual convective ventilation as an individual component of the complex Black Sea deoxygenation dynamics (Konovalov and Murray, 2001), in the context of the recent warming trend affecting the Black Sea (Miladinova et al., 2018)*". To our knowledge, a strict distinction of the ventilation and BGC terms should involve a dedicated model study which is left for further work. Please consider also related issue *2.4.*

**1.8** *L73-76: Maybe this "section list" is not needed? If you keep it, the last sentence seems to be incomplete.*

-> We'd like to maintain the section list to support partial reading. It has been completed.

**1.9** *L85: Assuming that density increases with depth, this density criterion only defines the upper limit of the CIL. How is the lower limit defined?*

-> On the basis of the temperature threshold. 'CIL points' from a profile are required to meet both temperature and density criterion. While the density criterion is needed to define the upper boundary, the temperature criterion is sufficient to identify the lower boundary, due to the stable higher temperature of deep waters. We reformulated the sentences LXXX in that sense

: "" *Although the use of a given temperature threshold to define the occurrence of convective mixing is subject to discussion, the existence of a fixed temperature threshold to characterize the CIL as a distinct water mass, and **in particular to identify its lower boundary**, is evident given the fixed value of ~ 9 ∘C that characterize the underlying deep waters (Stanev et al., 2019).* "

**1.10** *L111: Include the overall number of Argo profiles used and also include the minimum and maximum number for individual years (and state corresponding years) in order to give an idea about sampling error.*

-> We added "*In average, this set includes about 9 floats per year, with a minimum of two floats for 2005 and more than 12 floats from 2013 to 2019.*" RL148-149

**1.11** *Given the time series of CIL content rate of change (Fig. 6), I am also wondering whether it would make more sense to only use summer/fall profiles (as rate of change is very small during that period; analogous for the other datasets)?*

-> The answer is technical, and we prefer to avoid overweighting the manuscript. We answer the reviewer's question here, but opted not to include it in the manuscript for the sake of readability, considering that it is sufficient to state clearly that an annual averaging perspective is considered for all sources.

The most difficult time series to compose is probably the one from ship-cast profiles (ie. "Diva"), since it has to compose with (strongly) uneven sampling (both seasonally and spatially). This time series is extensively

described in a dedicated publication ([doi.org/10.1007/s10236-013-0683-4](doi.org/10.1007/s10236-013-0683-4)). In the present manuscript, we consider strictly the time series described in this publication (no temporal extension or revision of the methodology), in order to be able to simply refer to this previous work.

The time series is obtained as annual "trends" which is a by-product of the DIVA detrending methodology. To make it short, it corresponds to annual "additive anomalies" associated with observation points in order to consider these measurements in the construction of a long-term climatology. In particular, those annual trends are identified conjointly with seasonal trends, in the frame of an iterative process. So the notion of "annual average trend" is intimately embedded in the procedure. From thereon, the same "averaging-window" was adopted for each data source, as a requisite for comparability.

**1.12** *Table 1: I suggest listing the different datasets in the same order as in the text.*
-> Order of items in the text and in the table are now consistent.

**1.13** *I understand from the text description that the statistical model, based on atmospheric predictors, uses more than one (winter air temperature anomalies) predictors, right? If so, please state all of them either in the text, in the table or both.*
The statistical model used for this time series is now explicitly given at RL122

**1.14** *For convenience, the time periods covered by the different datasets could be added to the table.*
-> Time periods have been added to Table 1

**1.15** *L150-151: What is the smallest number of overlapping years between datasets? Is it appropriate to calculate correlations in the cases of least overlap?*
-> Those numbers are given explicitly in figure A1 that was erroneously transferred at the very end of the pdf. Smallest number is 8 years between "Argo" and "Atmos" time series, providing a p-value of <0.002 for the correlation.

**1.16** *Regarding the atmospheric predictor model and the 3D hydrodynamical model, have any of the observational datasets (Argo and ship casts) been used to calibrate/develop either model? If so, those datasets would not be fully independent (during overlapping periods), which would affect the composite C time series (especially, its uncertainty as it may in-/decrease if they were in/dependent) and should be addressed in the discussion.*
-> This point of inter-dependencies is answered in **1.2.** The question of the composite time-series construction is addressed in **4.5** (note that model time-series that may be affected by dependencies are given a smaller weight in this composition)**.** We did not attempt to estimate the uncertainty associated with the composite time series, as deriving uncertainty estimates for individual sources in a way that is consistent and allows a combined estimation of uncertainty for the composite time-series would be too complex**.**

**1.17** *L156: I think Appendix A is incomplete as I couldn't find the information referred to here*
-> Indeed, the information was to be found on FigA1 (at the end of the manuscript).

**1.18** *Section 2.3: It would be nice to show a time series similar to Fig. 1a also for oxygen in order to put the more recent development in CIL oxygen into historical context. I understand that  this is not possible for the two*

*models but perhaps it could be done for the ship casts? This would be particularly useful with respect to climate change.*

-> The long term relationship between oxygen and CIL cold content extracted from ship-casts has been explicitly re-illustrated in Capet et al 2016, Biogeosciences, including some of the present Argo data. We believe it is inappropriate to reproduce this here. Yet, following suggestions from Rev3, we now address the CIL-oxygen in large scale historical context (based on World Ocean Database data), using a TS diagram -> see. **3.13**.

**1.19** *L174/175: I am wondering whether the recently submitted work by Gordon et al. (https://doi.org/10.5194/bg-2020-119), who suggest a correction of BGC-Argo oxygen observations based on the sensor response time, could help to make use of both descending and ascending profiles?*

-> Thanks for this very interesting reference. In addition to the fact that complication arises to use both ascending and descending profiles, and although we understand that these complications can be technically alleviated on the basis of the proposed reference, we considered the following additional arguments : 1) descending and ascending profiles are very close to each other since they pertain to the same sampling cycle. While those might be relevant when considering very short time scales, they are quite redundant at the time scales of interests, and 2) in the dataset obtained from the coriolis center with the given selection criteria, we obtain 1800 descending O2 profiles and about 250 ascending profiles. We thus consider that including ascending profiles is not worth the burden of technical description in this precise case.

**1.20** *L203: It's probably my personal taste but I don't really like the term "routine" regime. Maybe "standard", "normal" or simply "average"?*

-> "Routine Regime" has been replaced by "standard regime". Thanks for the suggestion.

**1.21** *L213-215: Would it make sense to show annual (or winter) average surface air temperature as a small panel in Fig. 5 to better demonstrate this link?*

-> In L213-215, we referred to 'warm regime' in terms of CIL cold content. Besides, the "Atmospheric" time series is explicitly based on winter temperature anomaly, in a cumulative way which is more directly relevant to the CIL cold content (cf. **1.13**).

**1.22** *L218: More out of interest: would it be possible to also get intra-annual resolution from the statistical model? E.g. by not using winter time averages of the descriptor variables but monthly averages or so?*

-> We believe this level of detail is beyond the scope given to the paper. Detailed works on CIL formation and seasonal development have been done and are referenced within the manuscript. See in particular Capet et al, 2014, Akpinar et al 2017, Miladinova et al 2018. Also in L218 and paragraph, the idea is to quantify CIL formation and destruction rate from the difference between consecutive stocks. This can only be done on the basis of complete inventory such as can only be obtained for the synoptic 3D model product.

**1.23** *L225: I don't understand what you mean with "before, during and at the end the thermocline setting". Mainly because I am not sure what you mean with "thermocline setting"; is it the thermocline formation or the period during which a near-surface thermocline exists?*

-> Poor wording indeed. "thermocline setting" has been replaced by "thermocline season".

**1.24** *L231: Oxygen is from Argo. Would it make sense to do the intra-annual comparison of CIL formation rates also based on Argo then (or a combined C using 3D model and Argo)?*

-> This relates also to the answer given in **1.22**. CIL formation is not achieved at once, nor in a single location. Rather it proceeds from multiple small scale events in the open part, and in the northwestern shelf (see Akpinar 2017 and Miladinova 2018 for respective importance of both formation pathways). Argo sampling of the CIL is incomplete (important spatial variability and subsampling). Although we assumed to use it to build an annual metric of CIL content (with the caution notice of Sect. 2.1), we wouldn't use Argo to derive CIL formation rates at smaller temporal scales which demands differentiation between complete inventory estimates.

**1.25** *Discussion: As stated in the very beginning, it would be good if differences to earlier studies/novelty would be highlighted more explicitly. In my understanding, the main differences in terms of methodology are the longer period and the basin-scale integrated approach, which is needed to detect the regime shifts. This needs to be made more clear and it should also be discussed what the advantages (for the purpose of this study) and possible limitations are.*

-> see **1.3** for novelty. **The assets of considering the regime shift paradigm is discussed as follows : (RLXX)** *"Although, we acknowledge that the statistical advantage (AIC) of the regime shift description is subtle, we consider that it deserves further consideration as this difference in interpretation is fundamental in what regards the expected consequences on the Black Sea oxygenation status and in particular the threat on Black Sea marine populations, whose ecological adaptation (and rate of exploitation) have been built upon a ventilation regime and consequent biogeochemical balance, that may no longer prevail."*
The assets of combining different dataset in our analysis is now underlined in the sentences*:*
*RL354-356 : "The consistency between those products, and in particular the close correspondence between observational and mechanistic predictive time series supports the reliability of the composite series and its adequacy to describe the evolution of the Black Sea subsurface convective ventilation during the last 65 years."*
And *RL 363-365 "The synoptic CIL formation rates provided by the 3D hydrodynamical model, and the detailed description of oxygenation conditions provided by BGC-Argo floats, allowed us to detail the role of CIL formation in oxygenating, through convective ventilation, the upper part of the Black Sea intermediate layers".*
The asset of considering multiple data sources to extend the temporal extent the of study is mentioned in RLXX *"This composite time series is built from four different data products issued from observations and modelling, so as to optimize its temporal extent in regards to preceding studies \citep[e.g.][]{OGUZ2006}."*

**1.26** *L286/287: If one of the main conclusions focuses on the combined linear-periodic model then this combined model should be included in Figs. 3 and 4.*

-> In fact Fig. 3 and 4 have been removed. However, we completed the "descriptive models" section with explicit inclusion of the linear-periodic model. Please Also consider *4.9* and *4.10*

**1.27** *Appendix: There seem to be two figures missing: Fig. A1 (L307/308) and Fig. D1 (L379).*

-> Both figures A1 and D1 appear on the very last page of the pdf, available at the BG discussion website. As D1 is relatively heavy, scrolling down that page is somehow slow, which may explain why it has been overlooked. A1 is now placed consistently, while D1 was removed.

**1.28** *The appendix contains a few aspects that would fit into the discussion in most journals (incl. BG), e.g. the discussion of the suitability of the statistical methods. I assume this is owed to the previous submission to GRL. The authors may consider moving parts of it into the discussion (or even into the Methods, e.g. the statement on oxygen data on L378-380). However, it also works the way it is. Maybe the editor can have a say on that.*
-> Appendix A is maintained as was. Appendix D has been suppressed, with essential parts included in the Methods. As concerns appendix B and C, please refer to **4.6** and **4.7** (essential part of the regime shift analysis method has been embedded in the text, details have been maintained within a streamlined appendix).

**Technical corrections:**

- *L2-3: "from the mid-1970s to the early 1990s"; specify "recent years", e.g. "post-2005"? comma after "Here"* -> *Done*
- *L6: maybe "years without renewal of intermediate water"?*-> *Done*
- *L7: "density levels"*-> *Done*
- *L15: "While the reduction"* -> *Done*
- *L20: "at the surface"* -> *Done*
- *L24" "Strait"* -> *Done*
- *L27: no comma after "(halocline)"; remove "the" after "prevents"*-> *Done*
- *L33: "winter time"*-> *Done*
- *L40: "upper ~100"* -> *Done*
- *L41: "forcing"; "e.g." (check throughout the manuscript, also for "i.e.", the first "." is often missing)* -> *Done*
- *L42: give time scale for alterations, e.g. "(order of days)"* -> *"relatively fast inter-annual alterations"*
- *L53: "Miladinova et al. (2018)"* -> *Done*
- *L57: "feedback"* -> *Done*
- *L62: (2005-2018)* -> *Done*
- *L63: "trend leading to conditions"* -> *Done*
- *L65: "with increasing trends"* -> *Done*
- *L67: comma before "in particular"* -> *Done*
- *L68: remove "extended"* -> *Done*
- *L71: "an individual component"* -> *Done*
- *L74: "regime shift analysis"*-> *Done*
- *L78: use full term for CIL in section heading* -> *Done*
- *L81: "provides"* -> *Done*
- *L87: "water mass"* -> *Done*
- *L89: "for consistency with existing literature"* -> *Done*
- *L93: "i.e."* -> *Done*
- *L99: "dataset"* -> *Done*
- *L100: I'd suggest "errors" instead of "artifacts" since one usually uses the term "sampling error"; space before "Each data"*-> *Done*
- *L102: "netCDF"*-> *Done*
- *L107: "This data was"* -> *"This dataset was"*

- L110: "Good]."-> *Done*
- L115: Could you provide the total numbers of profiles in the central and peripheral basin regions? -> This would ask to define properly both regions, and appears as ineffective regarding the manuscript's objectives. However, the sentence has been rephrased : *"As Argo samplings are generally more abundant in the peripheral regions, i.e. outside of the divergent cyclonic gyres, this suggests that $C^{Argo}$ might be slightly biased towards high values."*
- L123: again I'd suggest "errors" -> *Done*
- Table 1: "guaranteed" instead of "granted" (under drawbacks for atmospheric predictors); "Three-dimensional" instead of "3D" in the rationale for GHER3D-> *Done*
- L124: please cite the "reference study" here-> *Done*
- L126: "consists of" -> *Done*
- L129: remove "defined earlier"-> *replaced by "defined by" + reference*
- L134: please cite the "reference study" here -> *Done*
- L136: "Three-dimensional (3D) hydrodynamic model" -> *Done*
- L136-144: Please include the time period for which the model was run*. -> Done*
- L147: "metric for the" -> *Done*
- L151: specify those series with little overlap " -> *Done*
- L154: here and throughout the manuscript: no space between "M J" in the unit; just "MJ" -> *Done*
- L159: "inter-annual" -> *Done*
- L161: "i.e." theme(legend.position ="none")+
- L164: "models' ability" -> *Done*
- L168: "i.e." -> *Done*
- L169: "which" instead of "that" -> *Done*
- L171: "shift" -> *Done*
- Fig. 1:
  - add legend; -> *If the reviewer agrees we prefer to describe the legend in the caption.*
  - larger panel and axes labels; -> *Done*
  - "MJ" instead of "M J" in y axes units; -> *Done*
  - no "-" between quantity and units on y axes (this applies to all figures); -> *Done*
  - caption: "Table" instead of "Tab.";- > *Done*
  - "time series" instead of "trend"; -> *'Trend' refers specifically to the DIVA methodology.*
  - why does the statistical model show a range and what value (e.g. mean) of that model was used for the analysis?-> *Caption completed "confidence bounds ($p$<0.01)".*
- L181: "Argo floats were operating" -> *Done*
- L182/183: "Argo floats profiling" -> *Done*
- Fig. 2: larger axes and panel labels; caption: "Argo floats" -> *Done*
- L185: use full term for CIL -> *Done*
- L187: "in Fig." -> *Done*
- L188: include use latex command "\cdot" instead of "." In "l1.i"; state what "i" is -> *Done*
- L193: "overestimate" -> *Done*
- Fig. 3:
  - in the legend, you use "Model3d", while you use "C^3dModel" on L140 (I suggest to use one term consistently); -> *All changed to Model3D.*

- - The figure has been removed.
- L196: "shift"; "i.e." -> *Done*
- L198: ";" before "not shown" -> *Done*
- L199: "i.e." -> *Done*
- Fig. 4: caption: "shift"; "point" -> *Done*
- L200: suggest using full term for "CIL" -> *Done*
- Fig. 5:
  - here you use "3D model" in the legend (different to Fig. 3), be consistent; -> *All changed to Model3D.*
  - caption: "blue shaded area" looks more purple; "gray shaded area" very hard to see, perhaps make slightly less transparent; no comma after "i.e." -> *Done*
- L207: no comma after "e.g." -> *Done*
- L209: ";" after "oscillation" -> *Done*
- L210: "20th"; "1950s; Ivanov et al., 2000))." -> *Done*
- L213: "prevails in the Black Sea since about ten years."; "low cold content" -> *Done*
- L217: suggest using full term for "CIL" -> *Done*
- L219: "before 2008" instead of "in precedent regimes"; specify "the latest period" -> *Done*
- L224: "period P2, P1/3 and P4, respectively." -> *Done*
- L226: for better readability: "about -1 MJ" -> *Done*
- L227: "in more detail" -> *Done*
- L228: "quasi-absence" and add reference to figure panel -> *Done*
- L229: "during" instead of "among"; "depict" -> *Done*
- L230: "while lower"; "simulated" or "shown" (this figure is not based on observations) -> *Done ("shown")*
- L231: unit missing for "16.0" -> *Done*
- L232: "increases"; "in the years 2012 and 2015—2017 when CIL formation is significant" -> *Done*
- L234: no space between "~ 14" -> *Done*
- L241: "decreases continuously" -> *Done*
- L242: "remain"; space before unit -> *Done*
- L247: "regime shift model" instead of "first" -> *Done*
- Fig. 6:
  - the yellow dots in P4 panel are difficult to see, suggest using different color; -> *Done*
  - add panel labels (a, b, c, d) for mores specific in-text referencing; -> *Done*
  - add "regime name" to panel titles; -> *Added "Period"*
  - caption: "time series"; "in Fig. 5" -> *Done*
- L250: "built" -> *Done*
- L251: "prevail" instead of "be considered as routine"? I think it's more important to emphasize that earlier assumptions might not be valid anymore -> *Done*
- L254: "trend" -> *Done*
- Fig. 7:
  - the shaded areas are hard to see, maybe make them a little less transparent? -> *Done*
  - Caption: "areas"; "." at end of caption -> *Done*
- Fig. 8:
  - significance is expressed by p-value, so why additional log10 of it? -> *Removed*

- ○ "Pearson Correlation Coefficient";-> *Done*
  - ○ Labels and legend should be a bit larger
- L265: "i.e."-> *Done*
- L269: "Sea" -> *Done*
- L271: put ", respectively" at end of sentence -> *Done*
- L284: "which has been" -> *Done*
- L286: space before "Statistical"; "indicate"-> *Done*
- L289: "i.e."; parenthesis not closed after sigma values -> *Done*
- L297: "feedbacks" -> *Done*
- L301/302: all links should be included in that section, not as footnotes-> *Done*
- L306: "denoted" -> *Done*
- L310-317: equations should be numbered; in the RMS equation, "N^m,n" should be under the square root; in the relative bias equation, use "\cdot" instead of "." -> *Done*
- L319: ";" before "Fig. 1"; no "." after "Fig. 1" -> *Done*
- L321: add reference for "published prognostic values" -> *Done*
- L331: "p-value" with cursive "p" -> *Done*
- L334: "six reject"-> *Done*
- L340: "i.e."; use "Akaike Information Criterion (AIC)" as you refer to this appendix before introducing AIC in the main text -> *Done*
- L341: this should be "Appendix C" (afterwards the numbering of appendix sections B1/B2/C1 seems off) -> Solved.
- L343: "guaranteed" -> *Done*
- L356: "Table C1; the"-> *Done*
- L360: "Table C1" -> *Done*
- Table C1, caption: "second column" and "third column" -> *Done*
- L365: "coverage"; comma after "For instance"-> *Done*
- L367: remove one "identified" -> *Done*
- L370: "in-situ data" -> *Done*
- L371: use "in situ" or "in-situ" consistently throughout the manuscript (preferably the former in cursive letters), here you use both-> *Done*
- L380: no space before the "."; don't use "in this study" instead of "in the following" -> *Done, i.e. replaced by "in this study", unclear comment.*
- L381: The heading for the "Author contributions" section is missing-> *Done*

**Reviewer #2**

*2.1 The title sounds interesting and attractive. However, for me, the result sounds as follows: There were two cold winters in the period of 2012-2019, and in each of them was not only the amount of cold waters larger, but the concentrations of oxygen were also higher. There is nothing new, except that this has been recorded by BioArgo.*

This appears as a reductive description of the manuscript results.

Result section can be divided in :

- Construction of composite long term (1955-2019) time series for CIL cold content using four sources of different nature (models, observations), and assessment of the coherences and discrepancies between these sources.
- Analysis of the above time-series by comparing different descriptive models, incl. identification of regime shifts and detailing of the various regimes.
- Detailing the contribution of Cold water formation to oxygen ventilation (over Argo period), including vertical (pycnal) distribution of the ventilating action of CIL formation.

All those steps are presented with quantitative considerations that we believe to be relevant for further works addressing 1) Black Sea specific dynamics and 2) interlinks between ocean warming and deoxygenation.

**2.2** *Neither, the scientific content of the manuscript supports this title.*
The title has been modified, following the suggestion given by Reviewer1, ie. "A new intermittent regime of convective ventilation threatens the Black Sea oxygenation status".

**2.3** *We read in the Abstract: "Oxygen records from the last decade indicate a clear relationship between cold water formation events and oxygenation status at different isopycnal levels, suggesting a leading role of convective ventilation in the oxygen budget of the upper intermediate layers." This just repeats what has been known for many years. This finding is just a confirmation of previous knowledge (see several papers of Konovalov et al).*
We acknowledge that the term "suggesting" is inappropriate, and apologize for this. The fact that CIL formation contributes to ventilation is indeed not a novel discovery and wasn't considered as such. As underlined by reviewer2 this relationship has been considered several times in the past Black Sea litterature, including in the works of S. Konovalov.

This is now explicitly mentioned in the introduction section RL73-80
However,
- considering this dynamics from the standing point of Argo profilers provided an insight at an intra-annual scale (ie. weekly), and enabled an detailed appreciation of the depth (pycnal) penetration of the CIL ventilation effect at an intra-annual time-scale, which is new.
- Here, we used Argo data to illustrate not only the ventilation effect, but mostly the absence of ventilation for the years where CIL ventilation does not take place, ie. in the context of the regime shift highlighted in the first part of the manuscript. This was done not to reveal the already known ventilation role of CIL formation but to highlight the potential consequences of a regime shift in CIL formation by revealing the absence of ventilation during the years where no new CIL waters are formed. This is now highlighted more explicitly in the Discussion Sect. 5 and Abstract.

**2.4** *Possible shifts of temperature-oxygen relationships in different periods have also been broadly addressed in these studies. Authors say nothing about that.*
Shifts in the temperature-oxygen relationships is now explicitly mentioned in the introduction part.
*(RLXX) :*
*"Indeed, (Konovalov and Murray, 2001) evidenced a clear relationship between oxygen conditions in the lower part of the CIL layer, and the temperature in that layer which is directly related to interannual variations in the CIL formation intensity. This relationship explain a large part of the inter-annual fluctuations in*

*oxygen concentration in that layer, which occur at a time scale of a few years, and are superimposed on the larger scale change in oxygenation condition that is attributed to an increase in the primary production induced by the eutrophication phase of the late 1970s."*

Our manuscript does not consider the material needed to address biogeochemical terms (eg. nutrients and production). Instead it focuses on highlighting and documenting the CIL ventilation term of the oxygen budget equation and more precisely the absence of this term for some years of the recent decade. The data considered to this aim (Sect 4.) cover a relatively short period [2011-2019], for which major shifts in biogeochemical regimes, such as the historical Black Sea eutrophication and recovery, can be conservatively ruled out.

Finally, shifts in the long-term oxygen and CIL cold content relationships have been explicitly illustrated in Capet et al 2016, Biogeosciences, and we did not wish to repeat these in the present manuscript. See however request from Rev3 (*3.12* & *3.13*), to put the finding of the Argo Period, in relation with historical perspectives. .

*2.5 This brings me to the major criticism. Authors have to make clear what the new knowledge is, which can be gained from their study compared to older ones.*
-> Please refer to *1.3*

*2.6 One big problem is that there is a lack of balance in the manuscript. Much attention has been given to the long-term variability. On this subject, I cannot find anything new.*
Regarding the long term variability: To our knowledge this is the first time that CIL dynamics is documented using an unified, integrative metric over such an extended period, that multiple sources of information are compared, and that the regime shift analysis paradigm is compared to other descriptive models, which is important 1) to highlight the existence of thresholds in the response of CIL formation to atmospheric temperature and 2) to highlight the exceptional nature of the absence of CIL formation for some years within the currently prevailing regime.

*2.7 The intimate link between analyses of 65-year and 7-year time series is not clear. The good part of the research is the analysis of data after 2012. However, its relation to the previous periods is completely unclear, and neither is it well articulated in the manuscript.*
The last seven year period falls within the identified last regime, and is used to illustrate the impact of this regime shift on oxygenation conditions. The model time-series allows to interlink the long-term framework to the recent period, since it extend over both period while proceeding from a unique model setup (ie. forcings and physical assumptions remains unchanged for the entire simulation time). A dedicated effort has been deployed to better express this in the discussion section, which has been completed in agreement with 3.12 and 3.13

*2.8  In this part* [ie. analysis of data after 2012.]*, authors should clearly describe which floats they use, and how these floats capture the temporal and spatial variability. Important to know is whether what is observed is a clear signal or just noise. The statistics of data, and how representative for the Black Sea state they are, need deeper consideration. Fig. 1 shows perhaps that data used in the analyses are not homogeneous. If the data is not homogenous, the subsequent interpretations of long-term changes are not credible.*

The spatial and seasonal distribution of oxygen Argo floats is easily accessible on numerous platforms including the ones referred to in the manuscript (see for instance the display tool of the Coriolis website). Also, adding maps of floats positions was not considered as essential.

This, of course, does not dismiss the need to properly consider the representativity of the data.

Here the reviewer comments on data analysis after 2012, hence the part on oxygen although the Fig. 1, which is referred to, concerns the CIL cold content.

- Regarding the CIL cold content series extracted from Argo, there is an expected spread between individual floats that is indeed illustrated on Fig 1 and reflects the well known spatial variability in CIL cold content. In the sake of setting this data set on the same level as other data sources annual means were extracted from the Argo dataset (see. RL150). The resulting annual averages were afterwards compared to other data sources (see. Appendix A. and Fig A1) to ensure consistency of the analysis. We also added a sentence giving annual number of floats used to characterize the CIL content (RL 149)

- Regarding oxygen, all available oxygen Argo data were used considering data selection criterion that are described in Sect 2.3. The question of representativeness is considered and explicitly discussed on L. 214-220 and Fig. 2, which now includes the Argo's ID. In particular, it illustrates the spread of oxygen conditions derived from individual floats and shows that the use of a pycnal scale provides a sufficient spatial homogeneity to characterize a basin wide condition. Finally, Fig. 5 also provides the interquartile ranges of monthly-binned Argo-derived oxygen concentrations at different pycnal levels. Considering that Argos profilers are spread across the basin, we would expect much more inconsistency and overlap between the interquartile range areas if, as the reviewer suggests, spatial variability overpasses the temporal signal embedded in this dataset.

**2.9** *The statement ". . . suggests that the CIL renewal, that was taking place systematically each year in precedent regimes, has now become occasional." contradicts what is known from earlier studies. They have to at least refer to Lee et al. (2002. Anthropogenic chlorofluorocarbons . . .) who claim that the residence time is ~5yr at 80-120m. I would recommend that they explain what the problem in this earlier estimate is, if any.*

We apologize here for a clear misunderstanding, due to inappropriate wording. Instead of "*complete renewal*" of the CIL layer as seems to have been understood here, we wanted to refer to a "partial renewal", ie. the formation of a certain volume of new cold water, adding to the remaining CIL content of the previous year and contributing to the gradual renewal of the CIL layer. We totally agree with the fact that the residence time of the CIL layer is larger than a year, this is indeed well known and is referred to in the revised RL60.: *"If a well-formed CIL was present during the previous year, subsurface waters exposed to atmospheric cooling are already cold, which increases the amount of newly formed CIL waters (Stanev et al., 2003). Due to this positive feed-back and to the accumulation of CIL waters formed during successive years, the inter-annual CIL dynamics is better described when winter air temperature anomalies are accumulated over 3 to 4 years, rather than considered on a year-to-year basis, which is in agreement with the 5 years upper bound estimate provided by Lee et al (2002) for the residence time in the CIL layer."*

**2.10** *The issue of regime shift is not convincing. The question is: can oxygen data over 7 years only identify regime shift? What was the previous oxygen regime?*

There seems to be another misunderstanding here.

- The regime shift paradigm is used to characterize the dynamics of CIL formation, and is based on the analysis of 64 years long time series.
- Oxygen data are used to illustrate the impact that this regime shift in CIL formation may bear on the oxygen dynamics.

**2.11** *What I see in the oxygen data are just two ventilation events, not a shift. Authors ignore referring to important works about regime shift (e.g., the review article of Oguz in Front. Mar. Sci., 25 April 2017). They have to study the references in this review. Based on my comments above, I am very sorry that I cannot recommend publishing.*

See **2.10** as concerns the remark on regime shift and oxygen. Note that we still want to consider ventilation aspects apart from other drivers. Again, the question of a shift in the biogeochemical terms of the Black Sea oxygenation balance is not directly addressed in the manuscript. We really want to focus on Cold water formation as the main ventilation mechanism, and to highlight a strong, sudden, non-linear reduction in this process, and to illustrate the impact it's likely to bear on Black Sea oxygenation status, regardless of other environmental shifts that may be happening in parallel.

**Reviewer #3**

*This is a very interesting paper. Basically the authors:*
*1. Assemble a time series data set of temperature data from the Black Sea from about*
*1955 to the present combining real observations and model results.*
*2. They integrate cold temperature anomalies in the Cold Intermediate Layer (CIL), relative to a reference temperature of 8.35°C which defines the upper and lower boundary of the CIL, which they call "the cold content".*
*3. They analyze the (extensive) variability in this "cold content", using several different approaches (linear and sine functions), but settle on a technique they call the "regime shift" paradigm or hypothesis.*
*4. Using regime shift they identify four periods that characterize different amounts of "cold content"*
*5. They argue that these periods reflect variable degrees of ventilation of the CIL*
*6. The last 11 years have been a period with unusually low ventilation and they argue that this is due to ocean warming.*

*I have a few specific comments,*
*3.1 The paper is a little hard to read because of the advanced data analysis techniques used and (though generally well written in English) some awkward word choices. I'm not sure who can fix that.*
-> We refer to exchanges with Rev4 as concerns the technical justification part that needed to be retained as essential to the main text, or that can be directed to the appendixes. A certain level of technical detail can not be avoided. As concerns the english, we hope that our careful revisions and the patient support of Reviewer1 will enhance the reading.

*3.2 L1 Abstract – a ~100 m ventilated surface layer is referred to but does that mean 0-100m No, it means the Cold Intermediate Layer which is more like 50 to 100m I think you should be more specific.*
-> No, in this sentence we indeed refer to the part that is not anoxic so 0-100m, and not 50 to 100m.

*3.3 L20 The early literature (e.g., Tolmazin 1985, Progress in Oceanography 15, 217) argued that as it appeared that the sea surface in the central gyres never got cold enough for replenishment of the CIL by winter convective, that the main source of water to the CIL on an annual basis was from the NW shelf where the key density surface was cooled. I agree that we have much more data now and starting with Gregg and Yakushev (2005), who observed a ventilation event (with real data), we now know that ventilation can occur from the central gyre regions. But the NW shelf hypothesis has been totally left out of all papers since the 1990s, such as those by Akpinar, Ivanov, Oguz and others. I looked back at those papers and they don't even mention the Tolmazin argument, much less argue why it would not play a role. So as far as I can tell, the NW shelf is a possible source of ventilated water for the CIL. If the Tolmazin hypothesis has been disproved, I missed that. I think Capet et al should take that into account. It may not show up in their model, depending on how it is parameterized.*

-> This important point is discussed extensively in *3.7* below.

*3.4 L29 Murray et al (1989) discovered the suboxic zone. Stanev et al (2018) is a nice paper but used model results to argue for what causes its origin.*

-> We have updated the reference accordingly. We apologize for this mistake.

*3.5 L104 Why not describe the data sources in the same order as presented in Table 1?*

-> We will reorder items in the table in order to have consistent ordering in the table and text (cf. *1.12*)

*3.6 L126 How were the Atmospheric Predictors converted into C and CIL temperature variability in the water column?? I don't think anything is said.*

-> The entire procedure, and the resulting statistical model is explicitly detailed in the reference given (Capet et al 2014). We did not restate the procedure and results of this study, because we want to focus the present debate on new investigations and results. However, seeing this comment is coherent with request *1.13,* the equation of this lagged regression model has been given explicitly.

*3.7 L136 Does that 3D hydrodynamic model include source water from the NW shelf?*

-> The 5 km 3Dmodel, whose  setup is described in the given references, does indeed resolve a shelf source of CIL waters (as well as a source from central gyres), but this is a result of internal dynamics, not a specifically imposed behavior. Because the reviewer seems interested in this question, I may refer to Fig 4 and 5, and Section 6.3  of  A. Capet et al. / Deep-Sea Research II 77-80 (2012) 128–142, that uses the same model. It depicts a second EOF of CIL Cold content which could correspond to the relative importance between the two sources of CIL water formation, which relates well to a longitudinal gradient in sea surface temperature.

The more recent works of Miladinova et al 2018, further extends on detailing the relative contribution of the two sources of CIW. I understand and share the reviewer's interest. It would be relevant to investigate whether a trend exists in the respective importance of the two CIL formation mechanisms, and if these two mechanisms imply distinct biogeochemical consequences (they certainly do, as the biogeochemical signature of those water's origin are drastically different). However, we prefer to maintain the focus and the general 'integrated' point of view. Besides, making a distinction between CIW of distinct origins would only be possible for the 3D model. In short, this is a related and relevant question, but we consider it extends beyond the scope

of the present manuscript. We've adapted RL134 to state that the CIL dynamics as resolved by the 3D model is explicitly described in this reference.

*3.8 Figure 5 with the intervals obtained from regime shift analysis is compelling. But I think back to the geological axiom "If I hadn't believed it, I wouldn't have seen it!". Visually (without the vertical lines) it looks like there is more variability than shows up as the std. deviations. But this is not my area of expertise so I hope other reviewers can evaluate that.*
-> See exchanges with reviewer4.

*3.9 L173 Section 2.3 The absolute values of O2 are uncertain unless the sensors are carefully calibrated. I suspect they were not. The relative changes are probably OK.*
-> Agreed. We only consider relative changes here. Also, we used monthly median values of multiple buoys with different ages we may assume an acceptable overall bias.

*3.10 L181 Why 2018 and not 2020?*
-> This is a mistake. 2010-2018 has been replaced by 2014-2018, ie. the period where multiple buoys are present at the same time, which better illustrates spatial variability.

*3.11 L289 I think Konovalov and Murray (2001) showed this in a figure.*
-> That is totally true. Please consider our answer in *2.3*

*3.12 I really prefer the real data over the model results, which just reflect what equations and parameters were put into the model. For this reason I would really like to see a T-S plot (with real data) maybe averaged (with std. dev.) for the 3 regimes, blown up to highlight the CIL region. The intervals in Figure 5 should show up clearly.*
-> We added the following figure in the discussion section to address this issue. Please consider the revised discussion section.

[Figure]

*3.13* *I think it would be useful to explain another reason why the CIL is important. In my view (see Murray et al., 1991) it is because it plays an important role in the formation of all deep-water in the Black Sea. To a first approximation, all deep water in the Black Sea forms by linear 2-end member mixing between the Bosporus outflow and the CIL. This must be because salinity increases all the way to the bottom and the only source of salinity is the Bosporus. See Figure 12 of Murray et al (1991). This mixing occurs on the SW shelf (Latif et al., 1991). Any curvature in the T-S plots is due to temporal variability in the signature of the CIL endmember. This mixing can be seen in the T and S sections from the Bosporus to the shelf break from Gregg et al (1999). See Murray et al (1991) for more discussion. If the CIL is warmer, and less dense, how will that impact deep water ventilation? I think the deepest layers will be ventilated less frequently.*

[Figure]

**Temperature and Salinity along the Bosporus**

- Some CIW advected into strait in upper half of interface
- T minimum in deep water erased over South Sill

Gregg et al (1999)

-> We thank the reviewer for this very important comment. Keeping the focus on oxygen, we issued the following figure which represents oxygen saturation concentration on the T-S diagram (WOD data, all periods, bottles,CTD and profilers data selected with "good" quality flags)). Taking, for instance, the 14.6 kg/m³ isopycnal as the CIL end-member for mixing of deep waters, it clearly appears that this layer is less oxygenated in the case of low CIL content, which relates to the fact that the CIL layer oxygen signature proceeds from a balance between underlying  biogeochemical consumption terms, and fresh oxygen imports associated with annual partial CIL renewal. Although we decline to explore the dynamical implications of a warmer CIL core on deep ventilation ventilation. We believe these considerations (please see revised discussion section) provides a link between the recent (Argo) period (fig 7 & 8), and the longer time scale as is required by Reviewer 2 (see. *2.7*).

[Figure]

*I was asked by the editors to review this manuscript with an eye towards an assessment of the statistical methods that were used. Oceanographically, the notion of different convective regimes and their link to Black Sea oxygen dynamics is established using (mainly) observational data. A causal relation, of course, cannot be inferred from this, but the physical linkages are plausible ones and great effort has gone into making sensible metrics for the cold intermediate layer, with a solid understanding of the limitations of each of the data types used. I don't know enough about the literature pertaining to convective processes, nor regional Black Sea oceanography, to comment on significance of this contribution and its novelty within oceanography. For the remainder of this this review, I will focus on the statistics and data analysis aspects.*

*The paper quantifies the ventilation of the Black Sea intermediate layers through a heat (cold) content metric, i.e. the CIL defined in eqn (1). Estimates for the values for the CIL over time are computed from 4 sources: the first two are observational data (ARGO floats and CTD casts); the next is output from a statistical model that relates air temperature to CIL; and the last is numerical ocean model output. The different CIL estimates have very good correspondence to one another, and a composite CIL time series is produced for further analysis. The central feature of the paper is a changepoint analysis of the CIL series was undertaken to isolate different convective regimes. This is then followed by correlation analysis to relate CIL changes to oxygen dynamics.*

*4.1 From an visual examination of the data, I doubt that the 4 regimes found exist as distinctive equilibrium states (if that is how one defines a regime), but are rather part of a continuously varying process with cycles and trends like we see in all climate data. These CIL data do clearly show underlying long period cycles and a decreasing trend since 1980. As for the existence of regimes, looking at the graphs without the regimes (Fig 3), and with the regimes superimposed (Fig 5) – one's eye is drawn to the regimes in Fig 5, but these are subtle at best. A statistical change point analysis will always pick up regimes, and in my experience the AIC criteria used here tends to choose overly complex model (here, implying more changepoints and regimes). But the above is a subjective assessment. Quantitatively, the analysis undertaken is clearly spelled out, its assumptions addressed, its application done properly, with the result that 4 regimes are statistically identified.*
-> We will stick with the last sentence. We acknowledge the fact that the regime shift perception is one among others, and the discussion section has adapted in this regard (RL297). In particular, what appears most important to us is that the recent change in CIL dynamics, as depicted by the regime change approach, is not a slow trending one, but includes a step change towards a new phenology of Black Sea ventilation. This is why we insist on this regime shift description, and on highlighting its potential consequences. In order to underline the difference between the actual and former regimes, besides statistical considerations, we followed the advice of Rev3, and produced the more direct illustration proposed in *3.12.*

*4.2 These regimes here are changes in the level (mean) of the CIL over time; there appears to be no changes in the higher order statistical moments. Note that there are many different ways to do change point analyses, and each need not be only based on the mean level, but could use other metrics (variance, autocorrelation, skewness) to break up the series.*

-> It appeared to us that considering only the first order moment would already appear as a complex approach (see other reviewer comments in that sense), yet it is sufficient to make our point. This is why we discarded the option to consider other higher order metrics.

*4.3 I feel there is value in putting forth this regime analysis and the results for discussion to the wider community, and hence support publication of the paper. I would, however, downplay the ambitious claims of the title. This is an analysis that brings out the recent decline in the oxygenation state, its relation the ventilation and the CIL, and suggests the possibility of CIL regimes. But these regimes are subtle at best and not easily separable from the general downward trend in CIL. The last regime might be different from those in the past (less cold), but further work would be needed to verify/validate this as a new regime.*
-> The main idea is to oppose non-linear regime dynamics to smooth linear and sine trends models. Yes, this is gladly open to criticism, and future observations will tell more about the relevance of this paradigm. Yet, it appears worthy of presentation to the interested audience.

*4.5 My main criticism is that I found it difficult to follow the methodology and how it was applied. The only reason that I was able to do so was since I have used most all these techniques before, and so could 'read between the lines' as to what was being done. The latter comment is important since it means that a**dequate and understandable descriptions of the statistical approaches used need to be included in the main body of the manuscript (not just buried the appendix details)**. Important points are glossed over (the composite time series, the atmospheric predictors model for CIL, and in particular the regime shift model). To compound this, basic statistics are discussed at length (like overfitting, AIC, and model selection, and fitting the curves of Fig 3), but in a way that is not clearly linked to the problem. Below I provide specific suggestions. Overall, however, it should be straightforward to insert the necessary methodological detail into the text, and streamline the appendices, so that the results would be understandable and reproducible by an educated reader. I view these as relatively minor changes (since everything is there, if you read carefully with an adequate background).*
-> See specific answers below.

*SPECIFIC:*
*4.6 Title: a bit over-reaching. There is no explicit link made to climate. But the paper clearly demonstrates the CIL decline and de-oxygenation since 1980 whether or not these are really new regimes.*
-> Title has been changed: see **1.1.**

*4.7 Describe the "Atmospheric Predictors" model in the text (line 126). It looks like a lagged regression model. Important to be specific. Why not a basic equation?*
-> The lagged regression model equation has been given in addition to the reference (RL122) . See similar comments in **1.13** and **3.6.**

*4.5 Describe the composite CIL time series (line 145) in more detail. It is a central quantity for the paper.*
- The construction of the time series was kept very simple. It simply consists of a weighted average of the 4 time-series, restricted to available sources for years during which all sources weren't available.  In order to emphasize the value of direct observations, we used relative weights of 1 for observation and 0.5 for model time-series.  We've ensured any ambiguity is alleviated in the text (cf RL165)

- Describing the time series itself is the object of Sect. 3.1.
- The composite time series is depicted in Fig5 (stated in caption but now added to the color legend).

**4.6** *Describe the regime shift model in text. It is a change-point model. These are easy to explain, if tricky to code. Be clear in the text the R-package you used, what it is based on, and that there are lots of different kinds of changepoint analysis and algorithms. Because of this lack of description, it is unclear why the authors then talk about model selection in the next paragraph (because different numbers of changepoints imply distinct statistical models).*

-> We answer here several comments related to this issue.
- We enforced the description of the regime shift analysis and change-point model in the main text (Sect2.2). This paragraph aims for accessibility and to allow non-specialized readers to get the basic notions of change-point models.
- The appendix have been streamlined, and are now organized as follows :
  - Appendix A is maintained as in the previous version : Comparison of C time series from multiple sources and definitions of metrics used, including Fig A1, that provides important information (see for instance request **1.15**).
  - Appendix B is now restricted to technical details for the change point model and regime shift analysis and validity of underlying assumptions. It has been compiled from the actual appendixes (" Regime shift analysis", "Statistics", "Normality","Autocorrelation" and "Biases between components of the composite time series").
  - Appendix D has been removed, and the discussion on oxygen absolute or saturation concentration has been simplified and embedded in the main text.

**4.7** *I think this paragraph on model selection and fit/complexity metrics to be too general and elementary – best put in the appendix.*

-> We would like to maintain the discussion about the regime shift approach, and it's relevance to describe the C time series as compared to other paradigms (linear,etc..). Mostly because this discussion highlights different conceptions of the response of CIL ventilation to atmospheric trends, and potential future evolution of this ventilation (see L244-248). We acknowledge that the statistical arguments in favor of the regime shift paradigm, although objective, are subtle at best. Our aim is mostly to propose this paradigm as a plausible point of view, which is now stressed with the sentence "*Although, we acknowledge that the statistical advantage (AIC) of the regime shift description is subtle, we consider that it deserves further consideration as this difference in interpretation is fundamental in what regards the expected consequences on the Black Sea oxygenation status and in particular the threat on Black Sea marine populations, whose ecological adaptation (and rate of exploitation) have been built upon a ventilation regime and consequent biogeochemical balance, that may no longer prevail.*".(RL297).

Because we want to maintain this discussion, it appears essential to keep a few lines on model selection in the Methods section. In our opinion, the fact that it is too elementary (and we are unsure that it will appear so to all readers) does not justify relegate it to appendix. We thus maintained a streamlined version of this paragraph in the Method section 2.2.

**4.8** *Regime shift analysis and number of changepoints. I am unclear on whether the Rpackage you used here computes the number of breakpoints as well as their time locations. The reason I bring this up is since you*

*have used an AIC criterion to choose the number of breakpoints – is this your addition to the analysis, or is it part of the R-package? State clearly as this is the central step that determines the 4 regimes (and hence underpins your conclusions).*

-> The R package strucchange fits change-point models by computing their time location and section-specific averages (homoscedastic approach) for a selected number of breakpoints (1 to 5 in our case). It provides RSS and BIC for each of those change points models. Here, we decided to use AIC for selection, following advice from colleagues, although both BIC and AIC provide the same ranking between the different change point models (ie. different by the number of change points) and between the optimal change point model and the linear, periodic and linear+periodic models. Of course, AIC computations for change point models accounts for the fact that mean for each period AND time locations of the change points are evaluated. This has been made more explicit in RL203.

**There is a correction to mention here,** although it does not impact much on the results. Upon data manipulation (successive re-computation upon resubmission of this work) the time series used for AIC computation of the change point model was limited to 2017, while it is now extended to include 2019. The complete time-series was correctly used for the design of the model and to get the means associated with segments, however, the AIC were derived in another version of the same scripts where the time series was erroneously limited to 2017 (and therefore shorter than the series used for the other models obviously leading to smaller AIC). So the only consequence is that AIC for the 4-segments model is corrected to 752, instead 730. 752 remains smaller than other models, so that no major changes in the conclusions arise, besides that the subtlety of this statistical support of the regime shift paradigm further needs to be stressed (cf. **4.1**).

**4.9** *Figure 3 and supporting text is not needed. Fitting a linear trend and a sinusoid to the CIL doesn't add anything to the paper (and the data is repeated in Fig 5 for the regime analysis).*
->We removed Fig 3, but maintained the discussion part, following arguments given in **4.7** and following the request **1.26**).

**4.10** *Figure 4. Similarly, all the model selection stuff like AIC versus changepoints is over-explained and too generic. It is enough to say in the text that AIC identified 3 breakpoints as optimal, then refer to appendix.*
-> Along successive submissions of this work, these parts have been displaced between supplementary materials, appendix and main body, several times, way and back. Here again, reviewer 1 asks instead to add a model to Fig. 3 (cf. **1.26**), while Rev4 proposes to minimize this section and send it back to appendixes. We can only conclude that this is a matter of personal appreciation. What matters to us here is to stress the fundamental difference between slow-trending/periodic variability, as opposed to the "abrupt" change in ventilation regimes which is a consequence of the non-linear dynamics of CIL formation. To us, the regime shift analysis better corresponds to this type of dynamics. Of course, as stressed above, this will always remain a matter of personal appreciation, even if statistical considerations objectivate the approach (see **4.1**). For this reason, and because linear trends and periodic descriptive models have been applied to describe Black Sea time series, we'd like to maintain this section in the main body. Yet, in agreement with the reviewer, the section has been streamlined and the importance of model selection minimized. It only serves the purpose to oppose two discussion points of view : the slow-trending or periodic routine to the new unprecedented regime, as will be stated more clearly. **We thus agree that the Fig4 gave unnecessary details, and removed it.**
AIC of the models (optimal  3 change point models, linear, periodic, linear + periodic) are now given in the text RLXX

**4.11** *Figure 6. Define dC/dt (difference between median oxygen concentrations between successive years). Is this the annual oxygenation index you refer to in line 239?*

- Caption has been completed : "Weekly averaged basin wide CIL cold content formation and destruction rates ($dC/dt$), obtained as differences between the weekly integrated CIL cold content provided by the 3D model.". So dc/dt on Fig 4 (former 6) is a **Weekly CIL** formation/destruction index.
- The **Annual oxygenation** index is defined on Lines RL283 (precising the "Annual" there and for the CIL formation mentioned below).
- On Fig 6, we assess the correlation between this Annual Oxygenation index, and a corresponding **Annual CIL formation** index, obtained as the differences between annual values of the CIL composite time series.

**4.12** *Figure 8. No need for both significance (colours) and p-values (size) since they give the same information. Significance values are thresholds for p-values.*

-> We totally agree this is redundant information. We saw no harm in this, but agree now that this may be confusing. This has been redrawn with equal size for each point, removing the point size legend, and thus considering only the significance color scale.

**4.13** *Rationalize the material in appendices. It is rather a strange grab bag of material, and perhaps is a consequence of a previous review process?*

-> See answer in *4.6*

**4.14** *Appendix A: Put a basic understandable description of the regime analysis the main body of text (as noted). Details here. Note also that there are many changepoint analysis of various levels of statistical sophistication. You've picked one of many.*

-> See answer in **4.6** (proposition of appendix structure) and **4.2** (selected of changepoint analysis)

**4.15** *Appendix B. I think this is a discussion of how the statistical assumptions of this regime analysis – which is based on linear regression - could be violated, and their possible consequences. Not sure you need most of this, and the important parts should be folded into Appendix A. Overall, I'll agree that within each regime, using average annual values, the normal iid assumption of OLS regression is OK (and hence the inference underlying the changepoint detection).*

-> We agree with the proposition and have integrated only the most important aspects in the dedicated Appendix (see **4.6**).

**4.16** *Appendix D. This appendix could be omitted and a brief statement in the text made as to how and why oxygen concentration vs saturation were used.*

-> We have removed Appendix D and inserted this important issue in the methods. Please refer to **1.7** for a full answer on the question of the selected oxygen variable.

[revised manuscript text omitted]

---

## Referee Report (RR1)

**(Re-)Review of Capet et al.**

The authors did a good job in addressing my comments and I think the manuscript has significantly improved, especially with regard to structure and inclusion of the most important methodological aspects, which were pointed out by Reviewer 4. I further think the authors made the right choices (in most cases) for what to move to the main text and what to keep in the appendices.

I am basically satisfied with the manuscript, I just have a few minor points that I would like to see addressed before endorsing publication in Biogeosciences. Therefore, I recommend publication after minor revisions. My detailed comments follow below. I don't need to see a revised version of the manuscript.

General comment

I think Reviewer 4 raised an important point (4.3 in the response document) that the new regime may not necessarily be a new "steady" regime—even though it has been identified as such by the change point analysis—but that it could be part of a downward trend that started in the mid-1990s. I appreciate the authors' response to the reviewer, which suggests that the data point to a non-linear change (i.e. different regimes) but that it may need more work (and probably a longer time series, which is not possible just now as it would need to be extended into the future) to verify that. I, therefore, would like to see a similar statement that this work proposes such new view (and that future data may be needed) in the Discussion and the Conclusions.

Specific comments (line numbers as, e.g. "L123")

L187: How did you verify the methodology? Do you refer to the discussion of the different assumptions provided in the appendix? Maybe remove "and verified", it's rather confusing.

L273-277: I seems like year 2013 really stands out in the time series as there is not even a slight sign of ventilation in February/March, different to all other years. Although it's rather a detail, it could be worthwhile to briefly mention it if you were able to link it to specific atmospheric conditions.

Fig. 4: For the reader's ease, you could consider adding two differently colored (e.g. blue/red) arrows starting at dC/dt = 0 (one up, one down) to the right of panels c and/or d with labels "formation" and "erosion". Caption: "Vertical dotted lines separate"; replace "evidenced" with "identified" or "detected".

L207/308: "almost never" seems to be a bit of an exaggeration. You could add the fractions of the records of such temperatures for the different periods to emphasize your point.

Fig. 7: I quite like this figure, however, I think you could combine it with Fig. 8: Panel e is not really needed as the 8.35°C line is a good reference for comparison between P4 and the other regimes, which is the key message. Perhaps you can instead plot Fig. 8a as panel e, with the 75% outlines (not filled) plotted over the O2-T-S data? Fig. 8b is not referred to anywhere, so I assume it can be omitted. Add time periods to labels of panels a-d. Caption: "contours delineate"; insert white space before "criterion"

L317/318: This is not exactly correct; the P4 regime corresponds with such reduction in frequency but this is not described by the regime shift analysis. Please rephrase.

L327-330: It took me quite a while to get the message from Fig. 8. I think the problem is that the time information is missing in Fig. 8a. Please see my suggestion above on combining Figs. 7 and 8a. This paragraph will then need rewriting.

L338: I think here would be a good location to add a discussion related to my general comment.

L362: Also related to my general comment, here would be a good location to add a brief statement that this study proposes this new "regime view" but that "only future will tell" if it is already a new regime or not. It's also questionable whether different superimposed oscillations with different periodicity can be resolved by a combination of one linear and one periodic function (as done in this study if I am not mistaken), which could be added to the Discussion as well (see previous comment).

L371-373: I suggest to rephrase this as it sounds like the Black Sea is a prototype ecosystem that can easily be compared to other ecosystems in terms of its O2 dynamics and effects of climate change. However, I would strongly disagree with that BECAUSE of its special geomorphology. Reading the following sentences, I understand that you don't mean to say that but it reads like this.

Technical corrections

L75: "explains"

L76: no comma after "years" but maybe split the sentence here to make it easier to follow; "oxygenation state"

L78: Maybe "to expand on these investigations"? "as a component"

L97: "characterizes"

L135: "present"

Table 2: This table leaves a lot of unused (white) space. I would suggest to either simply write the text in the table cells as regular text or to move the table to the appendix and state that "all data sets are either strictly independent or can be considered as such" with reference to the table in the appendix. Replace "time series" with "datasets" in the caption. Replace "data sets" and "data-set" in the Atmos-Model3D field with "datasets" and "dataset", respectively. "atmospheric fluxes' bulk formulations" (apostrophe after "fluxes"); "in situ"; "a common dataset"

L167: I wouldn't start a new line here.

L171: "e.g."; insert "and" between "C^Ship" and "C^Argo"

L172: remove text in parentheses; move "respectively" after "approaches" on next line

L178: "into a unique"

L184: "types"

L197: "i.e."

L207: I wouldn't start this sentence on a new line.

Fig. 3: Move labels to top (above figure frame) to not cover part of the time series; what are the red lines next to the "1984" and "2008"? Maybe also add the labels "P1" to "P4" to the figure as they are used frequently in the text.

L240: Should it be "standard regime"?

L242: "standard" instead of "routine" or just "this regime" (which makes it clear that it refers to the "standard regime")

L261/2622: Even though dC/dt < 0 implies erosion, erosion is positive: "between 0 and about 1 MJ m-2 d-1"

L265: insert Argo sampling period in parentheses after "sampling"

L269: "depicted in Fig. 5 denote"

Fig. 5: The data for the densest 2 or 3 levels look a bit odd in years 2011-2013; are those real data or are there data missing and points have just been connected by straight lines?

L288: "nonlinear system"

L289: "to external forcing". Only use "on the other hand" if "on the one hand" was used in the preceding statement. Perhaps "In contrast" at the start of the sentence works?

L291: "quantitative analysis presented in this study"

L294/295: "data sources used to construct this metric demonstrates"; "no multi-source"; "has been"; move "previously" before "no"

Fig. 6: the heading of the legend should state "p-value"; caption: no comma after "estimates"; "Color of the points"; everything after "while" can be removed

L304: "appear"

L307: "in situ" cursive and without dash

L308: "T > 8.35°C"; no [] around density range

L316: "described by Ivanov et al."

L320: Maybe "more profound consequence"?

L321: "Black Sea"; This text block is a bit lengthy and hard to follow. I suggest starting a new sentence after the references, e.g.: "This layer is characterized by a density between about 14.6 and 116 kg m^-3 and is formed following entrainment of CIL waters by the Bosporus inflow, the lower end-member, and subsequent lateral ventilation (Buesseler et al., 1991)." (Not sure if I placed the end-member note at the right location.)

L324: "years" instead of "yr" (twice)

L327: "indeed shows a generally deeper oxygenation"

L328: "characteristic for"

L333: "current" or "present" instead of "actual"

Fig. 8 caption: "T-S"; as stated I don't think Fig. 8b is needed (and it's not used in the text)

L336: "detail"; "asking, e.g. how"

L337: remove "on" before "planktonic"

L338: "detected" or "identified" instead of "depicted"

L340: remove "considering"; this ("ventilating … terms") is a bit repetitive from the previous lines

L345: "contributed 42% and 58%" (and remove "for 42% and 58%" at end of sentence)

L346: "assessments" or "an assessment"

L351: Maybe "For this purpose"?

L358: "standard regime"

L361: "cannot"

L363: "hydrodynamic"

L364/365: move "through convective ventilation" to end of sentence

L365-370: Very long sentence. I would try to split it after making the point that changes in CIL formation will affect oxygen. Then you can state that this will in turn have an impact on ecosystem structure/functioning given the structuring role of oxygen (please add a reference for this, too).

L367: "dominant"

L371: remove "on" after "impacts"

Fig. A1: Please add to the caption what the different acronyms mean ("sd", "RMS")

---

## Author Response (AR2)

**BG-2020-76- Revision Nov2020**

**Link to manuscript**

- https://www.biogeosciences-discuss.net/bg-2020-76/

**Answers to Reviews**

We thank the editors and Fabian große for their insightful comments and precious advice.
The reviewer' comments are reproduced here in their integrality (in blue), with specific answers (in black).
We'd like to repeat our gratitude to the reviewers and our appreciation for the pertinence of their remarks.

**(Re-)Review of Capet et al.**

The authors did a good job in addressing my comments and I think the manuscript has significantly improved, especially with regard to structure and inclusion of the most important methodological aspects, which were pointed out by Reviewer 4. I further think the authors made the right choices (in most cases) for what to move to the main text and what to keep in the appendices. I am basically satisfied with the manuscript, I just have a few minor points that I would like to see addressed before endorsing publication in Biogeosciences.
Therefore, I recommend publication after minor revisions. My detailed comments follow below. I don't need to see a revised version of the manuscript.

**General comment**

- I think Reviewer 4 raised an important point (4.3 in the response document) that the new regime may not necessarily be a new "steady" regime—even though it has been identified as such by the change point analysis—but that it could be part of a downward trend that started in the mid-1990s. I appreciate the authors' response to the reviewer, which suggests that the data point to a non-linear change (i.e. different regimes) but that it may need more work (and probably a longer time series, which is not possible just now as it would need to be extended into the future) to verify that. I, therefore, would like to see a similar statement that this work proposes such new view (and that future data may be needed) in the Discussion and the Conclusions.
  - Following related suggestions below, we included a dedicated paragraph in the discussion (L331) , which reads :"*It is important to highlight that the new regime may not necessarily be a new "steady" regime — even though it has been identified as such by the change point analysis — but that it could be part of a transient downward trend that started in the mid-1990s. The*

*reason why it appeared important to us to oppose non-linear regime dynamics to smooth linear and sine trends models is that the recent CIL dynamics, when depicted by the regime change approach, is not a slow trending one, but suggests a step change towards a new phenology for the intermediate Black Sea. Only future observations may now confirm or infirm the relevance of the proposed regime shift paradigm."*

- ○ In addition, an additional sentence was added in the conclusion, as suggested below. L366 : "*However, monitoring the future evolution of the CIL is necessary to confirm that this abrupt change description should be favored to that of a transient dynamics.*"

**Specific comments (line numbers as, e.g. "L123")**

- L187: How did you verify the methodology? Do you refer to the discussion of the different assumptions provided in the appendix? Maybe remove "and verified", it's rather confusing.
    - ○ -> We refer indeed to Appendix B, and in particular the procedure to test the presence of significant structural changes. We added (L187) a reference to this appendix for clarity.
- L273-277: I seems like year 2013 really stands out in the time series as there is not even a slight sign of ventilation in February/March, different to all other years. Although it's rather a detail, it could be worthwhile to briefly mention it if you were able to link it to specific atmospheric conditions.
    - ○ We thank the reviewer for this attentive remark. However, upon investigation and a closer look to Fig. 1, it appears that Argo floats show "some" CIL formation in 2013. As the 'anomaly' of 2013 cannot be supported at the multi-source level, we prefer not to open this discussion.
- Fig. 4: For the reader's ease, you could consider adding two differently colored (e.g. blue/red) arrows starting at dC/dt = 0 (one up, one down) to the right of panels c and/or d with labels "formation" and "erosion". -> Done
    - ○ Caption: "Vertical dotted lines separate" -> Corrected
    - ○ replace "evidenced" with "identified" or "detected". -> 'identified'
- L207/308: "almost never" seems to be a bit of an exaggeration. You could add the fractions of the records of such temperatures for the different periods to emphasize your point.
    - ○ To support the objectivity of this statement, we included additional density contours on Fig 7a-d. The statement has be reformulated accordingly and now reads : "*The extent to which the current regime differs from the previous ventilation regimes is clearly illustrated on a T-S diagram: in situ measurements from period P4 are commonly found in a range of the T-S diagram (T > 8.35°C and σ within 14.5–15 kg m−3 ) that was extremely rare in previous periods (see density contours on Fig. 7a,b,c,d; two-dimensional density estimates were obtained with R function MASS : kde2)*."
- Fig. 7: I quite like this figure, however, I think you could combine it with Fig. 8: Panel e is not really needed as the 8.35°C line is a good reference for comparison between P4 and the other regimes, which is the key message. Perhaps you can instead plot Fig. 8a as panel e, with the 75% outlines (not filled) plotted over the O2-T-S data? Fig. 8b is not referred to anywhere, so I assume it can be omitted.
    - ■ -> We modified Fig7 as suggested, see below. Fig8 has been removed.
    - ■ -> We computed bin-average for oxygen saturation values, and opted for a binned color scale to simplify the figure.
    - ■ Only the 75% contours for P2 and P4, the two extreme cases were included on panel e), for clarity.

[Figure]

- ○ Add time periods to labels of panels a-d. -> Added
- ○ Caption: "contours delineate";-> Corrected
- ○ insert white space before "criterion"- > Added
- L317/318: This is not exactly correct; the P4 regime corresponds with such reduction in frequency but this is not described by the regime shift analysis. Please rephrase.
  - ○ Indeed. The sentence has been replaced -> "Although inter-annual fluctuations in CIL formation rate still occur, **the regime shift analysis specifically describes a reduction in the average CIL cold content, which appears to be associated with a reduction in the frequency of significant ventilation events (Fig. 4)**, and therefore a potential decrease in the oxygen saturation signature in the lower part of the CIL."
- L327-330: It took me quite a while to get the message from Fig. 8. I think the problem is that the time information is missing in Fig. 8a. Please see my suggestion above on combining Figs. 7 and 8a. This paragraph will then need rewriting.
  - ○ -> See above for manipulation on the Figs 7,8
  - ○ Paragraph is reformulated as "*Displaying historical oxygen saturation data (1955--2020) on the T-S diagram (Fig 7e) indeed shows a generally deeper oxygenation during high CIL regimes than during low CIL regimes. For instance, oxygen saturation in the density range σ = 14.8 − 15.2 kg m−3 lies in the range of 30-70% in the T-S region characteristic to P2, while it only reaches 10-50 % in the T-S region specific to P4.*"
- L338: I think here would be a good location to add a discussion related to my general comment.
  - ○ See our answer to the general comment
- L362: Also related to my general comment, here would be a good location to add a brief statement that this study proposes this new "regime view" but that "only future will tell" if it is already a new regime or not. It's also questionable whether different superimposed oscillations with different periodicity can be resolved by a combination of one linear and one periodic function (as done in this study if I am not mistaken), which could be added to the Discussion as well (see previous comment).
  - ○ See our answer to the general comment
- L371-373: I suggest rephrasing this as it sounds like the Black Sea is a prototype ecosystem that can easily be compared to other ecosystems in terms of its O2 dynamics and effects of climate change.

However, I would strongly disagree with that BECAUSE of its special geomorphology. Reading the following sentences, I understand that you don't mean to say that but it reads like this.

- We completed the sentence as follows : "*To understand how global warming impacts the marine deoxygenation dynamics is a worldwide concern. The relatively fast and clear response that stems from the specific Black Sea geomorphology makes it a natural laboratory to study this dependency and related phenomena, although the specificity of this morphology also limits the direct transposition of Black Sea results to the global ocean*".

**Technical corrections**

- L75: "explains" -> Corrected
- L76: no comma after "years" but maybe split the sentence here to make it easier to follow;
  - The sentence has been split -> "*This relationship explains a large part of the inter-annual fluctuations in oxygen concentration in that layer, which occur at a time scale of a few years. Those fluctuations are superimposed on the larger scale change in oxygenation state that is attributed to an increase in the primary production induced by the eutrophication phase of the late 1970s.* "
- "oxygenation state" -> changed (see above)
- L78: Maybe "to expand on these investigations"? "as a component" -> both changed as recommended.
- L97: "characterizes" -> Corrected
- L135: "present" -> Corrected
- Table 2: This table leaves a lot of unused (white) space. I would suggest to either simply write the text in the table cells as regular text or to move the table to the appendix and state that "all data sets are either strictly independent or can be considered as such" with reference to the table in the appendix.
  - The table was moved into Appendix A. The sentence in Sect 2.1 now reads : "*With the exception of the pair C Atmos i and C Ships i , all data sets are either strictly independent or can be considered as such (see Tab. A1)*.
  - Replace "time series" with "datasets" in the caption. -> Done
  - Replace "data sets" and "data-set" in the Atmos-Model3D field with "datasets" and "dataset", respectively.  -> Done (x2)
  - "atmospheric fluxes' bulk formulations" (apostrophe after "fluxes"); "in situ"; "a common dataset" -> Done (x3)
- L167: I wouldn't start a new line here. -> Done
- L171: "e.g."; insert "and" between "C^Ship" and "C^Argo" -> Done  (x2)
- L172: remove text in parentheses; move "respectively" after "approaches" on next line -> Done  (x2)
- L178: "into a unique"-> Done
- L184: "types"-> Done
- L197: "i.e."-> Done
- L207: I wouldn't start this sentence on a new line.-> Done
- Fig. 3: Move labels to top (above figure frame) to not cover part of the time series; what are the red lines next to the "1984" and "2008"? Maybe also add the labels "P1" to "P4" to the figure as they are used frequently in the text.
  - Figure 3 and caption now appear as below (colors seem to be affected here, but are as before in the pdf).

[Figure]

**Figure 3.** Multi-decadal variability of the Black Sea CIL cold content and distinct periods identified by the regime shift analysis (P1, P2, P3 and P4). Confidence intervals on mean $C$ values are indicated by the orange shaded area for each period, and by the gray shaded area for the null hypothesis (i.e. considering no regime shifts). Confidence intervals on the time limits of each period are indicated with red error bars.

- L240: Should it be "standard regime"?-> Changed
- L242: "standard" instead of "routine" or just "this regime" (which makes it clear that it refers to the "standard regime")-> "This regime'
- L261/2622: Even though dC/dt < 0 implies erosion, erosion is positive: "between 0 and about 1 MJ m-2 d-1" -> Agreed and changed
- L265: insert Argo sampling period in parentheses after "sampling"-> Done
- L269: "depicted in Fig. 5 denote" > Done (x2 : "in" and ",")
- Fig. 5: The data for the densest 2 or 3 levels look a bit odd in years 2011-2013; are those real data or are there data missing and points have just been connected by straight lines?
    - The revised Fig 5 now excludes months without data instead of 'connecting' surrounding points. That's an important correction, thanks.
- L288: "nonlinear system"
- L289: "to external forcing".-> Done
- L289: Only use "on the other hand" if "on the one hand" was used in the preceding statement. Perhaps "In contrast" at the start of the sentence works? -> Modified as suggested
- L291: "quantitative analysis presented in this study"-> Modified as suggested.
- L294/295: "data sources used to construct this metric demonstrates";-> Completed as suggested
    - "no multi-source";-> Corrected
    - "has been";-> Corrected
    - move "previously" before "no" -> "*Previously*" was moved before "*achieved*" instead. The sentence now reads : "*To our knowledge, no multi-source comparison has been previously achieved over such an extended period.*"
- Fig. 6: the heading of the legend should state "p-value";-> Done
    - caption: no comma after "estimates"; -> Removed
    - "Color of the points"; everything after "while" can be removed -> Corrected
- L304: "appear" -> Corrected
- L307: "in situ" cursive and without dash -> Done
- L308: "T > 8.35°C"; no [] around density range -> Done(x2)
- L316: "described by Ivanov et al." -> Done
- L320: Maybe "more profound consequence"? -> It really is "deep" in the sense related to depth. See reformulation of the sentence proposed below.

- L321: "Black Sea"; This text block is a bit lengthy and hard to follow. I suggest starting a new sentence after the references, e.g.: "This layer is characterized by a density between about 14.6 and 116 kg m^-3 and is formed following entrainment of CIL waters by the Bosporus inflow, the lower end-member, and subsequent lateral ventilation (Buesseler et al., 1991)." (Not sure if I placed the end-member note at the right location.)
    - The revised paragraph now reads : "*Importantly, this reduction may also affect deeper layers of the Black Sea. Indeed, the mid-pycnocline (σ between about 14.6 and 16 kg m−3 ) is formed by the two end-member mixing line between the CIL layer and the Bosphorus inflow (Murray et al., 1991; Ivanov et al., 2000), which proceeds from the entrainment of CIL waters by the Bosporus inflow and subsequent lateral ventilation (Buesseler et al., 1991). Considering the characteristic residence time for the upper (about 5 years; Lee et al. (2002)) and intermediate pycnocline (9-15 years; Ivanov et al. (2000)), it is appropriate to consider such temporal average to 325 characterize the oxygen signature of the CIL member composing the mixture of pycnocline waters*"
- L324: "years" instead of "yr" (twice)  -> Done(x2)
- L327: "indeed shows a generally deeper oxygenation" -> Modified as suggested
- L328: "characteristic for"  -> Corrected (also for P2 in previous lines).
- L333: "current" or "present" instead of "actual" -> "current"
- Fig. 8 caption: "T-S"; as stated I don't think Fig. 8b is needed (and it's not used in the text) -> See specific comment on Fig 8.
- L336: "detail"; "asking, e.g. how" -> Done(x2)
- L337: remove "on" before "planktonic" -> Done
- L338: "detected" or "identified" instead of "depicted"  -> "identified"
- L340: remove "considering";  -> Done
    - this ("ventilating ... terms") is a bit repetitive from the previous lines
    -  -> We opted to maintain this, considering that the above lines provide specific examples of BGC terms, while this sentence stresses the need to consider in concert "physical" ventilation and BGC terms, in general. We simplified however, by retaining only "biogeochemical terms" instead of "biogeochemical production/consumption terms"
- L345: "contributed 42% and 58%" (and remove "for 42% and 58%" at end of sentence)-> Done
- L346: "assessments" or "an assessment" -> "a corresponding assessment"
- L351: Maybe "For this purpose"? -> Modified as suggested
- L358: "standard regime" -> Modified as suggested
- L361: "cannot" -> Corrected
- L363: "hydrodynamic" -> Corrected
- L364/365: move "through convective ventilation" to end of sentence  -> Modified as suggested
- L365-370: Very long sentence. I would try to split it after making the point that changes in CIL formation will affect oxygen. Then you can state that this will in turn have an impact on ecosystem structure/functioning given the structuring role of oxygen (please add a reference for this, too).
    - A reference was added as suggested. However, despite the reviewer' suggestion we prefer to maintain the sentence as is, since it really merges all required arguments to support the important message it delivers.
- L367: "dominant" -> Corrected
- L371: remove "on" after "impacts" -> Corrected
- Fig. A1: Please add to the caption what the different acronyms mean ("sd", "RMS")
    - -> The caption now reads : "*Statistics of comparison between the different data sources: Pearson correlation coefficient (r); root-mean-square deviation (RMS); average bias (bias); number of overlapping years between time series (N) and, on diagonal elements, the standard deviation of each time series (sd)*".

[revised manuscript text omitted]
  ($T >$ 8.35°C and $\sigma$ within 14.5–15 $\mathrm{kg\,m^{-3}}$) that was extremely rare in previous periods (see density contours on Fig. 7a,b,c,d; two-dimensional density estimates  obtained with R function MASS : kde2).

    As indicated by Stanev et al. (2019), this trend may lead to the disappearance of a characteristic layer of the Black Sea, that
330 constituted a major component of its thermo-haline structure and constrained the exchanges between surface, subsurface and intermediate layers. In particular, the authors highlight surface and subsurface salinity trends that indicate recent occurrences of diapycnal mixing at the lower base of the intermediate layer, where waters are characterized by a strong reduction potential due to the presence of reduced iron and manganese species, ammonium and finally hydrogen sulfide (Pakhomova et al., 2014).

    On a decadal time scale, the average oxygen signature of a given isopycnal layer within the CIL depends on the frequency
335 of CIL formation events of sufficient intensity (Sect. 4), which is in line with the ventilation dynamics  described by Ivanov et al. (2000) for the upper pycnocline. Although inter-annual fluctuations in CIL formation rate still occur, the regime

[Figure]

**Figure 7.** (a-d) Potential temperature versus salinity (T-S diagram) for bottle, CTD, and Argo in-situ data available from the World Ocean Database for the period 1955–2020 (Boyer et al., 2013). Data from periods P1, P2, P3 and P4 are shown on panels a), b), c) and d), respectively. Black contours  delineate 90%, 75% and 50% of the observations for each period. (d) Historical oxygen records collected from the World Ocean Data base for the period 1955–2020 (Boyer et al., 2013), averaged for hexagonal bins of the T-S diagram. The 75% contour of P2 (yellow) and P4 (red) are repeated  to highlight the difference in  oxygenation state at a given density between  the two corresponding contrasted CIL regimes. For all panels, the isopycnal $\sigma$ layers are indicated by curved grey lines. The black dotted line locates the $T_{CIL} = 8.35°C$ criterion used to identify CIL waters.

shift analysis specifically describes a reduction in the average CIL cold content, which appears to be associated with a reduction in the frequency of significant ventilation events (Fig. 4), and therefore a potential decrease in the oxygen saturation signature in the lower part of the CIL.

340  Importantly, this reduction may also affect deeper layers of the Black Sea. Indeed, the mid-pycnocline $\sigma$ between about 14.6  16 kg m$^{-3}$) is formed by the two end-member mixing line between the CIL layer and the Bosporus inflow (Murray et al., 1991; Ivanov et al., 2000), which proceeds from the entrainment of CIL waters by the  Bosporus

345 inflow and subsequent lateral ventilation (Buesseler et al., 1991). Considering the characteristic residence time for the upper (about 5 years; Lee et al. (2002)) and intermediate pycnocline (9-15

years; Ivanov et al. (2000)), it is appropriate to consider such temporal average to characterize the oxygen signature of the CIL member composing the mixture of pycnocline waters.

 Displaying historical oxygen saturation data (1955–2020) on  the T-S diagram (Fig
350  7e) indeed shows a generally deeper oxygenation during high CIL    regimes than during low CIL regimes. For instance, oxygen saturation in the density range $\sigma = 14.8 - 15.2$ kg m$^{-3}$ lies in the range of 30–70% in the T-S region that is characteristic to P2, while it only reaches 10–50 % in the T-S region characteristic to P4. This indicates that the analysis linking oxygenation and CIL formation
355 for the recent period (Sect. 4), can be extended to larger time scales by considering changes in the frequency of significant CIL formation events. Thus, the depth until which the reduction in CIL formation may impact on biogeochemical balance of the Black Sea (by affecting oxygenation level) will depend on the period over which the  current ventilation regime will remain.

360   Indeed, it is important to highlight that the current regime may not necessarily be a new steady regime - even though it has been identified as such by the change point analysis - but that it could be part of a transient downward trend that started in the mid-1990s. The reason why it appeared important to us to oppose non-linear regime dynamics to smooth linear and sine trends models is that the recent CIL
365 dynamics, when depicted by the regime change approach, is not a slow trending one, but suggests a step change towards a new phenology for the intermediate Black Sea. Only future observations may now confirm or infirm the relevance of the proposed regime shift paradigm.

[revised manuscript text omitted]

To understand how global warming impacts  the marine deoxygenation dynamics is a worldwide concern. The relatively fast and clear response that stems from the specific Black Sea geomorphology makes it a natural laboratory to study this dependency and related phenomena, although the specificity of this morphology also limits the direct transposition of Black Sea results to the global ocean. 
[revised manuscript text omitted]